# Landscape analysis of an improved power method for tensor decomposition

**Joe Kileel**
UT Austin

**Timo Klock**
Simula Research Laboratory

**João M. Pereira**
UT Austin

## Abstract

In this work, we consider the optimization formulation for symmetric tensor decomposition recently introduced in the Subspace Power Method (SPM) of Kileel and Pereira. Unlike popular alternative functionals for tensor decomposition, the SPM objective function has the desirable properties that its maximal value is known in advance, and its global optima are exactly the rank-1 components of the tensor when the input is sufficiently low-rank. We analyze the non-convex optimization landscape associated with the SPM objective. Our analysis accounts for working with noisy tensors. We derive quantitative bounds such that any second-order critical point with SPM objective value exceeding the bound must equal a tensor component in the noiseless case, and must approximate a tensor component in the noisy case. For decomposing tensors of size $D^{\times m}$, we obtain a near-global guarantee up to rank $\widetilde{o}(D^{\lfloor m/2 \rfloor})$ under a random tensor model, and a global guarantee up to rank $\mathcal{O}(D)$ assuming deterministic frame conditions. This implies that SPM with suitable initialization is a provable, efficient, robust algorithm for low-rank symmetric tensor decomposition. We conclude with numerics that show a practical preferability for using the SPM functional over a more established counterpart.

## 1 Introduction

From applied and computational mathematics [27] to machine learning [36] and multivariate statistics [29] to many-body quantum systems [41], high-dimensional data sets arise that are naturally organized into higher-order arrays. Frequently these arrays, known as hypermatrices or *tensors*, are decomposed into low-rank representations. In particular, the real symmetric CANDECOMP/PARAFAC (CP) decomposition [13] is often appropriate:

$$T = \sum_{i=1}^{K} \lambda_i a_i^{\otimes m}. \tag{1}$$

Here, we are given $T$, a real symmetric tensor of size $D^{\times m}$. The goal is to expand $T$ as a sum of $K$ rank-1 terms, coming from scalar/vector pairs $(\lambda_i, a_i) \in \mathbb{R} \times \mathbb{R}^D$. Importantly, the number of terms $K$ must be minimal possible for the given tensor, in which case $K$ is called the *rank* of the input $T$.

When $m > 2$ and $K = \mathcal{O}(D^m)$, CP decompositions are generically unique (up to permutation and scaling) by fundamental results in algebraic geometry [10]. An actionable interpretation [3] is that CP decomposition infers well-defined *latent variables* $\{(\lambda_i, a_i) : i \in [K]\}$ encoded by $T$. Indeed in learning applications, where symmetric tensors are formed from statistical moments (higher-order covariances) or multivariate derivatives (higher-order Hessians), CP decomposition has enabled parameter estimation for mixtures of Gaussians [20, 35], generalized linear models [34], shallow neural networks [19, 24, 42], deeper networks [17, 18, 30], hidden Markov models [5], among others.

Unfortunately, CP decomposition is NP-hard in the worst case [23]. In fact, it is believed to possess a computational-to-statistical gap [7], with efficient algorithms expected to exist for random tensors

35th Conference on Neural Information Processing Systems (NeurIPS 2021).

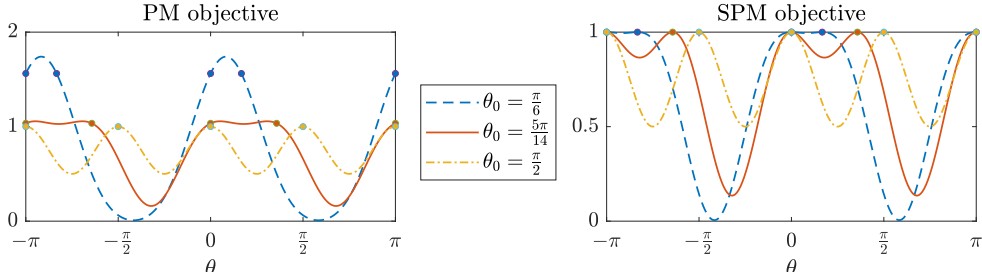

Figure 1: Illustration of **PM-P** objective (left) and **SPM-P** objective (right) for $D = K = 2$ and $m = 4$, where $\mathbb{S}^1$ is mapped to $(-\pi, \pi]$. We set $a_1 = (1, 0)$, $a_2 = (\cos\theta_0, \sin\theta_0)$ for different values of $\theta_0$. If $a_1$ and $a_2$ are not orthogonal ($\theta_0 \neq \frac{\pi}{2}$), then $a_1$ and $a_2$ do not coincide with global or local maximizers of **PM-P**. Too much correlation ($\theta_0 \leq \frac{\pi}{3}$) leads to a collapsing of the two distinct maxima into one isolated maximum for **PM-P**. On the other hand, **SPM-P** is not affected in this way by correlation: local maxima coincide with global maxima which are the tensor components $a_1, a_2$.

only up to rank $K = \mathcal{O}(D^{m/2})$. For this reason, and due to its non-convexity and high-dimensionality, CP decomposition has served theoretically as a key testing ground for better understanding mysteries of non-convex optimization landscapes. To date, this focus has been on the non-convex program

$$\max_{x \in \mathbb{R}^D : \|x\|_2 = 1} \left\langle T, x^{\otimes m} \right\rangle. \tag{PM-P}$$

We label the problem **PM-P**, standing for Power Method Program, because projected gradient ascent applied to **PM-P** corresponds to the Shifted Symmetric Higher-Order Power Method of Kolda and Mayo [28]. Important analyses of **PM-P** include Ge and Ma's [21] and the earlier [4] on overcomplete tensors, as well as [33] for low-rank tensors, and [31] which studied **PM-P** assuming the tensor components form a unit-norm tight frame with low incoherence.

In this paper, we perform an analysis of the non-convex optimization landscape associated with the Subspace Power Method (SPM) for computing symmetric tensor decompositions. The first and third authors introduced SPM in [25]. This method is based on the following non-convex program:

$$\max_{x \in \mathbb{R}^D : \|x\|_2 = 1} F_{\mathcal{A}}(x), \tag{SPM-P}$$

where $F_{\mathcal{A}}(x) := \|P_{\mathcal{A}}(x^{\otimes n})\|_F^2$, $n := \lceil m/2 \rceil$, $\mathcal{A} := \mathrm{Span}\{a_1^{\otimes n}, \dots, a_K^{\otimes n}\}$,

and $P_{\mathcal{A}} : (\mathbb{R}^D)^{\otimes n} \to \mathcal{A}$ is orthogonal projection with respect to Frobenius inner product.

Note that **SPM-P** is a particular polynomial optimization problem of degree $2n$ on the unit sphere.

There are at least two motivating reasons to study the optimization landscape of **SPM-P**. Firstly, it was observed in numerical experiments in [25] that the SPM algorithm is competitive within its applicable rank range of $K = \mathcal{O}(D^{\lfloor m/2 \rfloor})$. It gave a roughly one-order of magnitude speed-up over the decomposition methods in [26] as implemented in Tensorlab [40], while matching the numerical stability of FOOBI [14]. Thus SPM is a *practical* algorithm. Secondly, from a theory standpoint, the program **SPM-P** has certain desirable properties which **PM-P** lacks. Specifically for an input tensor $T = \sum_{i=1}^K \lambda_i a_i^{\otimes m}$ with rank $K \lesssim D^{\lfloor m/2 \rfloor}$ and Zariski-generic[1] $\{(\lambda_i, a_i)\}_{i=1}^K$, **SPM-P** is such that:

- Each component $\pm a_i$ is exactly a global maximum, and there are no other global maxima.
- The globally maximal value is known in advance to be exactly 1. So the objective value gives a certificate for global optimality, and non-global critical points can be discarded.

These properties were shown for **SPM-P** in [25], but both fail for **PM-P** (see Figure 1). Thus, **SPM-P** is more relevant *theoretically* than **PM-P** as a test problem for non-convex CP decomposition.

**Prior theory.** In [25], it is proven that projected gradient ascent applied to **SPM-P**, initialized at almost all starting points with a constant explicitly-bounded step size, must converge to a second-order critical point of **SPM-P** at a power rate or faster. However this left open the possible existence

---

[1]*Zariski-generic* means that the failure set can be described by the vanishing non-zero polynomials [22], so in particular, has Lebesgue measure 0.

of *spurious second-order critical points*, i.e., second-order points with reasonably high objective value that are not global maxima (unequal to, and possibly distant from, each CP component $\pm a_i$). Such critical points could pose trouble for the successful optimization of **SPM-P**. Furthermore all theory for **SPM-P** in [25] was restricted to the *clean case:* that is, when the input tensor $T$ is exactly of a sufficiently low rank $K$. The analysis for **PM-P** in [21, 31, 33] also assume noiseless inputs. However, tensors arising in practice are invariably noisy due, e.g., to sampling or measurement errors.

**Main contributions.** We perform a landscape analysis of **SPM-P** by characterizing all second-order critical points, using suitable assumptions on $a_1, \ldots, a_K$. Under deterministic frame conditions on $a_1, \ldots a_K$, which are satisfied by mutually incoherent vectors, near-orthonormal systems, and random ensembles of size $K = \mathcal{O}(D)$, Theorem 7 shows that all second-order critical points of **SPM-P** coincide exactly with $\pm a_i$. Theorem 16 shows the same result for overcomplete random ensembles of size $K = \tilde{o}(D^{\lfloor m/2 \rfloor})$, however requiring an additional superlevel set condition to exclude maximizers with vanishing objective values. Both results extend to noisy tensor decomposition, where SPM is applied to a perturbation $\hat{T} \approx T$. In this setting, second-order critical points with objective values exceeding $\mathcal{O}(\|\hat{T} - T\|_F)$ are $\mathcal{O}(\|\hat{T} - T\|_F)$-near to one of the components $\pm a_i$. We also show in Lemma 9 that spurious local maximizers (with lower objective values) do exist in the noisy case.

The results imply a clear separation of the functional landscape between near-global maximizers, with objective values close to 1, and spurious local maximizers with small objective value. Hence, the SPM objective can be used to validate a final iterate of projected gradient ascent in the noisy case. In Theorem 18, we combine our landscape analysis with bounds on error propagation incurred during SPM's deflation steps. This gives guarantees for *end-to-end* tensor decomposition using SPM.

Lastly, we expose the relation between **PM-P** and **SPM-P**. Specifically, **SPM-P** can be expressed as **PM-P** with the appropriate insertion of the inverse of the Grammian $(G_n)_{ij} := \langle a_i, a_j \rangle^n$. The resulting de-biasing effect on the local maximizers with respect to the components $\pm a_i$ (cf. Figure 1) is responsible for many advantages of **SPM-P**. Along the way, we state a conjecture about the minimal eigenvalue of $G_n$ when $a_i$ are i.i.d. uniform on the sphere, which may be of independent interest.

## 2 Notation

**Vectors and matrices.** When $x$ is a vector, $\|x\|_p$ is the $\ell_p$-norm ($p \in \mathbb{R}_{\geq 1} \cup \{\infty\}$). For $x, y \in \mathbb{R}^D$, the entrywise (or Hadamard) product is $x \odot y \in \mathbb{R}^D$, and the entrywise power is $x^{\odot s} := x \odot \ldots \odot x \in \mathbb{R}^D$ ($s \in \mathbb{N}$). When $M$ is a matrix, $\|M\|_2$ is the spectral norm. If $M$ is real symmetric, $\mu_j(M)$ is the eigenvalue of $M$ that is the $j$-th largest in absolute value. We denote the identity by $\mathsf{Id}_D \in \mathbb{R}^{D \times D}$.

**Tensors.** A real tensor of length $D$ and order $m$ is an array of size $D \times D \times \ldots \times D$ ($m$ times) of real numbers. Write $\mathcal{T}_D^m := (\mathbb{R}^D)^{\otimes m} \cong \mathbb{R}^{D^m}$ for the space of tensors of size $D^{\times m}$. Meanwhile, $\mathrm{Sym}(\mathcal{T}_D^m) \subseteq \mathcal{T}_D^m$ is the subspace of symmetric tensors (i.e., tensors unchanged by any permutation of indices). The Frobenius inner product and norm are denoted by $\langle \cdot, \cdot \rangle$ and $\|\cdot\|_F$, respectively. Given any linear subspace of tensors $\mathcal{A} \subseteq \mathcal{T}_D^m$, let $P_\mathcal{A} : \mathcal{T}_D^m \to \mathcal{A}$ denote the orthogonal projector onto $\mathcal{A}$ with respect to $\langle \cdot, \cdot \rangle$. In the case $\mathcal{A} = \mathrm{Sym}(\mathcal{T}_D^m)$, the projector $P_{\mathrm{Sym}(\mathcal{T}_D^m)}$ is the symmetrization operator, $\mathrm{Sym} : \mathcal{T}_D^m \to \mathrm{Sym}(\mathcal{T}_D^m)$. Given $T \in \mathcal{T}_D^{m_1}$ and $S \in \mathcal{T}_D^{m_2}$, the tensor (or outer) product is $T \otimes S \in \mathcal{T}_D^{m_1 + m_2}$, defined by $(T \otimes S)_{i_1, \ldots, i_{m_1 + m_2}} := T_{i_1, \ldots, i_{m_1}} S_{i_{m_1 + 1}, \ldots, i_{m_2}}$. For $T \in \mathcal{T}_D^m$ and $s \in \mathbb{N}$, the tensor power is $T^{\otimes s} := T \otimes \ldots \otimes T \in \mathcal{T}_D^{sm}$. For $T \in \mathcal{T}_D^{m_1}$, $S \in \mathcal{T}_D^{m_2}$ with $m_1 \geq m_2$, the contraction $T \cdot S \in \mathcal{T}_D^{m_1 - m_2}$ is defined by $(T \cdot S)_{i_1, \ldots, i_{m_1 - m_2}} := \sum_{j_1, \ldots, j_{m_2}} T_{i_1, \ldots, i_{m_1 - m_2}, j_1, \ldots, j_{m_2}} S_{j_1, \ldots, j_{m_2}}$. Let $\mathrm{Reshape}(T, [d_1, \ldots, d_\ell])$ be the function that reshapes the tensor $T$ to have dimensions $d_1, \ldots, d_\ell$, as in corresponding Matlab/NumPy commands.

**Other.** The unit sphere in $\mathbb{R}^D$ is $\mathbb{S}^{D-1}$, and $\mathrm{Unif}(\mathbb{S}^{D-1})$ is the associated uniform probability distribution. Given a function $f : \mathbb{R}^D \to \mathbb{R}$, the Euclidean gradient and Hessian matrix at $x \in \mathbb{R}^D$ are $\nabla f(x) \in \mathbb{R}^D$ and $\nabla^2 f(x) \in \mathrm{Sym}(\mathcal{T}_D^2)$. The Riemannian gradient and Hessian with respect to $\mathbb{S}^{D-1}$ at $x \in \mathbb{S}^{D-1}$ are $\nabla_{\mathbb{S}^{D-1}} f(x)$ and $\nabla^2_{\mathbb{S}^{D-1}} f(x)$ (see [1]). Write $\mathrm{Span}$ for linear span, $[K] := \{1, \ldots, K\}$, and $|A|$ for the cardinality of a finite set $A$. Lastly, we use asymptotic notation freely.

# 3   Symmetric tensor decomposition via Subspace Power Method

In this section, we outline the tensor decomposition method SPM of [25], and provide basic insights on the program **SPM-P**. Throughout we assume that $m \geq 3$ is an integer and define $n := \lceil m/2 \rceil$.

**SPM algorithm.** The input is a tensor $\hat{T} \in \mathrm{Sym}(\mathcal{T}_D^m)$, with the promise that $\hat{T} \approx T = \sum_{i=1}^K \lambda_i a_i^{\otimes m}$ for $\{(\lambda_i, a_i)\}_{i=1}^K$ Zariski-generic and $K \leq \binom{D+n-1}{n} - D$ (if $m$ is even) and $K \leq D^n$ (if $m$ is odd). As a first step, SPM obtains the orthogonal projector $P_{\hat{\mathcal{A}}} : \mathcal{T}_D^m \to \hat{\mathcal{A}}$ that projects onto the column span of $\mathrm{Reshape}(\hat{T}, [D^n, D^{m-n}])$, by using matrix singular value decomposition. Provided that $\hat{T} \approx T$, the associated subspace approximation error $\hat{\mathcal{A}} \approx \mathcal{A} = \mathrm{span}\{a_1^{\otimes n}, \ldots, a_K^{\otimes n}\}$ defined by

$$\Delta_{\mathcal{A}} := \left\| P_{\mathcal{A}} - P_{\hat{\mathcal{A}}} \right\|_{F \to F} = \sup_{T \in \mathcal{T}_D^n, \, \|T\|_F = 1} \left\| P_{\mathcal{A}}(T) - P_{\hat{\mathcal{A}}}(T) \right\|_F, \tag{2}$$

can be bounded as follows. (Note that by [9, Lem. 2.3], we know $\Delta_{\mathcal{A}} \leq 1$ a priori.)

**Lemma 1** (Error in subspace). *Let $m \geq 3$, $n = \lceil \frac{m}{2} \rceil$, $T \in \mathrm{Sym}(\mathcal{T}_D^m)$ and assume that $M :=$ $\mathrm{Reshape}(T, [D^n, D^{m-n}])$ has exactly $K$ nonzero singular values $\sigma_1(M) \geq \ldots \geq \sigma_K(M) > 0$. Let $\hat{T} \in \mathrm{Sym}(\mathcal{T}_D^m)$, $\hat{M} := \mathrm{Reshape}(T, [D^n, D^{m-n}])$. Assume $\Delta_M := \|M - \hat{M}\|_2 < \sigma_K(M)$. Then*

$$\left\| P_{\mathrm{Im}(M)} - P_{\mathrm{Im}_K(\hat{M})} \right\|_2 \leq \frac{\Delta_M}{\sigma_K(M) - \Delta_M}, \tag{3}$$

*where $\mathrm{Im}(M) \subseteq \mathbb{R}^{D^n}$ denotes the image of $M$ and $\mathrm{Im}_K(\hat{M}) \subseteq \mathbb{R}^{D^n}$ denotes the subspace spanned by the $K$ leading left singular vectors of $\hat{M}$. In particular, if $T = \sum_{i=1}^K \lambda_i a_i^{\otimes 2n}$, $\mathcal{A} = \mathrm{span}\{a_i^{\otimes n} : i \in [K]\}$, $\dim(\mathcal{A}) = K$, and $\hat{\mathcal{A}}$ is the subspace spanned by $K$ leading tensorized left singular vectors of $\hat{M}$, the right-hand side of (3) upper-bounds $\Delta_{\mathcal{A}}$.*

**Remark 2.** If $T = \sum_{i=1}^K \lambda_i a_i^{\otimes m}$, one coefficient $\lambda_i$ is small and the vectors $\{a_i : i \in [K]\}$ are not too correlated, then the flattened tensor $\mathrm{Reshape}(T, [D^n, D^{m-n}])$ has a small eigenvalue. This makes estimating the corresponding eigenvector sensitive to noise. See Remark S.41 in the appendix.

Given $\hat{\mathcal{A}}$, SPM seeks one tensor component $a_i$ by solving the noisy variant of **SPM-P** defined by

$$\max_{x \in \mathbb{S}^{D-1}} F_{\hat{\mathcal{A}}}(x), \quad \text{where} \quad F_{\hat{\mathcal{A}}}(x) := \left\| P_{\hat{\mathcal{A}}}(x^{\otimes n}) \right\|_F^2. \tag{nSPM-P}$$

Starting from a random initial point $x_0 \sim \mathrm{Unif}(\mathbb{S}^{D-1})$, the projected gradient ascent iteration

$$x \leftarrow \frac{x + \gamma P_{\hat{\mathcal{A}}}(x^{\otimes n}) \cdot x^{\otimes(n-1)}}{\|x + \gamma P_{\hat{\mathcal{A}}}(x^{\otimes n}) \cdot x^{\otimes(n-1)}\|_2}, \tag{4}$$

with a constant step-size $\gamma$, is guaranteed to converge to a second-order critical point of **nSPM-P** almost surely by [25]. Here we require that the step-size $\gamma$ is less than an explicit upper bound given in [25]. Denoting by $\hat{a}_i$ the final iterate obtained by SPM, we accept the candidate approximate tensor component $\hat{a}_i$ if $F_{\hat{\mathcal{A}}}(\hat{a}_i)$ is large enough; otherwise we draw a new starting point $x_0$ and re-run (4).

Next given $\hat{a}_i$, SPM evaluates a deflation formula based on Wedderburn rank reduction [12] from matrix algebra to compute the corresponding weight $\hat{\lambda}_i$. Then, we update the tensor $\hat{T} \leftarrow \hat{T} - \hat{\lambda}_i \hat{a}_i^{\otimes m}$.

To finish the tensor decomposition, SPM performs the projected gradient ascent and deflation steps $K$ times to compute all of the tensor components and weights $\{(\hat{\lambda}_i, \hat{a}_i)\}_{i=1}^K$.

**Preparatory material about nSPM-P.** The goal of this paper is to show that second-order critical points of **nSPM-P** with reasonable function value must be near the global maximizers $\pm a_1, \ldots, \pm a_K$ of **SPM-P**, under suitable incoherence assumptions on the rank-one components $a_1, \ldots, a_K$. Naturally, the optimality conditions for **nSPM-P** play an important part in this analysis.

**Proposition 3** (Optimality conditions). *Let $x \in \mathbb{S}^{D-1}$ be first and second-order critical for **nSPM-P**. Then for each $z \in \mathbb{S}^{D-1}$ with $z \perp x$, we have*

$$P_{\hat{\mathcal{A}}}(x^n) \cdot x^{n-1} = F_{\hat{\mathcal{A}}}(x)x, \tag{5}$$

$$F_{\hat{\mathcal{A}}}(x) \geq n \| P_{\hat{\mathcal{A}}}(x^{n-1}z) \|_F^2 + (n-1) \langle P_{\hat{\mathcal{A}}}(x^n), x^{n-2}z^2 \rangle. \tag{6}$$

*Furthermore, for any $y \in \mathbb{S}^{D-1}$ we have*

$$F_{\hat{\mathcal{A}}}(x) \geq n \left\| P_{\hat{\mathcal{A}}}(x^{n-1}y) \right\|_F^2 + (n-1) \langle P_{\hat{\mathcal{A}}}(x^n), x^{n-2}y^2 \rangle - 2(n-1) F_{\hat{\mathcal{A}}}(x) \langle x, y \rangle^2. \tag{7}$$

In the analysis later, we make frequent use of expressing the objective $F_\mathcal{A}(x)$ using the Gram matrix

$$G_n \in \mathrm{Sym}(\mathcal{T}_D^2) \quad \text{defined by} \quad (G_n)_{ij} := \langle a_i^{\otimes n}, a_j^{\otimes n} \rangle = \langle a_i, a_j \rangle^n. \tag{8}$$

Under linear independence of the tensors $a_1^{\otimes n}, \ldots, a_K^{\otimes n}$, which is implied by our assumptions made later, the inverse $G_n^{-1}$ exists and the noiseless program **SPM-P** can be expressed as follows.

**Lemma 4.** *Let* $A := [a_1 | \ldots | a_K] \in \mathbb{R}^{D \times K}$ *and* $\{a_i^{\otimes n} : i \in [K]\}$ *be linearly independent. We have*

$$F_\mathcal{A}(x) = \left\| P_\mathcal{A}(x^{\otimes n}) \right\|_F^2 = \left( (A^\top x)^{\odot n} \right)^\top G_n^{-1} \left( (A^\top x)^{\odot n} \right). \tag{9}$$

Lemma 4 exposes the relation between **SPM-P** and **PM-P**. While **PM-P** can be rewritten $\langle T, x^{\otimes m} \rangle = \langle (A^\top x)^{\odot n}, (A^\top x)^{\odot n} \rangle$ if $m$ is even, **SPM-P** takes into account correlations among the tensors $a_1^{\otimes n}, \ldots, a_K^{\otimes n}$ and inserts the Grammian $G_n^{-1}$ into (9). Consequently, correlations among the tensor components are considered in **SPM-P**, without any a priori knowledge of the tensors $a_1^{\otimes n}, \ldots, a_K^{\otimes n}$.

In the special case of orthonormal systems, or more generally systems that resemble equiangular tight frames [16, 37], the **SPM-P** and **PM-P** objectives coincide up to shift and scaling.

**Lemma 5.** *Assume there exist* $\rho \in (-1, 1) \setminus \{\frac{-1}{K-1}\}$ *and* $M \in \mathbb{R}$ *such that* $\langle a_i, a_j \rangle^n = \rho$ *for all* $i \neq j$ *and* $\sum_{i \in [K]} \langle x, a_i \rangle^n = M$ *for all* $x \in \mathbb{S}^{D-1}$. *Denote* $A = [a_1 | \ldots | a_K] \in \mathbb{R}^{D \times K}$. *Then*

$$F_\mathcal{A}(x) = (1 - \rho)^{-1} \|A^\top x\|_{2n}^{2n} - \left( (1 - \rho)^2 + K\rho(1 - \rho) \right)^{-1} \rho M^2. \tag{10}$$

# 4 Main results

In this section, we present the main results about local maximizers of the **nSPM-P** program. Section 4.1 is tailored to low-rank tensor models with $K = \mathcal{O}(D)$ components that satisfy certain deterministic frame conditions. Section 4.2 then considers the overcomplete case $K = \widetilde{o}(D^{\lfloor m/2 \rfloor})$ in an average case scenario, where $a_1, \ldots, a_K$ are modeled as independent copies of an isotropic random vector.

## 4.1 Low-rank tensors under deterministic frame conditions

Motivated by frame constants in frame theory [11], we measure the incoherence of the ensemble $a_1, \ldots, a_K$ by scalars $\rho_s \in \mathbb{R}_{\geq 0}$, which are defined via

$$\rho_s := \sup_{x \in \mathbb{S}^{D-1}} \sum_{i=1}^{K} |\langle x, a_i \rangle|^s - 1. \tag{11}$$

They satisfy the order relation $\rho_s \leq \rho_{s'}$ for $s' \leq s$, due to $\|a_i\|_2 = 1$, and can be related to extremal eigenvalues of Grammians $G_s$ and $G_{\lfloor s/2 \rfloor}$ as shown in the following result.

**Lemma 6.** *Let* $\{a_i : i \in [K]\} \subseteq \mathbb{S}^{D-1}$ *and* $(G_s)_{ij} := \langle a_i, a_j \rangle^s$ *for* $s \in \mathbb{N}$. *Then*

$$1 - \rho_s \leq \mu_K(G_s) \leq \mu_1(G_s) \leq 1 + \rho_s \leq \mu_1(G_{\lfloor s/2 \rfloor}). \tag{12}$$

The characterization in Lemma 6 allows to compute bounds for $\rho_s$ for low-rank tensors with mutually incoherent components or rank-$\mathcal{O}(D)$ tensor with random components. We provide details on this in Remark 8 below, but first state the main guarantee about local maximizers using $\rho_2$ and $\rho_n$.

**Theorem 7** (Main deterministic result). *Let* $\{a_i : i \in [K]\} \subseteq \mathbb{S}^{D-1}$ *and* $\mathcal{A} = \mathrm{Span}\{a_i^n : i \in [K]\}$. *Let* $\hat{\mathcal{A}} \subseteq \mathrm{Sym}(\mathcal{T}_D^n)$ *be a perturbation of* $\mathcal{A}$ *with* $\Delta_\mathcal{A} = \|P_\mathcal{A} - P_{\hat{\mathcal{A}}}\|_{F \to F}$. *Let*

$$\tau := \frac{1}{6} - n^2 \rho_2 - (n^2 + n)\rho_n \quad \text{and} \quad \Delta_0 := \frac{2\tau}{2 + 4\tau + 3n^2}. \tag{13}$$

*Then, if* $\Delta_\mathcal{A} < \Delta_0$, *the program* **nSPM-P** *has exactly* $2K$ *second-order critical points in the superlevel set where*

$$F_{\hat{\mathcal{A}}}(x) \geq \frac{2 + 2\tau + 3n^2}{2\tau} \Delta_\mathcal{A}. \tag{14}$$

Each of these critical points is a strict local maximizer for $F_{\hat{\mathcal{A}}}$. Further for each such point $x^*$, there exists unique $i \in [K]$ and $s \in \{-1, 1\}$ such that

$$\|x^* - sa_i\|_2^2 \leq \frac{2\Delta_{\mathcal{A}}}{n}. \tag{15}$$

In the noiseless case ($\Delta_{\mathcal{A}} = 0$), if $\tau \geq 0$ then there are precisely $2K$ second-order critical points of **SPM-P** with positive functional value, and they are the global maximizers $sa_i$ ($i \in [K], s \in \{-1, 1\}$).

**Remark 8.** Using Lemma 6, we identify two situations where $\rho_2$ and $\rho_n$ can be bounded from above.

1. *Mutually incoherent ensembles.* Let $a_1, \ldots, a_K$ have mutual incoherence $\rho := \max_{i \neq j} |\langle a_i, a_j \rangle|$. Using Gershgorin's circle theorem, we obtain

   $$\rho_s \leq \mu_1(G_{\lfloor s/2 \rfloor}) - 1 \leq \max_{i \in [K]} \sum_{j \neq i} |\langle a_i, a_j \rangle|^{\lfloor s/2 \rfloor} \leq (K-1)\rho^{\lfloor s/2 \rfloor} \quad \text{for any } s \in \mathbb{N}_{\geq 2},$$

   which implies that the conditions of Theorem 7 are saisfied if $K\rho$ is sufficiently small. This setting is comparable to the analysis for **PM-P** in [33]. Moreover, if $K = D$ and $a_1, \ldots, a_K$ are mutually orthogonal, then $\rho_s = 0$ for each $s \geq 2$. Therefore Theorem 7 holds with $\tau = 1/6$ for orthogonally decomposable tensors [6].

2. *Low-rank random ensembles.* Let $a_1, \ldots, a_K$ be independent copies of an isotropic unit-norm random vector $a$ with sub-Gaussian norm $\mathcal{O}(1/\sqrt{D})$ (e.g., $a_i \sim \text{Unif}(\mathbb{S}^{D-1})$). With high probability, the ensemble can achieve arbitrarily small $\rho_2$, provided that $K \leq CD$ for a sufficiently small constant $C > 0$. The proof of this fact relies on $A = [a_1 | \ldots | a_K]$ satisfying, with high probability, the so-called $(K, \delta)$ restricted isometry property [38, Thm. 5.65] as defined in Definition 10 in the next section. We note that despite requiring conditions milder than those in the unit-norm tight frame analysis for **PM-P** in [31], we achieve a comparable scaling of $K = \mathcal{O}(D)$.

In the noiseless case where $\Delta_{\mathcal{A}} = 0$, Theorem 7 shows that all local maximizers coincide with global maximizers $\pm a_1, \ldots, \pm a_K$, provided the tensor components are sufficiently incoherent to ensure $\tau \geq 0$. In the noisy case, all local maximizers with objective values $F_{\hat{\mathcal{A}}}(x) \geq C(n, \tau)\Delta_{\mathcal{A}}$ are $\mathcal{O}(\Delta_{\mathcal{A}})$-close to the global optimizers of the noiseless objective $F_{\mathcal{A}}$, where the constant $C(n, \tau)$ increases as the incoherence of the vectors $a_1, \ldots, a_K$ shrinks. Unfortunately, the presence of spurious local maximizers of $F_{\hat{\mathcal{A}}}$ with small objective values cannot be avoided under a deterministic noise model, as the next result shows.

**Lemma 9.** *Let $\delta \in (0, 1)$, $a \in \mathbb{S}^{D-1}$ and $\mathcal{A} = \text{span}\{a^{\otimes n}\}$. Then there exists a subspace $\hat{\mathcal{A}} \subseteq \text{Sym}(\mathcal{T}_D^n)$ with $\dim(\hat{\mathcal{A}}) = 1$ and $\|P_{\mathcal{A}} - P_{\hat{\mathcal{A}}}\|_{F \to F} = \delta$ such that* **nSPM-P** *possesses a strict local maximizer of objective value exactly $\delta^2$.*

## 4.2 Average case analysis of overcomplete tensors

The overcomplete case with $K = \tilde{o}(D^{\lfloor m/2 \rfloor})$ falls outside the range of Theorem 7, because $\tau$ in (13) becomes negative when $K \gg D$. Instead, our analysis for the overcomplete case relies on $A = [a_1 | \ldots | a_K] \in \mathbb{R}^{D \times K}$ obeying the $(p, \delta)$-restricted isometry property (RIP) for $p = \mathcal{O}(D/\log(K))$.

**Definition 10.** *Let $A \in \mathbb{R}^{D \times K}$, $1 \leq p \leq K$ be an integer, and $\delta \in (0, 1)$. We say that $A$ is $(p, \delta)$-RIP if every $D \times p$ submatrix $A_p$ of $A$ satisfies $\|A_p^\top A_p - \text{Id}_p\|_2 \leq \delta$.*

A consequence of the RIP, which is particularly useful in the analysis of overcomplete tensor models, is that the correlation coefficients $\{\langle a_i, x \rangle : i \in [K]\}$ can naturally be split into two groups.

**Lemma 11** (RIP-induce partitioning of correlation coefficients)**.** *Suppose that $A = [a_1 | \ldots | a_K] \in \mathbb{R}^{D \times K}$ satisfies the $(p, \delta)$-RIP for $p = \lceil c_\delta D/\log(K) \rceil$. Let $\tilde{c}_\delta := (1 + \delta)/c_\delta$. Then for all $x \in \mathbb{S}^{D-1}$ there is a subset of indices $\mathcal{I}(x) \subseteq [K]$ with cardinality $p$ such that*

$$1 - \delta \leq \sum_{i \in \mathcal{I}(x)} \langle a_i, x \rangle^2 \leq 1 + \delta \quad \text{and} \quad \langle a_i, x \rangle^2 \leq \tilde{c}_\delta \frac{\log(K)}{D} \text{ for } i \notin \mathcal{I}(x), \tag{16}$$

Let us now collect all assumptions needed to analyze the overcomplete case.

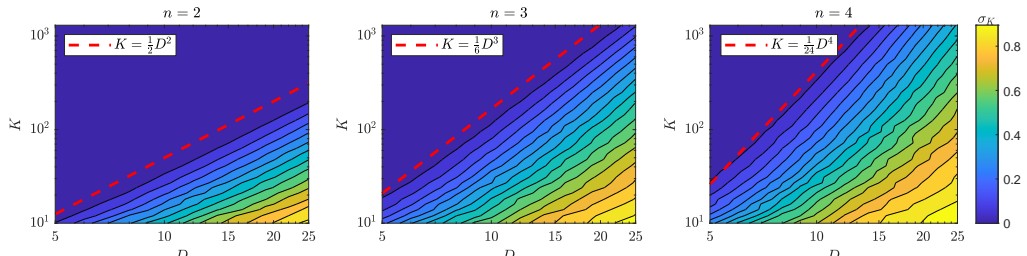

Figure 2: The average smallest eigenvalue of the Grammian $(G_n)_{ij} = \langle a_i, a_j \rangle^n$ for a random ensemble $a_1, \ldots, a_K$ consisting of independent copies of the random vector $a \sim \mathrm{Unif}(\mathbb{S}^{D-1})$, over 100 runs per $K, D, n$. From left to right, we consider different tensor orders $n = 2, 3, 4$. As long as $K \leq C(n)D^n$ for some constant $C(n)$ depending only on $n$ (red line), inverting the Grammian is well-posed and the smallest eigenvalue of $G_n$ is bounded from below, in accord with Conjecture 15.

**Definition 12.** Let $A := [a_1 | \ldots | a_K]$. We require the following assumptions:

    **A1** There exists $c_\delta > 0$, depending only on $\delta$, such that $A$ is $(\lceil c_\delta D / \log(K) \rceil, \delta)$-RIP.

    **A2** There exists $c_1 > 0$, independent of $K, D$, such that $\max_{i,j:i \neq j} \langle a_i, a_j \rangle^2 \leq c_1 \log(K)/D$.

    **A3** There exists $c_2 > 0$, independent of $K, D$, such that $\left\| G_n^{-1} \right\|_2 \leq c_2$.

To the best of our knowledge, there are no known deterministically constructed systems of size $K = \widetilde{o}(D^{\lfloor m/2 \rfloor})$ which satisfy **A1** - **A3** for small $\delta \ll 1$ and constants $c_\delta, c_1, c_2$ that do not depend on $K, D$. However, by modeling components via a sufficiently-spread random vector, i.e., by considering an average case scenario with $a_i \sim \mathrm{Unif}(\mathbb{S}^{D-1})$, they hold with high probability.

**Proposition 13** (**A1** - **A3** for random ensembles). *Let $a \sim \mathrm{Unif}(\mathbb{S}^{D-1})$ and assume $a_1, \ldots, a_K$ are $K$ independent copies of $a$, where $\log(K) = o(D)$. Fix an arbitrary constant $\delta \in (0, 1)$. There exist constants $C > 0$ and $D_0 \in \mathbb{N}$ depending only on $\delta$ such that for all $D \geq D_0$, and with probability at least $1 - K^{-1} - 2\exp(-C\delta^2 D)$, conditions **A1** and **A2** (with $c_1 \leq C$) hold. Furthermore, if $n = 2$ and $K = o(D^2)$, **A3** holds with probability $1 - C(eD/\sqrt{K})^{-C\sqrt{K}}$ for $c_2 \leq C$.*

**Remark 14** (**A3** when $n > 2$). Following [2, 15], we give a self-contained proof for **A3** when $n = 2$ in the appendix. We are currently not able to extend the proof to $n > 2$, because some technical tools such as the Hanson-Wright inequality [32] and an extension of [2, Thm. 3.3], which ensures the RIP for matrices whose columns have sub-exponential tails, have not yet been fully developed for random vectors with dependent entries and heavier tails (so-called $\alpha$-sub-exponential tails or sub-Weibull tails). However we strongly believe that **A3** holds for $n > 2$ and $K = o(D^n)$. For now, we formulate this as a conjecture, supplemented with numerical evidence presented in Figure 2. We also add that the conjecture, once proven, would complement recent advances on the well-posedness of random tensors with fewer statistical dependencies among the components [8, 39].

**Conjecture 15** (Grammians of independent symmetric rank-one tensors). *Let $a_1, \ldots, a_K$ be independent copies of the random vector $a \sim \mathrm{Unif}(\mathbb{S}^{D-1})$ and fix $\epsilon > 0$ arbitrarily. Then there exists some constant $\kappa_n > 0$ and an increasing function $\gamma_n : (0, \kappa_n) \to (0, 1)$, both depending only on $n$, such that $\gamma_n(\kappa) \to 1$ as $\kappa \to 0$, and if $K \leq \kappa D^n$ for some $\kappa < \kappa_n$ we have*

$$\mathbb{P}(\left\| G_n^{-1} \right\|_2 \geq \gamma_n(\kappa) - \epsilon) \to 1 \quad as \quad D \to \infty. \tag{17}$$

*In particular, if $K = o(D^n)$ then $\|G_n^{-1}\|_2 \to 1$ as $D \to \infty$.*

We now present our main theorem about local maximizers of **nSPM-P** in the overcomplete case.

**Theorem 16** (Main random overcomplete result). *Let $K, D \in \mathbb{N}$, define $\varepsilon_K := K \log^n(K)/D^n$, and suppose that $\lim_{D \to \infty} \varepsilon_K = 0$. Assume $a_1, \ldots, a_K \in \mathbb{R}^D$ satisfy **A1** - **A3** for some $\delta, c_1, c_2 > 0$. Then there exist $\delta_0$, depending only on $n$, $c_1$ and $c_2$, and $D_0, \Delta_0, C$, which depend additionally on $c_\delta$, such that if $\delta < \delta_0$, $D > D_0$, and $\Delta_{\mathcal{A}} \leq \Delta_0$, the program **nSPM-P** has exactly $2K$ second-order critical points in the superlevel set*

$$\left\{ x \in \mathbb{S}^{D-1} : F_{\hat{\mathcal{A}}}(x) \geq C\varepsilon_K + 5\Delta_{\mathcal{A}} \right\}. \tag{18}$$

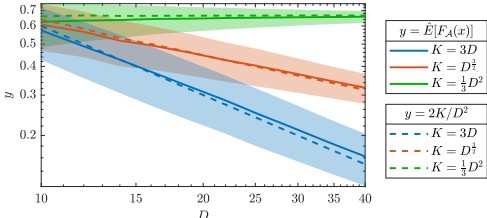

Figure 3: The empirical average of $F_{\mathcal{A}}(x)$ over 10000 trials of a random ensemble $a_1, \ldots, a_K, x$ consisting of $K + 1$ independent copies of the random vector $a \sim \mathrm{Unif}(\mathbb{S}^{D-1})$ for different scalings of $K$ and $D$. The shaded areas indicate plus/minus one empirical standard deviation.

*Each of these critical points is a strict local maximizer for $F_{\hat{\mathcal{A}}}$. Further for each such point $x^*$, there exists unique $i \in [K]$ and $s \in \{-1, 1\}$ such that*

$$\|x^* - sa_i\|_2^2 \leq \frac{2\Delta_{\mathcal{A}}}{n}. \tag{19}$$

*In particular, in the noiseless case ($\Delta_{\mathcal{A}} = 0$), the second-order critical points in the superlevel set (18) are exactly the global maximizers $sa_i$ ($i \in [K], s \in \{-1, 1\}$).*

By Theorem 16, all non-degenerate local maximizers in the superlevel set (18) are close to global maximizers of the noiseless objective $F_{\mathcal{A}}$. In the noiseless case $\Delta_{\mathcal{A}} = 0$, they coincide with global maximizers and the right hand side in (18) tends to 0 as $D \to \infty$. Hence, we have a clear separation between the global maximizers $\pm a_1, \ldots, \pm a_K$ and degenerate local maximizers with vanishing objective value, so that the objective value acts as a certificate for the validity of an identified maximizer.

Theorem 16 is not a global guarantee because a random starting point $x_0 \sim \mathrm{Unif}(\mathbb{S}^{D-1})$, used for starting the projected gradient ascent iteration (4), is, with high probability, not contained in (18).

**Remark 17** (Objective value at random initialization)**.** For a random sample $x_0 \sim \mathrm{Unif}(\mathbb{S}^{D-1})$ in the clean case $\Delta_{\mathcal{A}} = 0$, we empirically observe the objective value $F_{\mathcal{A}}(x_0) \approx CK/D^n$ as illustrated in Figure 3. As such, we conjecture $\mathbb{E}_{x_0, a_1, \ldots, a_k \sim \mathrm{Unif}(\mathbb{S}^{D-1})}[F_{\mathcal{A}}(x_0)] = \mathcal{O}(\frac{K}{D^n})$. Comparing the objective to the level set condition in Theorem 16,

$$\left\{ x \in \mathbb{S}^{D-1} : F_{\mathcal{A}}(x) \geq C \frac{K \log^n(K)}{D^n} + 5\Delta_{\mathcal{A}} \right\}, \tag{20}$$

a random starting point $x_0$ therefore falls short of satisfying (20) by a $\log^n(K)$-factor only. In this sense, Theorem 16 furnishes a "near-global" guarantee for **nSPM-P** in the random overcomplete case.

### 4.3 End-to-end tensor decomposition

By combining our landscape analyses with bounds for error propagation during deflation, we obtain a theorem about end-to-end tensor decomposition using the SPM algorithm. That is, under the conditions of Theorem 7 or 16, a tweaking of SPM (Algorithm 1 in the appendix) recovers the entire CP decomposition exactly in the noiseless case ($\hat{T} = T$). In the noisy regime, it obtains an approximate CP decomposition, and we bound the error in terms of $\Delta_{\mathcal{A}}$. Due to space constraints, we leave precise descriptions of Algorithm 1 and our deflation bounds to the supplementary material.

**Theorem 18** (Main result on end-to-end tensor decomposition)**.** *Let $T = \sum_{i=1}^{K} \lambda_i a_i^{\otimes m} \in \mathrm{Sym}(\mathcal{T}_D^m)$ and $M := \mathrm{Reshape}(T, [D^n, D^{m-n}])$. Let $\sigma_1(M) \geq \ldots \geq \sigma_K(M)$ be the singular values of $M$, and assume $\sigma_K(M) > 0$. For other tensor $\hat{T} \in \mathrm{Sym}(\mathcal{T}_D^m)$, let $\hat{M} = \mathrm{Reshape}(\hat{T}, [D^n, D^{m-n}])$, assume $\Delta_M := \|M - \hat{M}\|_2 < \frac{1}{2}\sigma_K(M)$ and let $\hat{\Delta}_{\mathcal{A}} = \frac{\Delta_M}{\sigma_K(M) - \Delta_M}$. Suppose that $T$ satisfies the assumptions of either Theorem 7 or Theorem 16, define $\Delta_0$ as in the corresponding theorem statement and let $\ell(\Delta_{\mathcal{A}})$ be the corresponding level set threshold.[2] Then there exist constants $C_1, C_2$, not depending on $\hat{T}$ or $\Delta_M$, such that if we define $\tilde{\Delta}_{\mathcal{A}} := C_1\hat{\Delta}_{\mathcal{A}} + C_2\sqrt{\hat{\Delta}_{\mathcal{A}}}$ the following holds. Assume*

---

[2]In both theorem statements, the level set threshold depends on $\Delta_{\mathcal{A}}$.

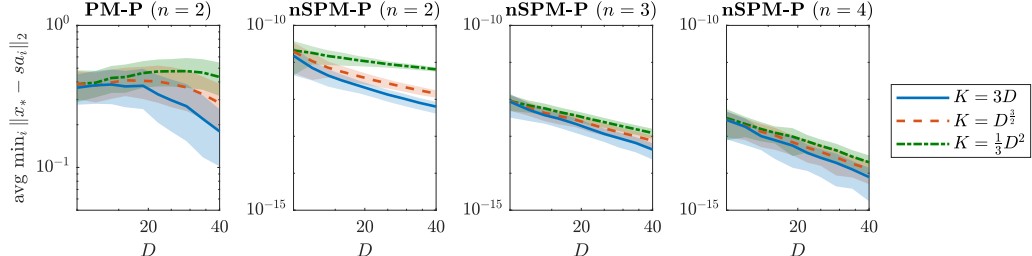

(a) Recovering individual tensor components $a_i$ using **PM-P** and **nSPM-P** in the noiseless case;

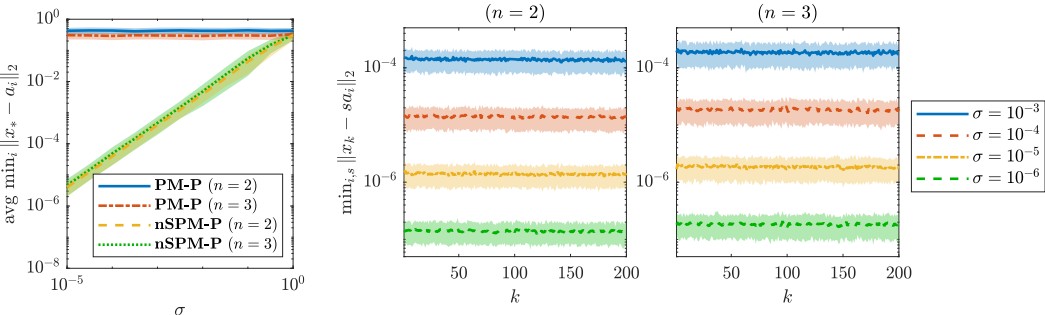

(b) **PM-P** and **nSPM-P** applied to noisy tensors;

(c) Error of individual recovery with **nSPM-P** after $k$ deflations.

Figure 4: Numerical results for symmetric tensor decomposition using **PM-P** versus **nSPM-P**. The shaded areas indicate plus/minus one standard deviation. The experiments are described in Section 5.

$\tilde{\Delta}_{\mathcal{A}} < \Delta_0$ *and when we apply Algorithm 1 to $\hat{T}$ each run of projected gradient ascent converges to a point with functional value at least $\ell(\tilde{\Delta}_{\mathcal{A}})$. Then, letting $(\hat{a}_i, \hat{\lambda}_i)_{i \in [K]}$ be the output of Algorithm 1 applied to $\hat{T}$, there exist a permutation $\pi \in \mathrm{Perm}(K)$ and signs $s_1, \ldots, s_K \in \{-1, +1\}$ with*

$$\|s_i a_{\pi(i)} - \hat{a}_i\|_2 \leq \sqrt{\frac{2\hat{\Delta}_{\mathcal{A}}}{n}} \quad and \quad \left| \frac{s_i^m}{\lambda_{\pi(i)}} - \frac{1}{\hat{\lambda}_i} \right| \leq \frac{2\sqrt{m/n}}{\sigma_K(M)} \sqrt{\hat{\Delta}_{\mathcal{A}}} + \frac{4}{\sigma_K(M)} \hat{\Delta}_{\mathcal{A}} \quad \forall i \in [K].$$

*In particular, in the noiseless case ($\Delta_M = 0$), Algorithm 1 returns the exact CP decomposition of $T$.*

We conclude that the Subspace Power Method, with an initialization scheme for **nSPM-P** that gives $x \in \mathbb{R}^D$ where $F_{\hat{\mathcal{A}}}(x) > \ell(\Delta_{\mathcal{A}})$, is a guaranteed algorithm for low-rank tensor decomposition.

## 5 Numerical experiments

Here we present numerical experiments that corroborate the theoretical findings of Section 4. We illustrate that SPM identifies exact tensor components in the noiseless case (in contrast to PM), and that SPM behaves robustly in noisy tensor decomposition. We use the implementation of SPM of the first and third authors, available at `https://github.com/joaompereira/SPM`, which is licensed under the MIT license. All of our experiments presented below may be conducted on a standard laptop computer within a few hours. For further numerical experiments, we refer the reader to [25], where SPM was tested in a variety of other scenarios, justifying it as a possible replacement for state-of-the-art symmetric tensor decomposition methods such as FOOBI [14] or Tensorlab [40].

**Global optimizers and noise robustness.** In the first set of experiments, we are interested in the recovery of individual tensor components $a_i$ for different $D$, $K$ and $m$ from noisy approximately low rank tensors. With $m = 2n$ as the tensor order, we create noiseless tensors with $a_i \sim \mathrm{Unif}(\mathbb{S}^{D-1})$ as the tensor components and $\lambda_i = \sqrt{D^m/K} \, \tilde{\lambda}_i$ as the tensor weights, where $\tilde{\lambda}_i \sim \mathrm{Unif}([1/2, 2])$. This way, the variance of each entry of the tensor is about 1. We construct noisy tensors $\hat{T} \approx T$ by adding independent copies of $\epsilon \sim \mathcal{N}(0, m!\sigma^2)$ to each entry of the tensor and then project onto

$\mathrm{Sym}(\mathcal{T}_D^m)$. The entries with all distinct indices of the projected noise tensor have variance $\sigma^2$; the variance of the remaining entries is also a multiple of $\sigma^2$ and the number of such entries is $o(D^m)$.

After constructing the tensor, we sample 10 points from $\mathrm{Unif}(\mathbb{S}^{D-1})$ as initializations for projected gradient ascent applied to **PM-P** and **nSPM-P**. We iterate until convergence and record the distance between the final iterate and the closest $\pm a_i$ among the tensor components. Averages and standard deviations (represented by shaded areas) over 1000 distances (10 distances per tensor, 100 tensors) are depicted in Figures 4a and 4b. In Figure 4a, we use noiseless tensors with $\sigma = 0$, while in Figure 4b $\sigma$ ranges from $10^{-5}$ to 1, and the plots for $4^{\text{th}}$ and $6^{\text{th}}$ order tensors are superposed. We set $D = 20, K = 100$ for the $4^{\text{th}}$ order tensor ($n = 2$) and $D = 10, K = 100$ for the $6^{\text{th}}$ order ($n = 3$).

Figure 4a illustrates that projected gradient ascent applied to **nSPM-P** always converges to the global maxima in the noiseless case (up to numerical error), provided the scaling is $K = \mathcal{O}(D^{\lfloor m/2 \rfloor}) = \mathcal{O}(D^2)$ and the constant adheres to the rank constraints pointed out in [25, Prop. 4.2]. This is in agreement with Theorem 16, which shows local maxima with sufficiently large objective have to coincide with global maximizers. In the noisy case, Figure 4b illustrates the robustness of **nSPM-P**, giving an error of $\mathcal{O}(\sigma)$. In contrast to **nSPM-P**, the **PM-P** objective suffers from a large bias and does not recover exact tensor components. The effect of the bias even dominates errors induced by moderately-sized entrywise noise.

**Deflation with SPM.** In the second experiment, we recover complete tensors by using the deflation procedure described in [25]. We are mostly interested in the noisy case, since deflation with exact $a_i$'s, as identified by **nSPM-P** in the noiseless case, does not induce any additional error.

We test $4^{\text{th}}$ order tensors ($n = 2$) with $D = 40, K = 200$ and $6^{\text{th}}$ order tensors ($n = 3$) with $D = 15, K = 200$, vary the noise level $\sigma$, and construct random tensors as in the previous experiment. Figure 4c plots the average error over 100 repetitions of successively recovered tensor components, where the $x$-axis ranges from the first recovered component at 1 to the last component at index 300. The figure illustrates that **nSPM-P** combined with modified deflation allows for recovering all tensor components up to an error of $\mathcal{O}(\sigma)$. Surprisingly, we do not observe error propagation, despite the fact that noisy recovered tensor components are being used within each deflation step.

## 6 Conclusion

We presented a quantitative picture for the optimization landscape of a recent formulation [25] of the non-convex, high-dimensional problem of symmetric tensor decomposition. We identified different assumptions on the tensor components $a_1, \ldots, a_K$ and bounds on the rank $K$ so that all second-order critical points of the optimization problem **SPM-P** with sufficiently high functional value must equal one of the input tensor's CP components. In Theorem 7 the assumptions were deterministic frame and low-rank conditions, while in Theorem 16 the hypotheses were random components and an overcomplete rank. Our proofs accommodated noise in the input tensor's entries, and we obtained robust results for only by analyzing the program **nSPM-P**. Our analysis has algorithmic implications. As the Subspace Power Method of [25] is guaranteed to converge to second-order critical points, by combining with analysis of deflation, it follows that SPM (with sufficient initialization) is provable under our assumptions. In Theorem 18 we gave guarantees for end-to-end decomposition using SPM.

Compared to the usual power method functional, the novelty of the SPM functional is the de-biasing role played by the inverse of a Grammian matrix recording correlations between rank-1 tensors (recall Eq. (8)). This Grammian matrix is responsible for many of the SPM functional's desirable properties, but it also complicated our analysis. We showed that there are theoretical and numerical advantages in using the SPM functional for tensor decomposition over its usual power method counterpart.

This paper suggests several directions for future research:

- What are the average-case properties of the Grammian matrix $G_n$? We formulated Conjecture 15 about the minimal eigenvalue of $G_n$. How about the other eigenvalues?

- In the random overcomplete setting, if we assume no noise ($\Delta_{\mathcal{A}} = 0$) then can we dispense with the superlevel condition in Theorem 16? This would give a fully global guarantee.

- Why do we see no error propagation when using deflation and sequential solves of **nSPM-P** for CP decomposition? Can errors accumulate if we choose the noise deterministically?

## Acknowledgements and funding disclosure

We thank Massimo Fornasier for suggesting this collaboration in the beginning, after a fortuitous exchange in Oaxaca, Mexico at the *Computational Harmonic Analysis and Data Science* workshop (October 2019). J.K. acknowledges partial support from start-up grants provided by the College of Natural Sciences and Oden Institute for Computational Engineering and Sciences at UT Austin.

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
