# Supplementary Material: Landscape analysis of an improved power method for tensor decomposition

In these appendices, we supply proofs for the statements made in the main body of the paper. We indicate references to sections or statements in the main body of the paper by adding M to the reference. Results solely within this supplementary material are labeled with a leading S.

**Organization.**

## A   Overview of main proofs

Before commencing with the exact details, let us first give a high-level overview of the main steps in the proofs of Theorems M.7 and M.16. We recall the definition $A := [a_1 | \ldots | a_K] \in \mathbb{R}^{D \times K}$. The proofs consist of three steps. We begin by plugging in $y = a_i$, where $i$ is the index such that $|\langle a_i, x \rangle| = \|A^\top x\|_\infty$, in (M.7). Then using either incoherence constants $\rho_s$ and $\rho_n$ for Theorem M.7, or using Assumptions M.**A1**-M.**A3** for Theorem M.16, a suitable rearrangement of (M.7) immediately gives

$$\|A^\top x\|_\infty^2 \geq \begin{cases} 1 - 2\rho_2 - 2\frac{\rho_n}{1-2\rho_n} - 3\frac{\Delta_A}{F_A(x)}, & \text{for Theorem M.7}, \\ 1 - 2\delta - \bar{C}\left(\frac{\varepsilon_K}{F_A(x)}\right)^{\frac{1}{n}} - 3\frac{\Delta_A}{F_A(x)}, & \text{for Theorem M.16}, \end{cases} \tag{S.1}$$

where the constant $\bar{C}$ in the second case depends on $n$, $c_1$ and $c_2$. For Theorem M.7, this immediately suggests the largest correlation coefficient satisfies $\|A^\top x\|_\infty \geq 1 - \mathcal{O}(\rho_2 + \rho_n + \Delta_A)$, provided $F_A(x) \gg \Delta_A$ as assumed in the statement. In the case of Theorem M.16, and within the level set $F_{\hat{A}}(x) \geq C\varepsilon_K + 5\Delta_A$, (S.1) first implies that $\|A^\top x\|_\infty$ is bounded from below by a constant that is independent of $D, K$. However, by $F_A(x) \geq \|A^\top x\|_\infty^{2n}$, we can update the lower bound on $F_A(x)$ and re-use it again in (M.7). The resulting bootstrap argument implies $\|A^\top x\|_\infty \geq 1 - \mathcal{O}(\varepsilon_K^{1/n} + \Delta_A)$, i.e., there must be at least one large correlation coefficient.

The second step is to show that the Riemannian Hessian $\nabla_{\mathbb{S}^{D-1}}^2 F_{\hat{A}}$ is negative definite on spherical caps around the tensor components $\pm a_i$. We prove, uniformly for all unit norm $z \perp x$, the bound

$$\frac{1}{2n} z^\top \nabla_{\mathbb{S}^{D-1}}^2 \hat{f}(x) z \leq \begin{cases} -1 + n\frac{2-\rho_n}{1-\rho_n}\eta + \frac{3n-2}{1-\rho_n}\rho_n + 4\Delta_A, & \text{for Theorem M.7}, \\ -1 + \bar{C}\sqrt{\eta} + \tilde{C}\varepsilon_K^{\frac{n-1}{2n}} + 4\Delta_A, & \text{for Theorem M.16}, \end{cases} \tag{S.2}$$

where $\eta := 1 - \max_{i \in [K]} \langle a_i, x \rangle^{2n}$, and $\bar{C}, \tilde{C}$ depend only on $n, c_1, c_2$ and $c_\delta$. Hence, the Riemannian Hessian is negative definite and $F_{\hat{\mathcal{A}}}$ is strictly concave in spherical caps around $\pm a_i$.

In a third step, we consider a spherical cap $B_{sa_i}$ ($s \in \{-1, 1\}$) around $sa_i$. By compactness, a local maximizer $x^*$ of $F_{\hat{\mathcal{A}}}$ must exist within the cap $B_{sa_i}$ and by strict concavity it must be unique. Moreover, its objective is bounded from below via $F_{\hat{\mathcal{A}}}(x^*) \geq F_{\hat{\mathcal{A}}}(a_i) \geq 1 - \Delta_{\mathcal{A}}$. Using a Taylor expansion of the function $g(t) = F_{\hat{\mathcal{A}}}(\gamma(t))$ where $\gamma$ is the geodesic path from $x^*$ to $a_i$, and combining this with the upper bound in (S.2), we obtain $\|x^* - sa_i\|_2^2 \leq 2\Delta_{\mathcal{A}}/n$. Since we can start the argument with a spherical cap which is strictly larger than $\{y \in \mathbb{S}^{D-1} : \langle y, sa_i \rangle \geq 1 - \Delta_{\mathcal{A}}/n\}$, it follows that $x^*$ is not on the boundary of $B_{sa_i}$, and thus $x^*$ is a local maximizer of **nSPM-P**.

We complete the proof by noting that all local maximizers of **nSPM-P** contained in the superlevel set, which is specified in the respective theorem, satisfy $\|A^\top x\|_\infty \geq 1 - \epsilon$, where $\epsilon$ depends on $\Delta_{\mathcal{A}}$ and $\rho_2, \rho_n$, or $\varepsilon_K$. In particular, they are contained in spherical caps $B_{sa_i}$ constructed in the third step.

# B   Setup and notation

**Additional notation.** Besides the notation in Section M.2, we will use the following in the rest of the supplementary materials.

- For tensors $S, T$, we abbreviate their tensor product via concatenation, $ST := S \otimes T$, and the tensor power by $S^n := S^{\otimes n}$. (It will be clear from context when concatenation denotes matrix multiplication instead.)

- We use $\text{Vec}(T)$ to denote the vectorization of a tensor $T \in \mathcal{T}_D^n$, i.e., $\text{Vec}(T) := \text{Reshape}(T, D^n)$. Also, $\text{Tensor}(-) := \text{Reshape}(-, [D, \ldots, D])$ indicates the inverse mapping, which tensorizes a vector.

- We denote the (columnwise) Khatri-Rao power by a bullet: if $a \in \mathbb{R}^D$ then $a^{\bullet n} := \text{Vec}(a^{\otimes n}) \in \mathbb{R}^{D^n}$. If $A = [a_1 | \ldots | a_K] \in \mathbb{R}^{D \times K}$ then $A^{\bullet n} := [a_1^{\bullet n} | \ldots | a_K^{\bullet n}] \in \mathbb{R}^{D^n \times K}$.

- We write the Tucker tensor product as follows: for $T \in \mathcal{T}_D^n$ and $M^{(1)}, \ldots, M^{(n)} \in \mathbb{R}^{E \times D}$, the Tucker product is denoted $T \times_1 M^{(1)} \times_2 \ldots \times_n M^{(n)} \in \mathcal{T}_E^n$ and defined by

$$\left( T \times_1 M^{(1)} \times_2 \ldots \times_n M^{(n)} \right)_{i_1, \ldots, i_n} := \sum_{j_1=1}^D \ldots \sum_{j_n=1}^D T_{j_1, \ldots, j_n} M_{i_1, j_1}^{(1)} \ldots M_{i_n, j_n}^{(n)}.$$

The next definition serves to collect together the key quantities appearing later in our proofs.

**Definition S.1.** Let $\{a_i : i \in [K]\} \subseteq \mathbb{S}^{D-1}$ be a given set of unit-norm vectors. Write $A = [a_1 | \ldots | a_K] \in \mathbb{R}^{D \times K}$. Consider the corresponding set of tensors $\{a_i^n : i \in [K]\}$, and the subspace $\mathcal{A} = \text{Span}\{a_i^n : i \in [K]\} \subseteq \text{Sym}(\mathcal{T}_D^n)$. Let $P_{\mathcal{A}} : \mathcal{T}_D^n \to \mathcal{A}$ denote orthogonal projection onto $\mathcal{A}$. Write $A^{\bullet n} := [\text{Vec}(a_1^n) | \ldots | \text{Vec}(a_K^n)] \in \mathbb{R}^{D^n \times K}$ for the $n$-th columnwise Khatri-Rao power of $A$.

1. Define the $n$-th *Grammian matrix* to be $G_n = (A^{\bullet n})^\top A^{\bullet n} \in \text{Sym}(\mathcal{T}_K^2)$. Thus

$$(G_n)_{ij} := \langle a_i^n, a_j^n \rangle = \langle a_i, a_j \rangle^n \qquad \forall i, j \in [K]. \tag{S.3}$$

2. Let $x \in \mathbb{S}^{D-1}$. Define the *correlation coefficients* of $x$ with respect to $\{a_i : i \in [K]\}$ to be $\zeta = \zeta(x) := A^\top x \in \mathbb{R}^K$. Thus

$$\zeta_i := \langle x, a_i \rangle \qquad \forall i \in [K]. \tag{S.4}$$

3. Let $x, y \in \mathbb{S}^{D-1}$ and $s \in \{0, 1, \ldots, n\}$. Define the *correlation coefficients* of $x^s y^{n-s}$ with respect to $\{a_i^n : i \in [K]\}$ to be $\eta = \eta(x^s y^{n-s}) = (A^{\bullet n})^\top \text{Vec}(x^s y^{n-s}) \in \mathbb{R}^K$. Thus

$$\eta_i := \langle x^s y^{n-s}, a_i^n \rangle = \langle x, a_i \rangle^s \langle y, a_i \rangle^{n-s} \qquad \forall i \in [K]. \tag{S.5}$$

4. Let $x \in \mathbb{S}^{D-1}$. Define the *expansion coefficients* of $P_{\mathcal{A}}(x^n)$ with respect to $\{a_i^n : i \in [K]\}$ to be $\sigma = \sigma(x) \in \mathbb{R}^K$ so that the following expansion holds

$$P_{\mathcal{A}}(x^n) =: \sum_{i=1}^{K} \sigma_i a_i^n. \tag{S.6}$$

(If $a_1^n, \ldots, a_K^n$ are linearly independent, then Eq. (S.6) uniquely defines $\sigma$; otherwise, we fix once and for all a choice of $\sigma$ so that (S.6) holds.)

5. Let $x \in \mathbb{S}^{D-1}$. For $i \in [K]$ and $s \in \{0, \ldots, n\}$, define the *remainder term* for $P_{\mathcal{A}}(x^n)$ with respect to $x^{n-s}a_i^s$ to be $R_{i,s} = R_{i,s}(x) \in \mathbb{R}$ so that the following equation holds

$$\langle P_{\mathcal{A}}(x^n), x^{n-s}a_i^s \rangle =: \sigma_i \zeta_i^{n-s} + R_{i,s}. \tag{S.7}$$

Last, define the maximal $s$-th *remainder size* to be $R_s = R_s(x) := \max_{i \in [K]} |R_{i,s}| \in \mathbb{R}$.

**Remark S.2.** Arguably all the quantities in Definition S.1 are natural to consider, with the possible exception of $R_{i,s}$ and $R_s$. In fact these terms play a key role in our proofs; see inequality (S.44) in Proposition S.18. We call $R_{i,s}$ a "remainder" because, under the assumptions of Theorems M.7 and M.16, as we will see it holds that $\langle P_{\mathcal{A}}(x^n), x^{n-s}a_i^s \rangle \approx \sigma_i \zeta_i^{n-s}$, whence $R_{i,s}$ and $R_s$ are small.

## C   Proofs of lemmas in Sections M.3 and M.4

In this section, we prove the various isolated results stated in Sections M.3 and M.4 of the main submission.

### C.1   Proof of Lemma M.1

**Lemma S.3** (= Lemma M.1). *Let $m \geq 3$, $n = \lceil \frac{m}{2} \rceil$, $T \in \mathrm{Sym}(\mathcal{T}_D^m)$ and assume $M := \mathrm{Reshape}(T, [D^n, D^{m-n}])$ has exactly $K$ nonzero singular values $\sigma_1(M) \geq \ldots \geq \sigma_K(M) > 0$. Let $\hat{T} \in \mathrm{Sym}(\mathcal{T}_D^m)$, $\hat{M} := \mathrm{Reshape}(T, [D^n, D^{m-n}])$. Assume $\Delta_M := \|M - \hat{M}\|_2 < \sigma_K(M)$. Then*

$$\left\| P_{\mathrm{Im}(M)} - P_{\mathrm{Im}_K(\hat{M})} \right\|_2 \leq \frac{\Delta_M}{\sigma_K(M) - \Delta_M}, \tag{S.8}$$

*where $\mathrm{Im}(M) \subseteq \mathbb{R}^{D^n}$ denotes the image of $M$ and $\mathrm{Im}_K(\hat{M}) \subseteq \mathbb{R}^{D^n}$ denotes the subspace spanned by the $K$ leading left singular vectors of $\hat{M}$. In particular, if $T = \sum_{i=1}^{K} \lambda_i a_i^{\otimes 2n}$, $\mathcal{A} = \mathrm{span}\{a_i^{\otimes n} : i \in [K]\}$, $\dim(\mathcal{A}) = K$, and $\hat{\mathcal{A}}$ is the subspace spanned by $K$ leading tensorized left singular vectors of $\hat{M}$, the right-hand side of (S.8) upper-bounds $\Delta_{\mathcal{A}}$.*

*Proof.* We note that (S.8) is valid for any matrices $W, \hat{W} \in \mathbb{R}^{p \times q}$, such that the rank of $W$ is $r$. That is, defining $\Delta_W := \|W - \hat{W}\|_2 < \sigma_r(W)$ and $\mathrm{Im}(W), \mathrm{Im}_r(\hat{W})$ as in the statement, we have

$$\left\| P_{\mathrm{Im}(W)} - P_{\mathrm{Im}_r(\hat{W})} \right\|_2 \leq \frac{\Delta_W}{\sigma_r(W) - \Delta_W}. \tag{S.9}$$

We first show the result if $W, \hat{W}$ are symmetric matrices. In that case, we denote the eigenvalue decomposition of $W, \hat{W}$ as

$$W = (V_1 \quad V_2) \begin{pmatrix} \Lambda_1 & 0 \\ 0 & \Lambda_2 \end{pmatrix} \begin{pmatrix} V_1^\top \\ V_2^\top \end{pmatrix} \quad \text{and} \quad \hat{W} = (\hat{V}_1 \quad \hat{V}_2) \begin{pmatrix} \hat{\Lambda}_1 & 0 \\ 0 & \hat{\Lambda}_2 \end{pmatrix} \begin{pmatrix} \hat{V}_1^\top \\ \hat{V}_2^\top \end{pmatrix},$$

where $\Lambda_1$ and $\hat{\Lambda}_1$ are diagonal matrices with the largest $K$ eigenvalues, in magnitude, of $W$ and $\hat{W}$ on the diagonal, respectively. Letting $\Sigma_1$ be a diagonal matrix with the singular values of $W$, we have $|\Lambda_1| = \Sigma_1$, and the same holds for $\hat{W}$. From the eigenvector decomposition, we can write $P_{\mathrm{Im}(W)} = V_1 V_1^\top$ and $P_{\mathrm{Im}_r(\hat{W})} = \hat{V}_1 \hat{V}_1^\top$. Lemma 2.3 in [5] implies that

$$\left\| P_{\mathrm{Im}(W)} - P_{\mathrm{Im}_r(\hat{W})} \right\|_2 = \|V_1^\top \hat{V}_2\|_2.$$

Then (S.9) holding for symmetric matrices follows from

$$\Delta_W = \|W - \hat{W}\|_2 \geq \|V_1^\top (W - \hat{W})\hat{V}_2\|_2 = \|\Lambda_1 V_1^\top \hat{V}_2 - V_1^\top \hat{V}_2 \hat{\Lambda}_2\|$$
$$\geq \|\Lambda_1 V_1^\top \hat{V}_2\|_2 - \|V_1^\top \hat{V}_2 \hat{\Lambda}_2\|_2 \geq \sigma_r(W)\|V_1^\top \hat{V}_2\|_2 - \|\hat{\Lambda}_2\|_2 \|V_1^\top \hat{V}_2\|_2$$
$$\geq (\sigma_r(W) - \Delta_W) \left\|P_{\mathrm{Im}(W)} - P_{\mathrm{Im}_r(\hat{W})}\right\|_2.$$

Here we used the following facts:

- $\|W - \hat{W}\|_2 \geq \|V_1^\top (W - \hat{W})\hat{V}_2\|_2$ since $V_1$ and $V_2$ have orthonormal columns.

- Since $\hat{V}_2^\top V_1(\Lambda_1^2 - \sigma_r(W)^2 \mathsf{Id}_r)V_1^\top \hat{V}_2$ is positive semi-definite, we have

$$\|\Lambda_1 V_1^\top \hat{V}_2\|_2 = \sqrt{\|\hat{V}_2^\top V_1 \Lambda_1^2 V_1^\top \hat{V}_2\|_2} \geq \sigma_r(W)\sqrt{\|\hat{V}_2^\top V_1 V_1^\top \hat{V}_2\|_2} = \sigma_r(W)\|V_1^\top \hat{V}_2\|_2.$$

- By Weyl's inequality [13], it holds $\|\hat{\Lambda}_2\| \leq \Delta_W$.

Now, if $W, \hat{W}$ are not symmetric, we apply the result for the symmetric matrices

$$H = \begin{pmatrix} 0 & W \\ W^\top & 0 \end{pmatrix} \quad \text{and} \quad \hat{H} = \begin{pmatrix} 0 & \hat{W} \\ \hat{W}^\top & 0 \end{pmatrix}.$$

Denote the singular vector decomposition of $W$ and $\hat{W}$ by $W = U\Sigma V^\top$ and $\hat{W} = \hat{U}\hat{\Sigma}\hat{V}^\top$. We can further split $U = (U_1, U_2)$, $\Sigma = \mathrm{Blockdiag}(\Sigma_1, \Sigma_2)$ and $V = (U_1, U_2)$, where $U_1, \Sigma_1, V_1$ correspond to the largest $K$ singular values of $W$, and split $\hat{U}, \hat{\Sigma}, \hat{V}$ analogously. The eigen-decomposition of $H$ is related to the singular vector decomposition of $W$ as follows

$$H = \begin{pmatrix} \frac{1}{\sqrt{2}}U & \frac{1}{\sqrt{2}}U \\ \frac{1}{\sqrt{2}}V & -\frac{1}{\sqrt{2}}V \end{pmatrix} \begin{pmatrix} \Sigma & 0 \\ 0 & -\Sigma \end{pmatrix} \begin{pmatrix} \frac{1}{\sqrt{2}}U^\top & \frac{1}{\sqrt{2}}V^\top \\ \frac{1}{\sqrt{2}}U^\top & -\frac{1}{\sqrt{2}}V^\top \end{pmatrix}, \tag{S.10}$$

and an analogous relation holds between $\hat{H}$ and $\hat{W}$. It can be checked that this is an eigen-decomposition by using that $W = U\Sigma V^\top$ to show that the RHS in (S.10) is equal to $H$, and use that $U$ and $V$ have orthonormal columns to show

$$\begin{pmatrix} \frac{1}{\sqrt{2}}U & \frac{1}{\sqrt{2}}U \\ \frac{1}{\sqrt{2}}V & -\frac{1}{\sqrt{2}}V \end{pmatrix}^\top \begin{pmatrix} \frac{1}{\sqrt{2}}U & \frac{1}{\sqrt{2}}U \\ \frac{1}{\sqrt{2}}V & -\frac{1}{\sqrt{2}}V \end{pmatrix} = \mathsf{Id}_{2r}.$$

Thus, if $W$ has rank $r$, $H$ has rank $2r$, and since $H$ is symmetric,

$$\left\|P_{\mathrm{Im}(H)} - P_{\mathrm{Im}_{2r}(\hat{H})}\right\|_2 \leq \frac{\Delta_H}{\sigma_{2r}(H) - \Delta_H} \tag{S.11}$$

Regarding the right-hand side, denote $\delta W = \hat{W} - W$. Equation (S.10) implies $\sigma_{2r}(H) = |\lambda_{2r}(H)| = \sigma_r(W)$, while

$$\Delta_H = \|H - \hat{H}\|_2 = \left\|\begin{pmatrix} 0 & \delta W \\ \delta W^\top & 0 \end{pmatrix}\right\|_2 = \left\|\begin{pmatrix} 0 & \delta W \\ \delta W^\top & 0 \end{pmatrix}^2\right\|_2^{\frac{1}{2}}$$
$$= \left\|\begin{pmatrix} \delta W \delta W^\top & 0 \\ 0 & \delta W^\top \delta W \end{pmatrix}\right\|_2^{\frac{1}{2}} = \max\{\|\delta W \delta W^\top\|_2, \|\delta W^\top \delta W\|_2\}^{\frac{1}{2}}$$
$$= \|\delta W\|_2 = \Delta_W$$

On the left-hand side, we have

$$P_{\mathrm{Im}(H)} = \begin{pmatrix} \frac{1}{\sqrt{2}}U_1 & \frac{1}{\sqrt{2}}U_1 \\ \frac{1}{\sqrt{2}}V_1 & -\frac{1}{\sqrt{2}}V_1 \end{pmatrix} \begin{pmatrix} \frac{1}{\sqrt{2}}U_1^\top & \frac{1}{\sqrt{2}}V_1^\top \\ \frac{1}{\sqrt{2}}U_1^\top & -\frac{1}{\sqrt{2}}V_1^\top \end{pmatrix} = \begin{pmatrix} U_1 U_1^\top & 0 \\ 0 & V_1 V_1^\top \end{pmatrix},$$

thus

$$\left\| P_{\mathrm{Im}(H)} - P_{\mathrm{Im}_{2r}(\hat{H})} \right\|_2 = \left\| \begin{pmatrix} U_1 U_1^\top - \hat{U}_1 \hat{U}_1^\top & 0 \\ 0 & V_1 V_1^\top - \hat{V}_1 \hat{V}_1^\top \end{pmatrix} \right\|_2$$

$$= \max \left\{ \left\| P_{\mathrm{Im}(W)} - P_{\mathrm{Im}_r(\hat{W})} \right\|_2, \left\| P_{\mathrm{Im}(W^\top)} - P_{\mathrm{Im}_r(\hat{W}^\top)} \right\|_2 \right\}$$

Replacing these in (S.11), we show (S.9) when $W, \hat{W}$ are not symmetric.

$$\left\| P_{\mathrm{Im}(W)} - P_{\mathrm{Im}_r(\hat{W})} \right\|_2 \leq \left\| P_{\mathrm{Im}(H)} - P_{\mathrm{Im}_{2r}(\hat{H})} \right\|_2 \leq \frac{\Delta_H}{\sigma_{2r}(H) - \Delta_H} = \frac{\Delta_W}{\sigma_r(W) - \Delta_W}.$$

The bound for $\Delta_{\mathcal{A}}$ follows from

$$\Delta_{\mathcal{A}} = \sup_{T \in \mathcal{T}_D^n, \, \|T\|_F = 1} \left\| P_{\mathcal{A}}(T) - P_{\hat{\mathcal{A}}}(T) \right\|_F = \sup_{u \in \mathbb{R}^{D^n}, \, \|u\|_2 = 1} \left\| P_{\mathrm{Im}(M)}(u) - P_{\mathrm{Im}_K(\hat{M})}(u) \right\|_2$$

$$= \left\| P_{\mathrm{Im}(M)} - P_{\mathrm{Im}_K(\hat{M})} \right\|_2. \qquad \square$$

## C.2 Proof of Lemma M.4

**Lemma S.4 (= Lemma M.4).** *Let $A := [a_1 | \dots | a_K] \in \mathbb{R}^{D \times K}$ and $\{a_i^{\otimes n} : i \in [K]\}$ be linearly independent. We have*

$$F_{\mathcal{A}}(x) = \left\| P_{\mathcal{A}}(x^{\otimes n}) \right\|_F^2 = \left( (A^\top x)^{\odot n} \right)^\top G_n^{-1} \left( (A^\top x)^{\odot n} \right). \tag{S.12}$$

*Proof.* This is the case $n = s$ in (S.13) in Lemma S.5 below, as $\eta(x^n) = \zeta(x)^{\odot n} = (A^\top x)^{\odot n}$. $\square$

**Lemma S.5.** *Assume $\{a_i^n : i \in [K]\} \subseteq \mathrm{Sym}(\mathcal{T}_D^n)$ is linearly independent. Then in the setup of Definition S.1, we have the following formulas in terms of the inverse Grammian:*

$$\left\| P_{\mathcal{A}}(x^s y^{n-s}) \right\|_F^2 = \eta(x^s y^{n-s})^\top G_n^{-1} \eta(x^s y^{n-s}) \quad \text{for all } x, y \in \mathbb{S}^{D-1}, \tag{S.13}$$

$$\text{and} \quad \sigma(x) = G_n^{-1} \eta(x^n) = G_n^{-1}(\zeta(x)^{\odot n}) \quad \text{for all } x \in \mathbb{S}^{D-1}. \tag{S.14}$$

*Proof.* Let $A^{\bullet n} := [\mathrm{Vec}(a_1^{\otimes n}) | \dots | \mathrm{Vec}(a_K^{\otimes n})] \in \mathbb{R}^{D^n \times K}$. Up to reshapings, $P_{\mathcal{A}}$ is projection onto the column space of $A^{\bullet n}$. Since $A^{\bullet n}$ has full column rank by assumption, projection onto the column space of $A^{\bullet n}$ is represented by the matrix

$$A^{\bullet n} \left( (A^{\bullet n})^\top A^{\bullet n} \right)^{-1} (A^{\bullet n})^\top = A^{\bullet n} G_n^{-1} (A^{\bullet n})^\top.$$

It follows that for $T \in \mathcal{T}_D^n$, we have

$$P_{\mathcal{A}}(T) = \mathrm{Tensor} \left( A^{\bullet n} G_n^{-1} (A^{\bullet n})^\top \mathrm{Vec}(T) \right), \tag{S.15}$$

$$\|P_{\mathcal{A}}(T)\|_F^2 = \langle P_{\mathcal{A}}(T), T \rangle = \mathrm{Vec}(T)^\top A^{\bullet n} G_n^{-1} (A^{\bullet n})^\top \mathrm{Vec}(T). \tag{S.16}$$

Substituting $T = x^s y^{n-s}$ in (S.15) and using $(A^{\bullet n})^\top \mathrm{Vec}(T) = \eta(x^s y^{n-s})$ yields (S.13), while substituting $T = x^n$ into (S.16) gives (S.14). This completes the calculation. $\square$

## C.3 Proof of Lemma M.5

**Lemma S.6 (= Lemma M.5).** *Assume there exist $\rho \in (-1, 1) \setminus \{\frac{-1}{K-1}\}$ and $M \in \mathbb{R}$ such that $\langle a_i, a_j \rangle^n = \rho$ for all $i \neq j$ and $\sum_{i \in [K]} \langle x, a_i \rangle^n = M$ for all $x \in \mathbb{S}^{D-1}$. Denote $A = [a_1 | \dots | a_K] \in \mathbb{R}^{D \times K}$. Then*

$$F_{\mathcal{A}}(x) = (1 - \rho)^{-1} \|A^\top x\|_{2n}^{2n} - \left( (1 - \rho)^2 + K\rho(1 - \rho) \right)^{-1} \rho M^2.$$

*Proof.* Write $\zeta := \zeta(x) = A^\top x$. From $\langle a_i^n, a_j^n \rangle = \langle a_i, a_j \rangle^n = \rho$ whenever $i \neq j$ and $\langle a_i^n, a_i^n \rangle = \langle a_i, a_i \rangle^n = 1$ for each $i$, we can write the Grammian of $a_1^n, \ldots, a_K^n$ as

$$G_n = (1-\rho)\mathsf{Id}_K + \rho \mathbb{1}_K \otimes \mathbb{1}_K, \tag{S.17}$$

where $\mathbb{1}_K \in \mathbb{R}^D$ denotes the all-ones vector. By the Sherman-Morrison inversion formula,

$$G_n^{-1} = ((1-\rho)\mathsf{Id}_K + \rho \mathbb{1}_K \otimes \mathbb{1}_K)^{-1} = \frac{1}{1-\rho}\left(\mathsf{Id}_K + \frac{\rho}{1-\rho}\mathbb{1}_K \otimes \mathbb{1}_K\right)^{-1}$$

$$= \frac{1}{1-\rho}\left(\mathsf{Id}_K - \frac{\frac{\rho}{1-\rho}}{1+K\frac{\rho}{1-\rho}}\mathbb{1}_K \otimes \mathbb{1}_K\right) = \frac{1}{1-\rho}\mathsf{Id}_K - \frac{\rho}{(1-\rho)^2 + K\rho(1-\rho)}\mathbb{1}_K \otimes \mathbb{1}_K.$$

Using that $\sum_{i\in[K]}\langle x, a_i \rangle^n = M$ uniformly on $\mathbb{S}^{D-1}$, we have $\langle \mathbb{1}_K, \zeta(x)^{\odot n} \rangle = M$. By writing $F_\mathcal{A}$ as in Lemma S.4, we can complete the proof:

$$F_\mathcal{A}(x) = \zeta^{\odot n} G_n^{-1} \zeta^{\odot n} = \frac{1}{1-\rho}\|\zeta\|_{2n}^{2n} - \frac{\rho}{(1-\rho)^2 + K\rho(1-\rho)}\left(\sum_{i=1}^K \langle a_i, x \rangle^n\right)^2$$

$$= \frac{1}{1-\rho}\|\zeta\|_{2n}^{2n} - \frac{\rho M^2}{(1-\rho)^2 + K\rho(1-\rho)}. \qquad \square$$

### C.4 Proof of Lemma M.6

**Lemma S.7** (= Lemma M.6). *Let $\{a_i : i \in [K]\} \subseteq \mathbb{S}^{D-1}$ and $(G_s)_{ij} := \langle a_i, a_j \rangle^s$ for $s \in \mathbb{N}$. Then we have*

$$1 - \rho_s \leq \mu_K(G_s) \leq \mu_1(G_s) \leq 1 + \rho_s \leq \mu_1(G_{\lfloor s/2 \rfloor}).$$

*Proof.* By Gershgorin's circle theorem and the fact $\|a_i\|_2 = 1$, the eigenvalue $\mu_\ell(G_s)$ for each $\ell = 1, \ldots, K$ adheres to

$$|\mu_\ell(G_s) - 1| \leq \sum_{i:i\neq\ell} |\langle a_\ell, a_i \rangle|^s \leq \rho_s, \quad \text{hence} \quad \mu_1(G_s) \leq 1 + \rho_s \quad \text{and} \quad \mu_K(G_s) \geq 1 - \rho_s.$$

First, we suppose that $s$ is even. By a classic result in frame theory [6, Prop. 3.6.7], the extremal eigenvalues of the Grammian $G_{s/2}$ satisfy

$$\mu_K(G_{s/2}) \leq \sum_{i=1}^K \langle T, a_i^{s/2} \rangle^2 \leq \mu_1(G_{s/2}), \quad \text{for all unit-norm } T \in \text{span}\{a_i^{s/2} : i \in [K]\}. \tag{S.18}$$

The right-most inequality of (S.18) implies

$$\sum_{i=1}^K \langle T, a_i^{s/2} \rangle^2 = \sum_{i=1}^K \left\langle P_{\text{span}\{a_i^{s/2}:i\in[K]\}}(T), a_i^{s/2} \right\rangle^2 \leq \mu_1(G_{s/2}) \quad \text{for all unit-norm } T \in \mathcal{T}_D^{s/2}. \tag{S.19}$$

Substituting $T = x^{s/2}$ into (S.19) and using that $s$ is even, we obtain

$$\sum_{i=1}^K |\langle x, a_i \rangle|^s = \sum_{i=1}^K \langle x^{s/2}, a_i^{s/2} \rangle^2 \leq \mu_1(G_{s/2}) \quad \text{for all } x \in \mathbb{S}^{D-1},$$

whence $\rho_s \leq \mu_1(G_{s/2}) - 1$. If instead $s$ is odd, we may similarly derive the upper bound $\rho_s \leq \mu_1(G_{\lfloor s/2 \rfloor})$, so that we have the following relation in general

$$1 - \rho_s \leq \mu_K(G_s) \leq \mu_1(G_s) \leq 1 + \rho_s \leq \mu_1(G_{\lfloor s/2 \rfloor}). \qquad \square$$

## C.5   Proof of Lemma M.9

**Lemma S.8** (= Lemma M.9). *Let $\delta \in (0,1)$, $a \in \mathbb{S}^{D-1}$ and $\mathcal{A} = \mathrm{span}\{a^{\otimes n}\}$. Then there exists a subspace $\hat{\mathcal{A}} \subseteq \mathrm{Sym}(\mathcal{T}_D^n)$ with $\dim(\hat{\mathcal{A}}) = 1$ and $\|P_{\mathcal{A}} - P_{\hat{\mathcal{A}}}\|_{F \to F} = \delta$ such that **nSPM-P** possesses a strict local maximizer of objective value exactly $\delta^2$.*

*Proof.* This argument relies on Lemma S.9, which is proven independently in the next section. Choose $b \in \mathbb{S}^{D-1}$ with $b \perp a$. Define $S := \sqrt{1 - \delta^2}a^n - \delta b^n \in \mathrm{Sym}(\mathcal{T}_D^n)$, and consider the corresponding subspace $\hat{\mathcal{A}} := \mathrm{span}\{S\}$.

For each $T \in \mathcal{T}_D^n$ with $\|T\|_F = 1$, using $\|a^n\|_F = \|b^n\|_F = 1$, $\langle a^n, b^n \rangle = 0$ and $\|S\|_F = 1$,

$$
\begin{aligned}
\left\|P_{\mathcal{A}}(T) - P_{\hat{\mathcal{A}}}(T)\right\|_F^2 &= \left\|\langle a^n, T\rangle a^n - \langle \sqrt{1-\delta^2}a^n - \delta b^n, T\rangle(\sqrt{1-\delta^2}a^n - \delta b^n)\right\|_F^2 \\
&= \left\|\langle \delta^2 a^n + \delta\sqrt{1-\delta^2}b^n, T\rangle a^n + \langle \delta\sqrt{1-\delta^2}a^n - \delta^2 b^n, T\rangle b^n\right\|_F^2 \\
&= \langle \delta^2 a^n + \delta\sqrt{1-\delta^2}b^n, T\rangle^2 + \langle \delta\sqrt{1-\delta^2}a^n - \delta^2 b^n, T\rangle^2 \\
&= \delta^2\langle a^n, T\rangle^2 + \delta^2\langle b^n, T\rangle^2.
\end{aligned}
$$

This quadratic is maximized over the unit sphere in $\mathcal{T}_D^n$ at any unit-norm $T \in \mathrm{span}\{a^n, b^n\}$, where its value is $\delta^2$. It follows $\|P_{\mathcal{A}} - P_{\hat{\mathcal{A}}}\|_{F \to F} = \delta$.

Also, we have

$$
F_{\hat{\mathcal{A}}}(b) = \|\langle S, b^n\rangle S\|_F^2 = \langle S, b^n\rangle^2 = \delta^2.
$$

We now verify that $b$ is a local maximizer of $F_{\hat{\mathcal{A}}}$ by checking the optimality conditions. By (S.20),

$$
\begin{aligned}
\nabla_{\mathbb{S}^{D-1}}F_{\hat{\mathcal{A}}}(b) &= 2nP_{\hat{\mathcal{A}}}(b^n) \cdot b^{n-1} - 2nF_{\hat{\mathcal{A}}}(b)b \\
&= 2n\langle S, b^n\rangle S \cdot b^{n-1} - 2n\delta^2 n \\
&= 2n(-\delta)(-\delta b) - 2n\delta^2 b \\
&= 0,
\end{aligned}
$$

which shows that $b$ is a stationary point of $F_{\hat{\mathcal{A}}}$. Now taking any unit norm $z \perp b$ and using (S.21),

$$
\begin{aligned}
z^\top \nabla_{\mathbb{S}^{D-1}}^2 F_{\hat{\mathcal{A}}}(b)z &= 2n^2\left\|P_{\hat{\mathcal{A}}}(b^{n-1}z)\right\|_F^2 + 2n(n-1)\langle P_{\hat{\mathcal{A}}}(b^n), b^{n-2}z^2\rangle - 2nF_{\hat{\mathcal{A}}}(b) \\
&= 2n^2\left\|\langle S, b^{n-1}z\rangle S\right\|_F^2 + 2n(n-1)\langle\langle S, b^n\rangle S, b^{n-2}z^2\rangle - 2n\delta^2 \\
&= -2n(n-1)\delta\sqrt{1-\delta^2}\langle a, b\rangle^{n-2}\langle a, z\rangle^2 - 2n\delta^2 \\
&\leq -2n\delta^2 \\
&< 0,
\end{aligned}
$$

where we used the fact $n \geq 2$ in third equality. Thus, the Riemannian Hessian of $F_{\hat{\mathcal{A}}}$ is strictly negative-definite at $b$. Therefore, $b$ is a strict local maximizer of $F_{\hat{\mathcal{A}}}$ as we wanted. $\qquad\square$

# D   Derivation of Riemannian derivatives and optimality conditions for nSPM-P

In this section, we derive the Riemannian derivatives and optimality conditions for **nSPM-P**.

## D.1   Riemannian derivatives of $F_{\hat{\mathcal{A}}}$ on the sphere

**Lemma S.9.** *Let $x, z \in \mathbb{S}^{D-1}$ with $z \perp x$. The Riemannian gradient and Hessian of $F_{\hat{\mathcal{A}}}$ satisfy*

$$\nabla_{\mathbb{S}^{D-1}}F_{\hat{\mathcal{A}}}(x) = 2nP_{\hat{\mathcal{A}}}(x^n) \cdot x^{n-1} - 2nF_{\hat{\mathcal{A}}}(x)x, \tag{S.20}$$

$$z^\top \nabla_{\mathbb{S}^{D-1}}^2 F_{\hat{\mathcal{A}}}(x)z = 2n^2\|P_{\hat{\mathcal{A}}}(x^{n-1}z)\|_F^2 + 2n(n-1)\langle P_{\hat{\mathcal{A}}}(x^n), x^{n-2}z^2\rangle - 2nF_{\hat{\mathcal{A}}}(x). \tag{S.21}$$

**Remark S.10.** In this lemma $z$ is a tangent vector to the unit sphere at $x$, since $z \perp x$. For convenience we work with normalized tangent vectors, which is why there is the further assumption that $\|z\|_2 = 1$. This is without loss of generality: second-order criticality for **nSPM-P** is the condition that the quadratic form on the tangent space given by $\nabla^2_{\mathbb{S}^{D-1}} F_{\hat{\mathcal{A}}}(x)$ is negative semi-definite. By homogeneity of the quadratic form, this is verified by just checking normalized tangent vectors.

*Proof.* We first calculate the Euclidean gradient and Hessian of $F_{\hat{\mathcal{A}}}$. Let $U_1, \ldots, U_K \in \mathrm{Sym}(\mathcal{T}_D^n)$ form an orthonormal basis of $\hat{\mathcal{A}}$. Then for all $S, T \in \mathcal{T}_D^n$,

$$P_{\hat{\mathcal{A}}}(T) = \sum_{i=1}^{K} \langle U_i, T \rangle U_i, \quad \langle P_{\hat{\mathcal{A}}}(T), S \rangle = \sum_{i=1}^{K} \langle U_i, T \rangle \langle U_i, S \rangle, \quad \text{and} \quad \|P_{\hat{\mathcal{A}}}(T)\|_F^2 = \sum_{i=1}^{K} \langle U_i, T \rangle^2.$$

Also, direct calculations verify that if $T \in \mathrm{Sym}(\mathcal{T}_D^n)$, then $\nabla \langle T, x^n \rangle = nT \cdot x^{n-1}$ and $\nabla^2 \langle T, x^n \rangle = n(n-1)T \cdot x^{n-2}$ (see [9, Lem. 3.1, Lem. 3.3]). Therefore,

$$\nabla F_{\hat{\mathcal{A}}}(x) = \nabla \|P_{\hat{\mathcal{A}}}(x^n)\|_F^2 = \sum_{i=1}^{K} \nabla \langle U_i, x^n \rangle^2$$

$$= 2 \sum_{i=1}^{K} \langle U_i, x^n \rangle \nabla \langle U_i, x^n \rangle = 2n \sum_{i=1}^{K} \langle U_i, x^n \rangle U_i \cdot x^{n-1} = 2n P_{\hat{\mathcal{A}}}(x^n) \cdot x^{n-1},$$

and

$$\nabla^2 F_{\hat{\mathcal{A}}}(x) = \sum_{i=1}^{K} \nabla^2 \langle U_i, x^n \rangle^2 = 2 \sum_{i=1}^{K} (\nabla \langle U_i, x^n \rangle)(\nabla \langle U_i, x^n \rangle)^\top + 2 \sum_{i=1}^{K} \langle U_i, x^n \rangle \nabla^2 \langle U_i, x^n \rangle$$

$$= 2n^2 \sum_{i=1}^{K} (U_i \cdot x^{n-1})(U_i \cdot x^{n-1})^\top + 2n(n-1) \sum_{i=1}^{K} \langle U_i, x^n \rangle U_i \cdot x^{n-2}.$$

Thus,

$$z^\top \nabla^2 F_{\hat{\mathcal{A}}}(x) z = 2n^2 \sum_{i=1}^{K} \langle U_i, x^{n-1} z \rangle^2 + 2n(n-1) \sum_{i=1}^{K} \langle U_i, x^n \rangle \langle U_i, x^{n-2} z^2 \rangle$$

$$= 2n^2 \|P_{\hat{\mathcal{A}}}(x^{n-1} z)\|_F^2 + 2n(n-1) \langle P_{\hat{\mathcal{A}}}(x^n), x^{n-2} z^2 \rangle.$$

We now calculate the Riemannian gradient and Hessian of $F_{\hat{\mathcal{A}}}$. In general, for a twice-differentiable function $g : \mathbb{R}^n \to \mathbb{R}$, the Riemannian gradient and Hessian of the restriction of $g$ to the unit sphere $\mathbb{S}^{D-1}$ are related to the Euclidean counterparts of $g$ as follows (see [1, Ex. 3.6.1] and [2, Eq. (10), Sec. 4.2]):

$$\nabla_{\mathbb{S}^{D-1}} g(x) = (I - xx^\top) \nabla g(x),$$

$$\text{and} \quad \nabla^2_{\mathbb{S}^{D-1}} g(x) = (I - xx^\top)(\nabla^2 g(x) - (x^\top \nabla g(x)) \mathsf{Id}_D)(I - xx^\top).$$

Applying this to $F_{\hat{\mathcal{A}}}$, and using $x^\top \nabla F_{\hat{\mathcal{A}}}(x) = 2n P_{\hat{\mathcal{A}}}(x^n) \cdot x^n = 2n F_{\hat{\mathcal{A}}}(x)$, $(I - xx^\top)z = z$ (since $z \perp x$) and $z^\top z = 1$ (since $z \in \mathbb{S}^{D-1}$), yields the result. $\qquad\square$

## D.2 Proof of Proposition M.3

**Proposition S.11** (= Proposition M.3). *Let $x \in \mathbb{S}^{D-1}$ be first and second-order critical for M.**nSPM-P**. Then for each $z \in \mathbb{S}^{D-1}$ with $z \perp x$, we have*

$$P_{\hat{\mathcal{A}}}(x^n) \cdot x^{n-1} = F_{\hat{\mathcal{A}}}(x) x, \tag{S.22}$$

$$F_{\hat{\mathcal{A}}}(x) \geq n \|P_{\hat{\mathcal{A}}}(x^{n-1} z)\|_F^2 + (n-1) \langle P_{\hat{\mathcal{A}}}(x^n), x^{n-2} z^2 \rangle. \tag{S.23}$$

*Furthermore, for any $y \in \mathbb{S}^{D-1}$ we have*

$$F_{\hat{\mathcal{A}}}(x) \geq n \left\| P_{\hat{\mathcal{A}}}(x^{n-1} y) \right\|_F^2 + (n-1) \langle P_{\hat{\mathcal{A}}}(x^n), x^{n-2} y^2 \rangle - 2(n-1) F_{\hat{\mathcal{A}}}(x) \langle x, y \rangle^2. \tag{S.24}$$

*Proof.* The first-order (stationary point) condition is given by setting the Riemannian gradient (S.20) to zero. The second-order condition is the requirement that the Riemannian Hessian be negative semi-definite, i.e., that (S.21) is non-positive for all unit norm $z \perp x$. Dividing by $2n$ and rearranging yields (S.22) and (S.23) respectively.

It remains to show (S.24). Assume $x \in \mathbb{S}^{D-1}$ is first and second-order critical for **nSPM-P**. For each $y \in \mathbb{S}^{D-1}$, let $\alpha, \beta \in [0,1]$, $z \in \mathbb{S}^{D-1}$ be such that $\beta = \sqrt{1-\alpha^2}$, $z \perp x$ and $y = \alpha x + \beta z$. Then,

$$
\begin{aligned}
\left\| P_{\hat{\mathcal{A}}}(x^{n-1}y) \right\|_F^2 &= \left\| \alpha P_{\hat{\mathcal{A}}}(x^n) + \beta P_{\hat{\mathcal{A}}}(x^{n-1}z) \right\|_F^2 \\
&= \alpha^2 \left\| P_{\hat{\mathcal{A}}}(x^n) \right\|_F^2 + \beta^2 \left\| P_{\hat{\mathcal{A}}}(x^{n-1}z) \right\|_F^2 + 2\alpha\beta \langle P_{\hat{\mathcal{A}}}(x^n), P_{\hat{\mathcal{A}}}(x^{n-1}z) \rangle \\
&= \alpha^2 F_{\hat{\mathcal{A}}}(x) + \beta^2 \left\| P_{\hat{\mathcal{A}}}(x^{n-1}z) \right\|_F^2 + 2\alpha\beta \langle P_{\hat{\mathcal{A}}}(x^n), x^{n-1}z \rangle \\
&= \alpha^2 F_{\hat{\mathcal{A}}}(x) + \beta^2 \left\| P_{\hat{\mathcal{A}}}(x^{n-1}z) \right\|_F^2 + 2\alpha\beta \langle P_{\hat{\mathcal{A}}}(x^n) \cdot x^{n-1}, z \rangle \\
&= \alpha^2 F_{\hat{\mathcal{A}}}(x) + \beta^2 \left\| P_{\hat{\mathcal{A}}}(x^{n-1}z) \right\|_F^2 + 2\alpha\beta \langle F_{\hat{\mathcal{A}}}(x)x, z \rangle &\text{(S.25)} \\
&= \alpha^2 F_{\hat{\mathcal{A}}}(x) + \beta^2 \left\| P_{\hat{\mathcal{A}}}(x^{n-1}z) \right\|_F^2, &\text{(S.26)}
\end{aligned}
$$

where we used the stationary point condition (S.22) in (S.25) and $x \perp z$ in (S.26). By similar logic,

$$
\begin{aligned}
\langle P_{\hat{\mathcal{A}}}(x^n), x^{n-2}y^2 \rangle &= \langle P_{\hat{\mathcal{A}}}(x^n), x^{n-2}(\alpha x + \beta z)^2 \rangle \\
&= \alpha^2 \left\| P_{\hat{\mathcal{A}}}(x^n) \right\|_F^2 + 2\alpha\beta \langle P_{\hat{\mathcal{A}}}(x^n), x^{n-1}z \rangle + \beta^2 \langle P_{\hat{\mathcal{A}}}(x^n), x^{n-2}z^2 \rangle \\
&= \alpha^2 F_{\hat{\mathcal{A}}}(x) + \beta^2 \langle P_{\hat{\mathcal{A}}}(x^n), x^{n-2}z^2 \rangle. &\text{(S.27)}
\end{aligned}
$$

Finally, we substitute (S.26) and (S.27) into (S.24), and use the second-order condition (S.23):

$$
\begin{aligned}
n \left\| P_{\hat{\mathcal{A}}}(x^{n-1}y) \right\|_F^2 &+ (n-1)\langle P_{\hat{\mathcal{A}}}(x^n), x^{n-2}y^2 \rangle \\
&= (2n-1)\alpha^2 F_{\hat{\mathcal{A}}}(x) + \beta^2 \left( n \left\| P_{\hat{\mathcal{A}}}(x^{n-1}z) \right\|_F^2 + (n-1)\langle P_{\hat{\mathcal{A}}}(x^n), x^{n-2}z^2 \rangle \right) \\
&\leq (2n-1)\alpha^2 F_{\hat{\mathcal{A}}}(x) + \beta^2 F_{\hat{\mathcal{A}}}(x) \\
&= (2n-1)\alpha^2 F_{\hat{\mathcal{A}}}(x) + (1-\alpha^2)F_{\hat{\mathcal{A}}}(x) \\
&= (1 + (2n-2)\alpha^2)F_{\hat{\mathcal{A}}}(x). &\text{(S.28)}
\end{aligned}
$$

Substituting $\alpha = \langle x, y \rangle$ into (S.28) gives (S.24) as desired. $\square$

# E   Preparatory results on random ensembles over the sphere and Conjecture M.15

In this section, we prove the results on random ensembles over the sphere that are used in the analysis of the overcomplete random tensor model. Namely, we prove Lemma M.11 and Proposition M.13.

## E.1   Proof of Lemma M.11

**Lemma S.12** (= Lemma M.11). *Suppose that $A = [a_1 | \ldots | a_K] \in \mathbb{R}^{D \times K}$ satisfies the $(p, \delta)$-RIP for $p = \lceil c_\delta D / \log(K) \rceil$. Let $\tilde{c}_\delta := (1+\delta)/c_\delta$. Then for each $x \in \mathbb{S}^{D-1}$, there exists a subset of indices $\mathcal{I}(x) \subseteq [K]$ with cardinality $p$ such that*

$$
1 - \delta \leq \sum_{i \in \mathcal{I}(x)} \langle a_i, x \rangle^2 \leq 1 + \delta \quad \text{and} \quad \langle a_i, x \rangle^2 \leq \tilde{c}_\delta \frac{\log(K)}{D} \text{ for } i \notin \mathcal{I}(x).
$$

*Proof.* Fix $x \in \mathbb{S}^{D-1}$, and assume without loss of generality that the indices are ordered according to $\langle a_1, x \rangle^2 \geq \ldots \geq \langle a_K, x \rangle^2$. We claim that the index set $\mathcal{I}(x) = [p]$ satisfies the specified conditions. Denote by $A_p \in \mathbb{R}^{D \times p}$ the submatrix of $A$ consisting of the first $p$ columns. The $(p, \delta)$-RIP implies

$$
1 + \delta = (1+\delta)\|x\|_2^2 \geq x^\top A_p A_p^\top x = \sum_{i=1}^p \langle a_i, x \rangle^2 \geq (1-\delta)\|x\|_2^2 = 1 - \delta. \tag{S.29}
$$

Hence, $\mathcal{I}(x)$ satisfies the first condition. As for the second condition, by the ordering assumption,

$$\sum_{i=1}^{p} \langle a_i, x \rangle^2 \geq p \langle a_p, x \rangle^2. \tag{S.30}$$

Inequalities (S.29) and (S.30) imply $\langle a_p, x \rangle^2 \leq (1+\delta)p^{-1}$. Then again by the ordering relation,

$$\max_{i \notin \mathcal{I}(x)} \langle a_i, x \rangle^2 \leq \langle a_p, x \rangle^2 \leq \frac{1+\delta}{p} \leq \frac{1+\delta}{c_\delta} \frac{\log(K)}{D} = \tilde{c}_\delta \frac{\log(K)}{D}. \qquad \square$$

### E.2  Proof of Proposition M.13

Now consider a random ensemble of independent copies $a_1, \ldots, a_K$ of the random vector $a \sim \mathrm{Unif}(\mathbb{S}^{D-1})$. Before showing Proposition M.13, which states that for this ensemble the assumptions **A1**-**A2** hold with high probability for all $n$, and **A3** holds with high probability if $n = 2$, we will recall these assumptions. Write $A := [a_1| \ldots |a_K] \in \mathbb{R}^{D \times K}$.

**A1** There exists $c_\delta > 0$, depending only on $\delta$, such that $A$ is $(\lceil c_\delta D / \log(K) \rceil, \delta)$-RIP.

**A2** There exists $c_1 > 0$, independent of $K, D$, such that $\max_{i \neq j} \langle a_i, a_j \rangle^2 \leq c_1 \log(K)/D$.

**A3** There exists $c_2 > 0$, independent of $K, D$, such that $\left\| G_n^{-1} \right\|_2 \leq c_2$.

The proof for **A3** is lengthy and more technically involved (even though we only prove it for the case of $n = 2$). So, we first give the arguments for **A1**-**A2**.

**Lemma S.13** (**A1**-**A2** in Proposition M.13)**.** *Let $a_1, \ldots, a_K$ be independent copies of the random vector $a \sim \mathrm{Unif}(\mathbb{S}^{D-1})$, and define $A := [a_1| \ldots |a_K] \in \mathbb{R}^{D \times K}$. Assume $\log K = o(D)$.*

*1. Fix $\delta \in (0, 1)$. There exists a universal constant $C > 0$ and constants $D_0 \in \mathbb{N}$, $c_\delta > 0$ depending only on $\delta$ such that, if $D \geq D_0$ and we put $p := \lceil c_\delta D / \log(K) \rceil$, then $A$ is $(p, \delta)$-RIP with probability at least $1 - 2\exp(-C\delta^2 D)$.*

*2. There exists a universal constant $c_1 > 0$ such that*

$$\mathbb{P}\left( \max_{i,j: i \neq j} \langle a_i, a_j \rangle^2 \leq c_1 \frac{\log(K)}{D} \right) \geq 1 - \frac{1}{K}.$$

*Proof.* 1. By [11, Thm. 5.65], $A$ satisfies the $(p, \delta)$-RIP with probability at least $1 - 2\exp(-C\delta^2 D)$, provided that $D \geq C'\delta^{-2} p \log(eK/p)$, where $C, C' > 0$ are universal constants. Thus it suffices to check the latter inequality holds if $c_\delta, D_0$ are appropriately chosen. Because $\log K = o(D)$, for each fixed $c_\delta > 0$, there exists $D_0$ such that $p > e$ whenever $D \geq D_0$. In this case, we have

$$C'\delta^{-2}p\log(eK/p) \; < \; C'\delta^{-2}p\log(K) \; = \; C'\delta^{-2}\lceil c_\delta D/\log(K) \rceil \log(K)$$
$$\leq \; C'\delta^{-2}c_\delta D + C'\delta^{-2}\log(K) \; < \; C'\delta^{-2}\tfrac{3}{2}c_\delta D, \tag{S.31}$$

where the last inequality in (S.31) is because $p > e$ implies $\log(K) < \tfrac{1}{2}c_\delta D$. We note that the right-most quantity in (S.31) is bounded above by $D$, as desired, if we choose $c_\delta \leq 2\delta^2/(3C')$.

2. Let $d_{ij} := |\langle a_i, a_j \rangle|$. By rotational symmetry, $d_{ij} \overset{d}{=} |a_{11}|$ for $i \neq j$ in distribution, where $a_{11} := \langle a_1, e_1 \rangle$ denotes the first coordinate of $a_1$. By [12, Thm. 3.4.6], $a_{11}$ is a sub-Gaussian random variable with sub-Gaussian norm $\|a_{11}\|_{\psi_2} \leq c_1'D^{-\frac{1}{2}}$ for a universal constant $c_1' > 0$. Using this, a union bound and a tail bound for sub-Gaussian random variables [12, Eq. 2.14],

$$\mathbb{P}\left( \max_{i \neq j} d_{ij} \leq t \right) \; = \; 1 - \mathbb{P}(\text{there exists } i \neq j \text{ s.t. } d_{ij} > t) \geq 1 - \sum_{i=1}^{K} \sum_{j=i+1}^{K} \mathbb{P}(d_{ij} > t)$$

$$= \; 1 - \frac{K(K-1)}{2}\mathbb{P}(|a_{11}| > t) \; > \; 1 - K^2 \exp\left( -\frac{t^2 D}{c_1''} \right)$$

for a universal constant $c_1'' > 0$. We finish by setting $t = \sqrt{3c_1'' \log(K)/D}$ and $c_1 = 3c_1''$. $\qquad \square$

Next we come to **A3** for a random ensemble on the sphere when $n = 2$.

**Proposition S.14** (**A3** in Proposition M.13). *Let $K, D \in \mathbb{N}$ satisfy $1 \leq K \leq D^2$. Let $a_1, \ldots, a_K$ be independent copies of the random vector $a \sim \mathrm{Unif}(\mathbb{S}^{D-1})$. Define the Grammian $G_2 \in \mathrm{Sym}(\mathcal{T}_K^2)$ by $(G_2)_{ij} := \langle a_i^{\bullet 2}, a_j^{\bullet 2} \rangle = \langle a_i, a_j \rangle^2$. Then there exists a universal constant $C > 0$ such that the minimal eigenvalue of $G_2$ satisfies*

$$\mu_K(G_2) \geq 1 - (C+1)\frac{\sqrt{K}}{D}\log\left(\frac{eD}{\sqrt{K}}\right), \quad \text{with probability at least } 1 - C\left(\frac{eD}{\sqrt{K}}\right)^{-C\sqrt{K}} \quad \text{(S.32)}$$

*In particular, if $K = o(D^2)$, then for each constant $c_2 > 1$, we have $\|G_2^{-1}\|_2 \leq c_2$ with probability tending to $1$ as $D \to \infty$.*

The statement is implied by $(p, \delta)$-RIP with $p = K$ for matrices like $[a_1^{\bullet 2}|\ldots|a_K^{\bullet 2}]$ (columns are Khatri-Rao squares of vectors). Such RIP statements have been analyzed in [7]. The main technical ingredients are [3, Thm. 3.3], which proves RIP for matrices containing columnwise sub-exponential random vectors, and the fact that certain vectorized Khatri-Rao products of sub-Gaussian random vectors are sub-exponential with favorable sub-exponential norm. For completeness, we include a self-contained proof of Proposition S.14. First, here is a version of [3, Thm. 3.3] tailored to our needs.

**Proposition S.15** (Tailored version of [3, Thm. 3.3]). *Let $K, D \in \mathbb{N}$ satisfy $1 \leq K \leq D^2$. Let $X_1, \ldots, X_K \in \mathbb{R}^{D^2}$ be independent copies of a sub-exponential random vector $X \in \mathbb{R}^{D^2}$ with $\mathcal{O}(1)$ sub-exponential norm (i.e., upper bounded independently of $D$ and $K$). Furthermore, assume that $\|X\|_2 = D$ almost surely. Define $M := \frac{1}{D}[X_1|\ldots|X_K] \in \mathbb{R}^{D^2 \times K}$. Then there exists a universal constant $C > 0$ such that*

$$\|M^\top M - \mathsf{Id}_K\|_2 \leq C\frac{\sqrt{K}}{D}\log\left(\frac{eD}{\sqrt{K}}\right), \quad \text{with probability at least } 1 - C\left(\frac{eD}{\sqrt{K}}\right)^{-C\sqrt{K}} \quad \text{(S.33)}$$

*In particular, in this event the minimal eigenvalue of the associated Grammian satisfies*

$$\mu_K(M^\top M) \geq 1 - C\frac{\sqrt{K}}{D}\log\left(\frac{eD}{\sqrt{K}}\right). \quad \text{(S.34)}$$

*Proof.* The first part of the statement is [3, Thm. 3.3], in the special case (relating their notation to our notation) $n = D^2$, $m = K$, $N = K$, $r = 1$, $K = 1$, $K' = 1 + \epsilon$ for any $\epsilon \in (0, 1)$ and $\theta' \to 0$, where we note that $M$ satisfies $(K, \delta)$-RIP if and only if $\|M^\top M - \mathsf{Id}_K\|_2 \leq \delta$, since $M$ has $K$ columns. The second part follows immediately from the definition of the spectral norm. $\square$

To deduce Proposition S.14, we apply Proposition S.15 to a centered and scaled version of the Khatri-Rao squares $a_i^{\bullet 2} = \mathrm{Vec}(a_i^2)$, following [7].

*Proof of Proposition S.14.* Consider the Khatri-Rao square $A^{\bullet 2} := [a_1^{\bullet 2}|\ldots|a_K^{\bullet 2}] \in \mathbb{R}^{D^2 \times K}$ and note that $G_2 = (A^{\bullet 2})^\top A^{\bullet 2}$. Instead of working with $a_1^{\bullet 2}, \ldots, a_K^{\bullet 2}$, we introduce auxiliary variables

$$X_i := \sqrt{\frac{D}{D-1}}\,\mathrm{Vec}(Da_i^2 - \mathsf{Id}_D) \in \mathbb{R}^{D^2},$$

in order to be able to apply Proposition S.15. Then $X_1, \ldots, X_K$ are independent copies of the random vector $X \in \mathbb{R}^{D^2}$ given by

$$X := \sqrt{\frac{D}{D-1}}\,\mathrm{Vec}(Da^2 - \mathsf{Id}_D), \quad \text{where } a \sim \mathrm{Unif}(\mathbb{S}^{D-1}). \quad \text{(S.35)}$$

Note that $\mathbb{E}[X_i] = \sqrt{\frac{D}{D-1}}\left(\mathrm{Vec}(D\mathbb{E}[a_i^2] - \mathsf{Id}_D)\right) = 0$. Using $a_i \in \mathbb{S}^{D-1}$, we also have

$$\|X_i\|_2^2 = \frac{D}{D-1}\|Da_ia_i^\top - \mathsf{Id}_D\|_F^2 = \frac{D}{D-1}\left(\|Da_ia_i^\top\|_F^2 - 2\langle Da_ia_i^\top, \mathsf{Id}_D\rangle + \|\mathsf{Id}_D\|_F^2\right)$$

$$= \frac{D}{D-1}\left(D^2\|a_i\|_2^4 - 2D\|a_i\|_2^2 + D\right) = \frac{D}{D-1}\left(D^2 - 2D + D\right) = D^2,$$

so that $\|X_i\|_2 = D$ as required in Proposition S.15.

Define $M := \frac{1}{D}[X_1| \dots |X_K] \in \mathbb{R}^{D^2 \times K}$. For each $i, j \in [K]$, again using $a_i, a_j \in \mathbb{S}^{D-1}$, we have

$$
\begin{aligned}
\left(M^\top M\right)_{i,j} &= \frac{1}{D(D-1)} \langle D a_i a_i^\top - \mathsf{Id}_D, D a_j a_j^\top - \mathsf{Id}_D \rangle_F \\
&= \frac{1}{D(D-1)} \left( D^2 \langle a_i a_i^\top, a_j a_j^\top \rangle_F - D \langle a_i a_i^\top + a_j a_j^\top, \mathsf{Id}_D \rangle_F + \langle \mathsf{Id}_D, \mathsf{Id}_D \rangle_F \right) \\
&= \frac{1}{D(D-1)} \left( D^2 \langle a_i, a_j \rangle^2 - D\|a_i\|_2^2 - D\|a_j\|_2^2 + D \right) \\
&= \frac{D}{D-1} \langle a_i, a_j \rangle^2 - \frac{1}{D-1}.
\end{aligned}
$$

Hence, $M^\top M = \frac{D}{D-1} G_2 - \frac{1}{D-1} \mathbb{1}_K \mathbb{1}_K^\top$, which may be rewritten $G_2 = \frac{D-1}{D} M^\top M + \frac{1}{D} \mathbb{1}_K \mathbb{1}_K^\top$. This implies

$$
\mu_K(G_2) \geq \frac{D-1}{D} \mu_K(M^\top M). \tag{S.36}
$$

Thus, we focus on the Grammian associated with the shifted and centered random variables $X_1, \dots, X_K$. To apply Proposition S.15 to $M^\top M$, it remains to show that the sub-exponential norm of $X$ in (S.35) is bounded by some universal constant, independent of $K, D$.

We briefly indicate how this is done. We first note that all random vectors $\sqrt{D} a_1, \dots, \sqrt{D} a_K$ have the so-called convex concentration property for some universal constant $C' > 0$, see [7, Thm. 6, Def. 7, Thm. 8]. Following [7] and the references therein, this can be used to prove the Hanson-Wright type inequality

$$
\mathbb{P}\left( \left| D(a_i^\top Y a_i - \mathbb{E}[a_i^\top Y a_i]) \right| > t \right) \leq 2 \exp\left( -C' \min\left( \frac{t^2}{\|Y\|_F^2}, \frac{t}{\|Y\|_2} \right) \right), \tag{S.37}
$$

for all (deterministic) $Y \in \mathbb{R}^{D \times D}$ and $t > 0$. Taking now an arbitrary unit-norm vector $y \in \mathbb{R}^{D^2}$, denoting $Y \in \mathbb{R}^{D \times D}$ as the matrix satisfying $y = \text{vec}(Y)$, and using $\mathbb{E}[aa^\top] = \frac{1}{D}\mathsf{Id}_D$, we have

$$
\begin{aligned}
\langle X, y \rangle &= \sqrt{\frac{D}{D-1}} \langle D a a^\top - \mathsf{Id}_D, Y \rangle \\
&= \sqrt{\frac{D}{D-1}} \langle D a a^\top - D\mathbb{E}[aa^\top], Y \rangle \\
&= \sqrt{\frac{D}{D-1}} D(a^\top Y a - \mathbb{E}[a^\top Y a]). \tag{S.38}
\end{aligned}
$$

Combining (S.38), (S.37) and $\|Y\|_2 \leq \|Y\|_F = 1$, we obtain

$$
\mathbb{P}\left( \langle X, y \rangle > \sqrt{\frac{D}{D-1}} t \right) \leq 2 \exp(-C' \min(t^2, t)).
$$

This implies $\langle X, y \rangle$ is sub-exponential with sub-exponential norm $C'\sqrt{D/(D-1)} = \mathcal{O}(1)$. Taking a supremum over all unit-norm vectors $y \in \mathbb{R}^{D^2}$ does not change this bound, since it is independent of $y$, and so the random vector $X$ has sub-exponential norm $\mathcal{O}(1)$. Thus, Proposition S.15 applies and implies there exists a universal constant $C > 0$ such that (S.34) holds with the probability in (S.33).

We now notice that (S.32) follows by substituting (S.34) into (S.36) and using that $1 \leq K \leq D^2$ implies $\frac{1}{D} \leq \frac{\sqrt{K}}{D} \log(eD/\sqrt{K})$, because then

$$
\mu_K(G_2) \geq (1 - \frac{1}{D}) \left( 1 - C \frac{\sqrt{K}}{D} \log\left( \frac{eD}{\sqrt{K}} \right) \right) \geq 1 - (C+1) \frac{\sqrt{K}}{D} \log\left( \frac{eD}{\sqrt{K}} \right). \tag{S.39}
$$

To conclude, we justify the last sentence in Proposition S.14 where $K = o(D^2)$. It only remains to note that, in this case, the right-most quantity in (S.39) tends to 1 as $D \to \infty$. However, this holds because $\frac{\sqrt{K}}{D} \to 0^+$ by $K = o(D^2)$, and $\lim_{x \to 0^+} x \log(\frac{1}{x}) = 0$ by L'Hôpital's rule. $\qquad \square$

As mentioned in the main body of the paper, this proof technique cannot be immediately applied to the case $n > 2$ because some key technical results are missing. We are not aware of extensions of Proposition S.15 to random variables with tails heavier than subexponential random variables. Further we do not know how to show that an auxiliary variable $\tilde{z}$ formed from higher-order vectorized tensors $a_i^n$ satisfies a suitable tail bound, which would be needed in an extended Proposition S.15.

## F  Technical tools for proving Theorems M.7 and M.16

In this section we prove the key technical tools to be used in the proofs of our landscape theorem.

**Lemma S.16** (Subspace perturbation effect). *Let $\mathcal{A}$, $\hat{\mathcal{A}}$ be any two subspaces of $\mathrm{Sym}(\mathcal{T}_D^n)$, and define*
$$\Delta_{\mathcal{A}} := \|P_{\mathcal{A}} - P_{\hat{\mathcal{A}}}\|_{F \to F} = \sup_{T \in \mathrm{Sym}(\mathcal{T}_D^n),\, \|T\|_F = 1} \|P_{\mathcal{A}}(T) - P_{\hat{\mathcal{A}}}(T)\|_F.$$
*Then for all $S, T$ in $\mathcal{T}_D^n$, we have*
$$|\langle P_{\mathcal{A}}(T) - P_{\hat{\mathcal{A}}}(T), S \rangle| \le \Delta_{\mathcal{A}} \|\mathrm{Sym}(T)\|_F \|\mathrm{Sym}(S)\|_F \le \Delta_{\mathcal{A}} \|T\|_F \|S\|_F, \tag{S.40}$$
*and*
$$\left| \|P_{\mathcal{A}}(T)\|_F^2 - \|P_{\hat{\mathcal{A}}}(T)\|_F^2 \right| \le \Delta_{\mathcal{A}} \|\mathrm{Sym}(T)\|_F^2 \le \Delta_{\mathcal{A}} \|T\|_F^2. \tag{S.41}$$

*Proof.* Using the fact $\mathcal{A}, \hat{\mathcal{A}} \subseteq \mathrm{Sym}(\mathcal{T}_D^n)$, the Cauchy-Schwarz inequality, and the definition of $\Delta_{\mathcal{A}}$,
$$\begin{aligned}
\left| \langle P_{\mathcal{A}}(T) - P_{\hat{\mathcal{A}}}(T), S \rangle \right| &= \left| \langle P_{\mathcal{A}}(\mathrm{Sym}(T)) - P_{\hat{\mathcal{A}}}(\mathrm{Sym}(T)), \mathrm{Sym}(S) \rangle \right| \\
&\le \left\| P_{\mathcal{A}}(\mathrm{Sym}(T)) - P_{\hat{\mathcal{A}}}(\mathrm{Sym}(T)) \right\|_F \|\mathrm{Sym}(S)\|_F \\
&\le \Delta_{\mathcal{A}} \|\mathrm{Sym}(T)\|_F \|\mathrm{Sym}(S)\|_F,
\end{aligned}$$
which gives (S.40). Equation (S.41) then follows by setting $S = T$ in (S.40), and using $\langle P_{\mathcal{A}}(T) - P_{\hat{\mathcal{A}}}(T), T \rangle = \|P_{\mathcal{A}}(T)\|_F^2 - \|P_{\hat{\mathcal{A}}}(T)\|_F^2$ which holds since $P_{\mathcal{A}}$ and $P_{\hat{\mathcal{A}}}$ are orthogonal projectors.  □

Next we recall the remainder terms for $P_{\mathcal{A}}(x^n)$ with respect to $x^{n-s} a_i^s$ (S.7). These are defined by
$$\langle P_{\mathcal{A}}(x^n), x^{n-s} a_i^s \rangle =: \sigma_i \zeta_i^{n-s} + R_{i,s}.$$

**Lemma S.17.** *The following alternative expressions for the remainder terms hold:*

1. *For any $x \in \mathbb{S}^{D-1}$, $i \in [K]$ and $0 \le s \le n$, we have*
$$R_{i,s} = \langle P_{\mathcal{A}}(x^n), x^{n-s} a_i^s \rangle - \sigma_i \zeta_i^{n-s}. \tag{S.42}$$

2. *For any $x \in \mathbb{S}^{D-1}$ and $i \in [K]$, we have*
$$R_{i,n} = \zeta_i^n - \sigma_i. \tag{S.43}$$

In the first step of the proofs of our two main results, we derive lower bounds for $\|A^\top x\|_\infty$, where $x$ is a second-order critical point. For both theorems, this step is based on the following inequality.

**Proposition S.18** (Tool #1). *For any second-order critical point $x$ of **nSPM-P** and $i \in [K]$, it holds*
$$(1 + 2(n-1)\zeta_i^2) F_{\mathcal{A}}(x) \ge (2n-1)\sigma_i \zeta_i^{n-2} + n\zeta_i^{n-2} R_{i,n} + (n-1) R_{i,2} - (4n-2)\Delta_{\mathcal{A}}. \tag{S.44}$$

*Proof.* We substitute $y = a_i$ into the inequality (S.24), which is a consequence of $x$ being first and second-order critical for **nSPM-P**:
$$(1 + 2(n-1)\zeta_i^2) F_{\hat{\mathcal{A}}}(x) \ge n \left\| P_{\hat{\mathcal{A}}}(x^{n-1} a_i) \right\|_F^2 + (n-1)\langle P_{\hat{\mathcal{A}}}(x^n), x^{n-2} a_i^2 \rangle. \tag{S.45}$$
Then by the subspace perturbation results (S.40) and (S.41), we have
$$F_{\hat{\mathcal{A}}}(x) = \|P_{\hat{\mathcal{A}}}(x^n)\|_F^2 \le \|P_{\mathcal{A}}(x^n)\|_F^2 + \Delta_{\mathcal{A}} = F_{\mathcal{A}}(x) + \Delta_{\mathcal{A}}, \tag{S.46}$$

$$\begin{aligned}
\|P_{\hat{\mathcal{A}}}(x^{n-1} a_i)\|_F^2 &\ge \|P_{\mathcal{A}}(x^{n-1} a_i)\|_F^2 - \Delta_{\mathcal{A}} \ge \|P_{\mathrm{Span}\{a_i^n\}}(x^{n-1} a_i)\|_F^2 - \Delta_{\mathcal{A}} \\
&= \zeta_i^{2n-2} - \Delta_{\mathcal{A}} = \zeta_i^{n-2}(R_{i,n} + \sigma_i) - \Delta_{\mathcal{A}},
\end{aligned} \tag{S.47}$$

$$\langle P_{\hat{\mathcal{A}}}(x^n), x^{n-2} a_i^2 \rangle \ge \langle P_{\mathcal{A}}(x^n), x^{n-2} a_i^2 \rangle - \Delta_{\mathcal{A}} = R_{i,2} + \sigma_i \zeta_i^{n-2} - \Delta_{\mathcal{A}}. \tag{S.48}$$
Here we used $\mathrm{Span}\{a_i^n\} \subseteq \mathcal{A}$ in the second inequality of (S.47), the identity (S.43) in the second equality of (S.47), and the identity (S.42) in the equality of (S.48). Substituting (S.46), (S.47) and (S.48) into (S.45) completes the proof.  □

In the second part of both proofs, we show concavity holds in a spherical cap near each $sa_i$, $i \in [K]$, $s \in \{-1, 1\}$. As a starting point in both arguments, we use the next statement to upper bound the eigenvalues of the Riemannian Hessian.

**Proposition S.19** (Tool #2). *For any $x, z \in \mathbb{S}^{D-1}$ with $z \perp x$, the Riemannian Hessian of $F_{\hat{\mathcal{A}}}$ satisfies, for any $i \in [K]$,*

$$\frac{1}{2n} z^\top \nabla^2_{\mathbb{S}^{D-1}} F_{\hat{\mathcal{A}}}(x) z \leq n \left\| P_{\mathcal{A}}(x^{n-1} z) \right\|_F^2 + (n-1) \langle P_{\mathcal{A}}(x^n), x^{n-2} z^2 \rangle - \zeta_i^{2n} + 4\Delta_{\mathcal{A}}. \quad \text{(S.49)}$$

*Proof.* We first show that for all integers $0 \leq s \leq n$

$$\left\| \mathrm{Sym}(x^{n-s} z^s) \right\|_F = \binom{n}{s}^{-1/2}. \quad \text{(S.50)}$$

Let $R \in \mathrm{SO}(D)$ (special orthogonal group). Then we have (denoting Tucker product on the RHS)

$$\left\| \mathrm{Sym}(x^{n-s} z^s) \right\|_F = \left\| \mathrm{Sym}(x^{n-s} z^s) \times_1 R \times_2 \ldots \times_n R \right\|_F \quad \text{(S.51)}$$

by rotational invariance of Frobenius norm. Furthermore,

$$\mathrm{Sym}(x^{n-s} z^s) \times_1 R \times_2 \ldots \times_n R = \mathrm{Sym}(x^{n-s} z^s \times_1 R \times_2 \ldots \times_n R) = \mathrm{Sym}((Rx)^{n-s}(Rz)^s) \quad \text{(S.52)}$$

by a direct calculation. Inserting (S.52) into (S.51), and choosing $R$ appropriately,

$$\left\| \mathrm{Sym}(x^{n-s} z^s) \right\|_F = \left\| \mathrm{Sym}(e_1^{n-s} e_2^s) \right\|_F$$

where $e_1, e_2$ are the first two standard basis vectors of $\mathbb{R}^D$. However, $\mathrm{Sym}(e_1^{n-s} e_2^s)$ is the tensor whose $(i_1, \ldots, i_n)$-entry equals $\binom{n}{s}^{-1}$ if $(i_1, \ldots, i_n)$ consists of $n - s$ ones and $s$ twos (in some order), and equals $0$ otherwise. Hence,

$$\left\| \mathrm{Sym}(e_1^{n-s} e_2^s) \right\|_F = \sqrt{\binom{n}{s} \cdot \binom{n}{s}^{-2}} = \binom{n}{s}^{-1/2}.$$

Then, by the formula for the Riemannian Hessian (S.21),

$$\frac{1}{2n} z^\top \nabla^2_{\mathbb{S}^{D-1}} F_{\hat{\mathcal{A}}}(x) z = n \left\| P_{\hat{\mathcal{A}}}(x^{n-1} z) \right\|_F^2 + (n-1) \langle P_{\hat{\mathcal{A}}}(x^n), x^{n-2} z^2 \rangle - F_{\hat{\mathcal{A}}}(x). \quad \text{(S.53)}$$

To upper-bound this with quantities involving $\mathcal{A}$ rather than $\hat{\mathcal{A}}$, we use the subspace perturbation result Lemma S.16, and (S.50):

$$\left\| P_{\hat{\mathcal{A}}}(x^{n-1} z) \right\|_F^2 \leq \left\| P_{\mathcal{A}}(x^{n-1} z) \right\|_F^2 + \Delta_{\mathcal{A}} \left\| \mathrm{Sym}(x^{n-1} z) \right\|_F^2 = \left\| P_{\mathcal{A}}(x^{n-1} z) \right\|_F^2 + \frac{1}{n} \Delta_{\mathcal{A}},$$

$$\langle P_{\hat{\mathcal{A}}}(x^n), x^{n-2} z^2 \rangle \leq \langle P_{\mathcal{A}}(x^n), x^{n-2} z^2 \rangle + \left\| \mathrm{Sym}(x^{n-2} z^2) \right\|_F \Delta_{\mathcal{A}}$$

$$= \langle P_{\mathcal{A}}(x^n), x^{n-2} z^2 \rangle + \frac{\sqrt{2}}{\sqrt{n(n-1)}} \Delta_{\mathcal{A}} \leq \langle P_{\mathcal{A}}(x^n), x^{n-2} z^2 \rangle + \frac{2}{n-1} \Delta_{\mathcal{A}},$$

$$F_{\hat{\mathcal{A}}}(x) = \left\| P_{\hat{\mathcal{A}}}(x^n) \right\|_F^2 \geq \left\| P_{\mathcal{A}}(x^n) \right\|_F^2 - \Delta_{\mathcal{A}} \geq \left\| P_{\mathrm{Span}(a_i^n)}(x^n) \right\|_F^2 - \Delta_{\mathcal{A}} = \zeta_i^{2n} - \Delta_{\mathcal{A}}.$$

Inserting these bounds into (S.53) gives (S.49) as announced. $\qquad \square$

We now recall some basics about geodesic convexity on the sphere.

**Definition S.20.** Spherical caps and geodesics are defined on the unit sphere as follows.

1. For $y \in \mathbb{S}^{D-1}$ and $r \in (0, 1)$, define the *spherical cap* with center $y$ and height $r$ by

$$B_r(y) := \{ x \in \mathbb{S}^{D-1} : \langle x, y \rangle \geq 1 - r \} \subseteq \mathbb{S}^{D-1}.$$

2. Given distinct points $x_1, x_2 \in B_r(y)$, the *geodesic segment* connecting $x_1$ to $x_2$ is the curve $c : [0, 1] \to B_r(y)$ defined by

$$c(t) := \cos(t\theta)x_1 + \sin(t\theta)x_2^{\perp} \quad \text{for } 0 \leq t \leq 1. \tag{S.54}$$

Here $\theta := \cos^{-1}(\langle x_1, x_2 \rangle) \in (0, \pi)$ is the angle between $x_1$ and $x_2$, and $x_2^{\perp}$ is the component of $x_2$ that is orthogonal to $x_1$ (normalized to lie on the unit sphere) given by

$$x_2^{\perp} := \frac{x_2 - \langle x_1, x_2 \rangle x_1}{\|x_2 - \langle x_1, x_2 \rangle x_1\|_2} = \frac{-\langle x_1, x_2 \rangle}{\sqrt{1 - \langle x_1, x_2 \rangle^2}} x_1 + \frac{1}{\sqrt{1 - \langle x_1, x_2 \rangle^2}} x_2 \in \mathbb{S}^{D-1}.$$

(If $x_1 = x_2$, the geodesic segment connecting $x_1$ to $x_2$ is the constant curve at $x_1$.)

The standard notion of concavity for twice-differentiable functions on the sphere is as follows.

**Definition S.21.** Let $F$ be a real-valued function defined on an open subset of $\mathbb{S}^{D-1}$ containing the spherical cap $B_r(y)$. Assume that $F$ is twice-differentiable.

1. We say $F$ is *geodesically strictly concave* on $B_r(y)$ if the Riemannian Hessian of $F$ is negative definite throughout $B_r(y)$,

$$z^{\top} \nabla^2_{\mathbb{S}^{D-1}} F(x) z < 0, \qquad \forall x \in B_r(y) \; \forall z \in \mathbb{S}^{D-1} \text{ with } x \perp z. \tag{S.55}$$

2. Let $\mu > 0$. We say $F$ is *geodesically $\mu$-strongly concave* on $B_r(y)$ if the Riemannian Hessian of $F$ satisfies the following eigenvalue bound throughout $B_r(y)$,

$$z^{\top} \nabla^2_{\mathbb{S}^{D-1}} F(x) z \leq -\mu, \qquad \forall x \in B_r(y) \; \forall z \in \mathbb{S}^{D-1} \text{ with } x \perp z. \tag{S.56}$$

The terminology in Definition S.21 is justified by the following standard lemma.

**Lemma S.22** (Restricting to geodesics). *Let $F$ be a real-valued function defined on an open subset of $\mathbb{S}^{D-1}$ containing the spherical cap $B_r(y)$. Assume that $F$ is twice-differentiable.*

1. *If $F$ is geodesically strictly concave on $B_r(y)$, then for all geodesic segments $c : [0, 1] \to B_r(y)$ with image contained in $B_r(y)$ and $c(0) \neq c(1)$, the pulled-back function $F \circ c : [0, 1] \to \mathbb{R}$ is strictly concave on $[0, 1]$.*

2. *Let $\mu > 0$. If $F$ is geodesically $\mu$-strongly concave on $B_r(y)$, then for all geodesic segments $c : [0, 1] \to B_r(y)$ with image contained in $B_r(y)$ and $c(0) \neq c(1)$, the pulled-back function $F \circ c : [0, 1] \to \mathbb{R}$ is $\mu L(c)^2$-strongly concave on $[0, 1]$. Here $L(c) := \int_0^1 \|\dot{c}(t)\|_2 dt = \cos^{-1}(\langle c(0), c(1) \rangle)$ denotes the length of $c$.*

*Proof.* See [4, Thm. 11.19(3), Def. 11.3] and [4, Thm. 11.19(2), Def. 11.5] respectively. $\square$

The usual implications of geodesic concavity are as follows.

**Lemma S.23.** *Assume the setup of Lemma S.22.*

1. *If $F$ is geodesically strictly concave on $B_r(y)$, there exists at most one point $x \in B_r(y)$ where $\nabla_{\mathbb{S}^{D-1}} F(x) = 0$. Such a point $x$ is automatically a strict local maximizer of $F$.*

2. *Let $\mu > 0$. If $F$ is geodesically $\mu$-strongly concave on $B_r(y)$, then for all geodesics segments $c$ as in the lemma above, $F \circ c$ is upper-bounded by a concave quadratic function via:*

$$(F \circ c)(t) \leq (F \circ c)(0) + (F \circ c)'(0) \, t - \frac{\mu L(c)^2}{2} t^2, \qquad \forall t \in [0, 1].$$

*Proof.* 1. We first show that for all geodesic segments $c : [0, 1] \to B_r(y)$ with $c(0) \neq c(1)$, the derivative $(F \circ c)'$ is strictly decreasing on $[0, 1]$. To see this, note that since $F$ is twice-differentiable, $F \circ c$ is twice-differentiable; and since $c$ is a geodesic it holds

$$(F \circ c)''(t) = \dot{c}(t)^{\top} \nabla^2_{\mathbb{S}^{D-1}} F(c(t)) \dot{c}(t), \tag{S.57}$$

by [4, Eq. (5.30), page 106]. The right-hand side of (S.57) is strictly negative by (S.55) and the fact $\dot{c}(t) \neq 0$ (note $\|\dot{c}(t)\|_2 = \cos^{-1}(\langle c(0), c(1) \rangle)$ and we are assuming $c(0) \neq c(1)$). Thus $(F \circ c)''$ is strictly negative on $[0, 1]$. Hence by the mean value theorem, $(F \circ c)'$ is strictly decreasing on $[0, 1]$.

Now, to deduce item 1 in the lemma, assume for a contradiction that there exist two distinct points $x_1, x_2 \in B_r(y)$ where the Riemannian gradient vanishes. Let $c$ be the geodesic segment connecting $x_1$ to $x_2$. Again by [4, Eq. (5.30), page 106],

$$(F \circ c)'(t) = \dot{c}(t)^\top \nabla_{\mathbb{S}^{D-1}} F(c(t)).$$

By the vanishing Riemannian gradient assumption, this implies $(F \circ c)'(0) = 0$ and $(F \circ c)'(1) = 0$. But that contradicts the fact that $(F \circ c)'$ is strictly decreasing. So item 1 follows.

2. By Taylor's theorem applied to $F \circ c$, for each $t \in [0, 1]$ there exists $\xi \in [0, t]$ such that

$$(F \circ c)(t) = (F \circ c)(0) + (F \circ c)'(0)\, t + \frac{(F \circ c)''(\xi)}{2} t^2. \tag{S.58}$$

From (S.57), (S.56) and $\|\dot{c}(\xi)\|_2 = L(c)$, we get $(F \circ c)''(\xi) \leq -\mu L(c)^2$. Insert this into (S.58). $\quad\square$

The next tool is how we complete the proofs of our main results, by showing a second-order critical point with sufficiently large functional value must land in a spherical cap around one of $sa_i$.

**Proposition S.24** (Tool #3). *Let $\{a_i : i \in [K]\} \subseteq \mathbb{S}^{D-1}$ be any system of vectors, let $\mathcal{A} = \mathrm{Span}\{a_i^n : i \in [K]\} \subseteq \mathrm{Sym}(\mathcal{T}_D^n)$ be the corresponding subspace of tensors and let $\hat{\mathcal{A}} \subseteq \mathrm{Sym}(\mathcal{T}_D^n)$ be another subspace of tensors which is a perturbation of $\mathcal{A}$ with approximation error $\Delta_\mathcal{A}$. Let $0 < r \leq R < 1$, so there is an inclusion of spherical caps $B_r(sa_i) \subseteq B_R(sa_i)$. Assume $F_{\hat{\mathcal{A}}}$ is geodesically $n$-strongly concave on the inner cap $B_r(sa_i)$, the height of the inner cap satisfies $r > \Delta_\mathcal{A}/n$, and $F_{\hat{\mathcal{A}}}$ is geodesically strictly concave on the outer cap $B_R(sa_i)$. Then, in $B_R(sa_i)$ there exists a unique first-order critical point $x^*$ of **nSPM-P**. Further, this point is a strict local maximizer for **nSPM-P** on $\mathbb{S}^{D-1}$. Finally, the distance between $x^*$ and $sa_i$ is upper-bounded independently of $R$ and $r$ via:*

$$\|x^* - sa_i\|_2^2 \leq \frac{2\Delta_\mathcal{A}}{n}. \tag{S.59}$$

*Proof.* By Lemma S.23(a) and the assumption that $F_{\hat{\mathcal{A}}}$ is strictly concave on the outer cap, there exists at most one critical point of $F_{\hat{\mathcal{A}}}$ in $B_R(sa_i)$. To prove the rest of the proposition, we let the maximum of $F_{\hat{\mathcal{A}}}$ in the inner cap $B_r(sa_i)$ be attained at $x^* \in B_r(sa_i)$; such a point exists by compactness of $B_r(sa_i)$. We will first show that $x^*$ satisfies (S.59), which is the main assertion. To this end, by the choice of $x^*$, note $F_{\hat{\mathcal{A}}}(x^*) \geq F_{\hat{\mathcal{A}}}(sa_i)$. Of course, $1 \geq F_{\hat{\mathcal{A}}}(x^*)$. By the perturbation bound (S.41), $F_{\hat{\mathcal{A}}}(sa_i) \geq F_\mathcal{A}(sa_i) - \Delta_\mathcal{A} = 1 - \Delta_\mathcal{A}$. Combining the last three sentences gives

$$-\Delta_\mathcal{A} \leq F_{\hat{\mathcal{A}}}(sa_i) - F_{\hat{\mathcal{A}}}(x^*). \tag{S.60}$$

Now let $c$ be the geodesic segment (S.54) connecting $x^*$ to $sa_i$. By Lemma S.23(b) and the assumption that $F_{\hat{\mathcal{A}}}$ is $n$-strictly concave on the inner cap, we have $F_{\hat{\mathcal{A}}}(sa_i) - F_{\hat{\mathcal{A}}}(x^*) \leq (F \circ c)'(0) - \frac{nL(c)^2}{2}$. By the choice of $x^*$, $(F \circ c)(0) \geq (F \circ c)(t)$ for all $t \in [0, 1]$, whence $(F \circ c)'(0) \leq 0$. It follows

$$F_{\hat{\mathcal{A}}}(sa_i) - F_{\hat{\mathcal{A}}}(x^*) \leq -\frac{nL(c)^2}{2}. \tag{S.61}$$

Putting Eq. (S.60) and (S.61) together, we get

$$-\Delta_\mathcal{A} \leq -\frac{nL(c)^2}{2} \iff L(c)^2 \leq \frac{2\Delta_\mathcal{A}}{n}. \tag{S.62}$$

Here the geodesic length $L(c)$ is $\theta_0$, where $\theta_0 = \cos^{-1}(\langle x^*, sa_i \rangle)$ is the angle between $x^*$ and $sa_i$ in radians. This is related to the squared Euclidean distance between $x^*$ and $sa_i$ as follows:

$$\|x^* - sa_i\|_2^2 = 2 - 2\langle x^*, sa_i \rangle = 2 - 2\cos(\theta_0).$$

Using the elementary inequality $\cos(\theta) \geq 1 - \frac{1}{2}\theta^2$, which is true for all $\theta \in \mathbb{R}$, we see

$$\|x^* - sa_i\|_2^2 \leq \theta_0^2 = L(c)^2.$$

Thus (S.59) follows from (S.62) as desired. We can quickly obtain the rest of the statement, and see that $x^*$ is a strict local maximizer of $F_{\hat{\mathcal{A}}}$ on $\mathbb{S}^{D-1}$ (hence also a first-order critical point), by noting

$$2 - 2\langle x^*, sa_i \rangle = \|x^* - sa_i\|_2^2 \leq \frac{2\Delta_{\mathcal{A}}}{n} \implies 1 - \langle x^*, sa_i \rangle \leq \frac{\Delta_{\mathcal{A}}}{n}.$$

For then from the assumption that $r > \Delta_{\mathcal{A}}/n$, we get $1 - \langle x^*, sa_i \rangle < r$. Hence $x^*$ lies strictly in the interior of the inner cap $B_r(sa_i)$. Therefore $x^*$ is a local maximizer of $F_{\hat{\mathcal{A}}}$ on $\mathbb{S}^{D-1}$. By Lemma S.23(a) and strict concavity, $x^*$ is in fact a strict local maximizer. This ends the proof. $\quad\square$

# G  Proof of Theorem M.7

In this section we prove our deterministic theorem. Here we define $\zeta$, $\sigma$ and $R_{i,s}$ as in Definition S.1.

## G.1  Bounds on $R_{i,s}$ and $\|G_n^{-1}\|_2$

We start by relating the auxiliary scalars $R_{i,s}$ to the frame constants $\rho_s$.

**Lemma S.25.** *For each $i \in [K]$ and $s \in [n]$, we have*

$$|R_{i,s}| \leq \max_{\ell:\ell\neq i} |\sigma_\ell \zeta_\ell^{n-s}| \rho_s \leq \|\sigma \odot \zeta^{\odot n-s}\|_\infty \rho_s. \tag{S.63}$$

*If $s$ is even, then we have a slight refinement:*

$$\min_{\ell:\ell\neq i} \sigma_\ell \zeta_\ell^{n-s} \rho_s \leq R_{i,s} \leq \max_{\ell:\ell\neq i} \sigma_\ell \zeta_\ell^{n-s} \rho_s. \tag{S.64}$$

*Proof.* For each $i$ and $s$, use the definition of $R_{i,s}$, the triangle inequality and the definition of $\rho_s$:

$$|R_{i,s}| = |\sum_{\ell:\ell\neq i} \sigma_\ell \zeta_\ell^{n-s} \langle a_i, a_\ell \rangle^s| \leq \sum_{\ell:\ell\neq i} |\sigma_\ell \zeta_\ell^{n-s}||\langle a_i, a_\ell \rangle|^s \leq \max_{\ell:\ell\neq i} |\sigma_\ell \zeta_\ell^{n-s}| \sum_{\ell:\ell\neq i} |\langle a_i, a_\ell \rangle|^s$$

$$\leq \max_{\ell:\ell\neq i} |\sigma_\ell \zeta_\ell^{n-s}|(\sum_{\ell=1}^{K} |\langle a_i, a_\ell \rangle|^s - 1) \leq \max_{\ell:\ell\neq i} |\sigma_\ell \zeta_\ell^{n-s}| \rho_s \leq \|\sigma \odot \zeta^{\odot n-s}\|_\infty \rho_s.$$

If $s$ is even, then each term $\langle a_i, a_\ell \rangle^s$ is nonnegative. So we have

$$R_{i,s} = \sum_{\ell:\ell\neq i} \sigma_\ell \zeta_\ell^{n-s} \langle a_i, a_\ell \rangle^s \leq \max_{\ell:\ell\neq i} \sigma_\ell \zeta_\ell^{n-s} \sum_{\ell:\ell\neq i} \langle a_i, a_\ell \rangle^s \leq \max_{\ell:\ell\neq i} \sigma_\ell \zeta_\ell^{n-s} \rho_s,$$

and likewise $R_{i,s} \geq \min_{\ell:\ell\neq i} \sigma_\ell \zeta_\ell^{n-s} \rho_s$ as announced. $\quad\square$

**Remark S.26.** The frame constants are small under the assumptions of Theorem M.7 with $\tau > 0$, for then:

$$n^2 \rho_2 \leq n^2 \rho_2 + (n^2 + n)\rho_n < \frac{1}{6} \implies \rho_2 < \frac{1}{6n^2} \leq \frac{1}{24} \tag{S.65}$$

$$\text{and } (n^2 + n)\rho_n \leq n^2 \rho_2 + (n^2 + n)\rho_n < \frac{1}{6} \implies \rho_n < \frac{1}{6(n^2 + n)} = \frac{1}{36}. \tag{S.66}$$

Consequently in the setting of Theorem M.7, the system $\{a_i^n : i \in [K]\}$ is linearly independent.

**Corollary S.27.** *Under the assumptions of Theorem M.7, the Grammian matrix given by $(G_n)_{i,j} = \langle a_i, a_j \rangle^n$ is invertible. In fact,*

$$\|G_n^{-1}\|_2 \leq \frac{1}{1 - \rho_n} \leq \frac{36}{35}.$$

*Proof.* This follows immediately from (S.66), and the bound $1 - \rho_n \leq \mu_n(G_n)$ in Lemma M.6. $\quad\square$

## G.2 Lower bound on maximum correlation coefficient

We begin the proof of Theorem M.7 by showing that any constrained second-order critical point $x$ has at least one large correlation coefficient.

**Proposition S.28.** *Consider a system $\{a_i : i \in [K]\}$ that satisfies the assumptions of Theorem M.7. For a second-order critical point $x$ of* **nSPM-P***, then either $F_{\mathcal{A}}(x) = 0$ or we have*

$$
\|\zeta\|_\infty^2 \geq 1 - \left(1 + \frac{n}{2(n-1)}\right)\frac{\rho_2}{1+\rho_2} - \frac{n}{2(n-1)}\frac{\rho_n}{(1-2\rho_n)(1+\rho_2)} - \frac{4n-2}{2(n-1)}\frac{\Delta_{\mathcal{A}}}{F_{\mathcal{A}}(x)}
$$

$$
\geq 1 - 2\rho_2 - 2\rho_n - 3\frac{\Delta_{\mathcal{A}}}{F_{\mathcal{A}}(x)}.
$$

*Proof.* Fix a second-order critical point $x \in \mathbb{S}^{D-1}$ of **nSPM-P**, and let $\zeta = \zeta(x) = A^\top x \in \mathbb{R}^K$. Since we can change $a_\ell$ to $-a_\ell$ without altering the function $F_{\mathcal{A}}$ or the constants $\rho_s$, we may assume without loss of generality that $\zeta_\ell \geq 0$ for each $\ell \in [K]$. We may also assume that $F_{\mathcal{A}}(x) \neq 0$.

Our starting point is Eq. (S.44) in Proposition S.18, which says that for each $i \in [K]$ we have

$$
\left(1 + 2(n-1)\zeta_i^2\right)F_{\mathcal{A}}(x) \geq (2n-1)\sigma_i\zeta_i^{n-2} + n\zeta_i^{n-2}R_{i,n} + (n-1)R_{i,2} - (4n-2)\Delta_{\mathcal{A}}.
$$
(S.67)

Since we want to deduce that $\|\zeta\|_\infty$ is large, we fix $i \in [K]$ to be an index such that the first (main) term on the right-hand side of (S.67) is maximal; that is, let $i \in \arg\max_{\ell \in [K]} \sigma_\ell \zeta_\ell^{n-2}$. Now notice

$$
0 < F_{\mathcal{A}}(x) = \langle P_{\mathcal{A}}(x^n), x^n \rangle = \langle \sum_{\ell=1}^K \sigma_\ell a_\ell^n, x^n \rangle = \sum_{\ell=1}^K \sigma_\ell \zeta_\ell^n \leq \sigma_i \zeta_i^{n-2} \sum_{\ell=1}^K \zeta_\ell^2,
$$
(S.68)

from which it follows that $\sigma_i\zeta_i^{n-2} > 0$. In (S.67), our goal is to bound $F_{\mathcal{A}}(x)$ on the left-hand side from above, and $R_{i,n}$, $R_{i,2}$ on the right-hand side from below, all by quantities involving $\zeta_i$.

Firstly using (S.68), the fact $\sigma_i\zeta_i^{n-2} > 0$ and the definition of $\rho_2$, we have

$$
F_{\mathcal{A}}(x) \leq \sum_{\ell=1}^K \sigma_\ell \zeta_\ell^n \leq \sigma_i \zeta_i^{n-2} \sum_{\ell=1}^K \zeta_\ell^2 = \sigma_i\zeta_i^{n-2}(1+\rho_2).
$$
(S.69)

Then by Lemma S.25, Eq. (S.64) with $s = n$ we have

$$
R_{i,2} \geq \min_{\ell:\ell\neq i} \sigma_\ell \zeta_\ell^{n-2}\rho_2,
$$
(S.70)

while by (S.63) with $s = n$ we have

$$
\zeta_i^{n-2}R_{i,n} \geq -\zeta_i^{n-2}\|\sigma\|_\infty \rho_n.
$$
(S.71)

To further bound the right-hand sides in (S.70) and (S.71) in terms of $\zeta_i$, we now show that $\sigma_i \approx \|\sigma_\infty\|$ and that $\min_{\ell\neq i}\sigma_\ell\zeta_\ell^{n-2} \geq -\sigma_i\zeta_i^{n-2}$.

*Showing $\sigma_i \approx \|\sigma\|_\infty$:* Let $j \in [K]$ be an index such that $|\sigma_j| = \|\sigma\|_\infty$. We note that $\sigma_j \geq 0$, because if $\sigma_j < 0$, then using $\zeta_j \geq 0$ and (S.43) we would have

$$
\|\sigma\|_\infty = |\sigma_j| \leq \left|\sigma_j - \zeta_j^n\right| = |R_{j,n}| | \sum_{\ell\neq j} \sigma_\ell\langle a_j, a_\ell\rangle^n| \leq \|\sigma\|_\infty \rho_n.
$$

This contradicts the assumption $\rho_n < 1$, so indeed $\sigma_j \geq 0$. Now consider the three facts:

1. $\sigma_i\zeta_i^{n-2} \geq \sigma_j\zeta_i^{n-2}$ by definition of $i$;

2. $\sigma_j \geq \sigma_i$ by $\sigma_j \geq 0$ and definition of $j$;

3. $\zeta_i \geq 0$ and $\zeta_j \geq 0$.

It follows that $\zeta_i \geq \zeta_j$. Using this and the bound for the auxiliary constants (S.63), we can estimate:

$$\sigma_j - \sigma_i \leq \sigma_j - \zeta_j^n + \zeta_i^n - \sigma_i = -R_{j,n} + R_{i,n} \leq |R_{j,n}| + |R_{i,n}| \leq 2\sigma_j\rho_n.$$

Rearranging gives

$$\sigma_j \geq \sigma_i \geq (1 - 2\rho_n)\sigma_j = (1 - 2\rho_n)\|\sigma\|_\infty. \tag{S.72}$$

By the assumptions in Theorem M.7 and (S.66), in particular $\rho_n/(1 - 2\rho_n) > 0$. Hence multiplying throughout by $\rho_n/(1 - 2\rho_n)$ in (S.72) yields

$$\frac{\rho_n}{1 - 2\rho_n}\sigma_i \leq \rho_n\|\sigma\|_\infty,$$

and then substituting into (S.71) we obtain

$$\zeta_i^{n-2}R_{i,n} \geq -\zeta_i^{n-2}\|\sigma\|_\infty\rho_n \geq -\zeta_i^{n-2}\frac{\rho_n}{1 - 2\rho_n}\sigma_i. \tag{S.73}$$

_Showing $\min_{\ell \neq i} \sigma_\ell\zeta_\ell^{n-2} \geq -\sigma_i\zeta_i^{n-2}$_: Assume to the contrary that $\min_{\ell \neq i} \sigma_\ell\zeta_\ell^{n-2} < -\sigma_i\zeta_i^{n-2}$ and let $k$ be an index attaining the minimum on the left-hand side. Since $\sigma_i\zeta_i^{n-2} > 0$ and $\zeta_k^{n-2} \geq 0$, it must be that $\sigma_k < 0$. Using $\sigma_k < 0$ and $\zeta_k \geq 0$, we estimate

$$|\sigma_k| \leq |\sigma_k - \zeta_k^n|_\infty = |R_{k,n}| \leq \|\sigma\|_\infty\rho_n.$$

This implies

$$\left\|\sigma \odot \zeta^{n-2}\right\|_\infty = \max\{\sigma_i\zeta_i^{n-2}, -\sigma_k\zeta_k^{n-2}\} = -\sigma_k\zeta_k^{n-2} \leq \|\sigma\|_\infty\zeta_k^{n-2}\rho_n.$$

Similarly to the previous part, using $-\sigma_k\zeta_k^{n-2} > \sigma_k\zeta_j^{n-2}$ we can see $\zeta_k \geq \zeta_j$, where $j$ is again the index such that $\sigma_j = \|\sigma\|_\infty$. Hence, as before, we have

$$\sigma_j - \sigma_k = \sigma_j - \zeta_j^n + \zeta_k^n - \sigma_k = -R_{j,n} + R_{k,n} \leq 2\sigma_j\rho_n,$$

which rearranges to $\sigma_k \geq (1 - 2\rho_n)\sigma_j$. Since we have previously shown $\sigma_j \geq 0$ and we know $\rho_n < 1/2$ by (S.66), this is in contradiction with $\sigma_k < 0$. Thus it follows that $\min_{\ell:\ell \neq i} \sigma_\ell\zeta_\ell^{n-2} \geq -\sigma_i\zeta_i^{n-2}$ as desired. Substituting into (S.70) yields

$$R_{i,2} \geq \min_{\ell:\ell \neq i} \sigma_\ell\zeta_\ell^{n-2}\rho_2 \geq -\sigma_i\zeta_i^{n-2}\rho_2. \tag{S.74}$$

We can now complete the proof of the proposition. Substituting the bounds (S.69), (S.73), and (S.74) into the second-order criticality inequality (S.67), we get

$$\left(1 + 2(n-1)\zeta_i^2\right)(1 + \rho_2)\sigma_i\zeta_i^{n-2} \geq (2n-1)\sigma_i\zeta_i^{n-2} - n\frac{\rho_n}{1 - 2\rho_n}\sigma_i\zeta_i^{n-2}$$
$$- (n-1)\rho_2\sigma_i\zeta_i^{n-2} - (4n-2)\Delta_\mathcal{A}.$$

Dividing by $(1 + \rho_2)\sigma_i\zeta_i^{n-2} > 0$, subtracting 1 and then dividing by $2(n-1)$, this simplifies to

$$\zeta_i^2 \geq \frac{1}{1 + \rho_2} - \frac{n}{2(n-1)}\left(\frac{\rho_n}{(1 - 2\rho_n)(1 + \rho_2)} + \frac{\rho_2}{1 + \rho_2}\right) - \frac{4n-2}{2(n-1)}\frac{\Delta_\mathcal{A}}{(1 + \rho_2)\sigma_i\zeta_i^{n-2}}.$$

We replace the denominator in the last term by $F_\mathcal{A}(x)$ by re-using the bound $F_\mathcal{A}(x) \leq \sigma_i\zeta_i^{n-2}(1 + \rho_2)$ (S.69). Then we rearrange the terms and substitute $\|\zeta\|_\infty^2 \geq \zeta_i^2$. This gives the first inequality in the proposition. The second stated inequality is an immediate consequence of the first, where we simplify each term up to a constant factor using $\frac{n}{2(n-1)} \leq 1$, $1 + \rho_2 \geq 1$, $1 - 2\rho_n > \frac{1}{2}$ and $\frac{4n-2}{2(n-1)} \leq 3$. $\quad\square$

## G.3 Concavity

In the next proof step we analyze the Riemannian Hessian and show that it is strictly negative definite.

**Proposition S.29.** *Consider a system $\{a_i : i \in [K]\}$ that satisfies the assumptions in Theorem M.7. For any $x, z \in \mathbb{S}^{D-1}$ with $z \perp x$, the Riemannian Hessian of $F_{\hat{\mathcal{A}}}$ satisfies*

$$z^\top \nabla^2_{\mathbb{S}^{D-1}} F_{\hat{\mathcal{A}}}(x) z \leq -2n + 2n^2 \frac{2 - \rho_n}{1 - \rho_n} \left( 1 - \|\zeta\|^{2n}_\infty \right) + 2n(3n - 2) \frac{\rho_n}{1 - \rho_n} + 8n\Delta_{\mathcal{A}}. \quad \text{(S.75)}$$

*Proof.* Let $i \in [K]$ be such that $|\zeta_i| = \|\zeta\|_\infty$ where $\zeta = \zeta(x) = A^\top x$. Our starting point is (S.49) in Proposition S.19, which reads

$$\frac{1}{2n} z^\top \nabla^2_{\mathbb{S}^{D-1}} F_{\hat{\mathcal{A}}}(x) z \leq n \left\| P_{\mathcal{A}}(x^{n-1} z) \right\|^2_F + (n-1)\langle P_{\mathcal{A}}(x^n), x^{n-2} z^2 \rangle - \zeta_i^{2n} + 4\Delta_{\mathcal{A}}. \quad \text{(S.76)}$$

Expanding $P_{\mathcal{A}}(x^n) = \sum_{\ell=1}^K \sigma_\ell a_\ell^n$ as in Definition S.1, we can upper-bound the second term on the right-hand side of (S.76) as follows by applying Hölder's inequality and the definition of $\rho_n$:

$$\langle P_{\mathcal{A}}(x^n), x^{n-2} z^2 \rangle = \sum_{\ell=1}^K \sigma_\ell \zeta_\ell^{n-2} \langle a_\ell, z \rangle^2 = \sum_{\ell=1}^K \zeta_\ell^{2n-2} \langle a_\ell, z \rangle^2 + \sum_{\ell=1}^K (\sigma_\ell - \zeta_\ell^n) \zeta_\ell^{n-2} \langle a_\ell, z \rangle^2$$

$$\leq \sum_{\ell=1}^K \zeta_\ell^{2n-2} \langle a_\ell, z \rangle^2 + \|\sigma - \zeta^{\odot n}\|_\infty \sum_{\ell=1}^K \zeta_\ell^{n-2} \langle a_\ell, z \rangle^2$$

$$\leq \sum_{\ell=1}^K \zeta_\ell^{2n-2} \langle a_\ell, z \rangle^2 + \|\sigma - \zeta^{\odot n}\|_\infty \left( \sum_{\ell=1}^K \zeta_\ell^n \right)^{\frac{n-2}{n}} \left( \sum_{\ell=1}^K \langle a_\ell, z \rangle^n \right)^{\frac{2}{n}}$$

$$\leq \sum_{\ell=1}^K \zeta_\ell^{2n-2} \langle a_\ell, z \rangle^2 + \|\sigma - \zeta^{\odot n}\|_\infty (1 + \rho_n). \quad \text{(S.77)}$$

Further we can quickly bound the second term in the right-hand side of (S.77). Firstly by (S.43) and (S.63), we have $\|\sigma - \zeta^{\odot n}\|_\infty = \max_{i \in [K]} |R_{i,n}| \leq \|\sigma\|_\infty \rho_n$. Then we can bound $\|\sigma\|_\infty$ via

$$\|\sigma\|_\infty = \|\sigma - \zeta^{\odot n} + \zeta^{\odot n}\|_\infty \leq 1 + \|\sigma - \zeta^{\odot n}\|_\infty \leq 1 + \|\sigma\|_\infty \rho_n \quad \Longrightarrow \quad \|\sigma\|_\infty \leq (1 - \rho_n)^{-1}.$$

Putting the last two sentences together gives us

$$\|\sigma - \zeta^{\odot n}\|_\infty (1 + \rho_n) \leq \|\sigma\|_\infty \rho_n (1 + \rho_n) \leq \frac{1 + \rho_n}{1 - \rho_n} \rho_n. \quad \text{(S.78)}$$

As for the first term on the right-hand side of (S.77), we first split the sum:

$$\sum_{\ell=1}^K \zeta_\ell^{2n-2} \langle a_\ell, z \rangle^2 = \zeta_i^{2n-2} \langle a_i, z \rangle^2 + \sum_{\ell: \ell \neq i} \zeta_\ell^{2n-2} \langle a_\ell, z \rangle^2. \quad \text{(S.79)}$$

We use Bessel's inequality and the assumptions $x, z \in \mathbb{S}^{D-1}$ and $x \perp z$ to get

$$\zeta_i^{2n-2} \langle a_i, z \rangle^2 = \zeta_i^{2n-2} (\langle a_i, z \rangle^2 + \langle a_i, x \rangle^2 - \zeta_i^2) \leq \zeta_i^{2n-2} (\|a_i\|_2^2 - \zeta_i^2) \leq \zeta_i^{2n-2} (1 - \zeta_i^2). \quad \text{(S.80)}$$

Then we estimate

$$\sum_{\ell: \ell \neq i} \zeta_\ell^{2n-2} \langle a_\ell, z \rangle^2 \leq \sum_{\ell: \ell \neq i} \zeta_\ell^{2n-2} = \sum_{\ell=1}^K \zeta_\ell^{2n-2} - \zeta_i^{2n-2} \leq 1 + \rho_{2n-2} - \zeta_i^{2n-2} \leq 1 + \rho_n - \zeta_i^{2n-2}, \quad \text{(S.81)}$$

where the last inequality used $2n - 2 \geq n$. Finally we substitute (S.80) and (S.81) into (S.79) to get

$$\sum_{\ell=1}^K \zeta_\ell^{2n-2} \langle a_\ell, z \rangle^2 \leq 1 - \zeta_i^{2n} + \rho_n. \quad \text{(S.82)}$$

Combining (S.77), (S.82), (S.78) we arrive at:

$$\langle P_{\mathcal{A}}(x^n), x^{n-2} z^2 \rangle \leq 1 - \zeta_i^{2n} + \rho_n + \frac{1 + \rho_n}{1 - \rho_n} \rho_n. \quad \text{(S.83)}$$

At this point, we turn to the first term on the right-hand side of (S.76). Recalling Lemma S.5, Definition S.1 and Corollary S.27, we may write

$$\left\|P_{\mathcal{A}}(x^{n-1}z)\right\|_F^2 = \eta(x^{n-1}z)^\top G_n^{-1}\eta(x^{n-1}z), \tag{S.84}$$

where $\eta(x^{n-1}z)_\ell := \langle x^{n-1}z, a_\ell^n \rangle = \zeta_\ell^{n-1}\langle a_\ell, z \rangle$. Clearly (S.84) is less than or equal to

$$\left\|G_n^{-1}\right\|_2 \|\eta(x^{n-1}z)\|_2^2 = \left\|G_n^{-1}\right\|_2 \sum_{\ell=1}^K \zeta_\ell^{2n-2}\langle a_\ell, z \rangle^2.$$

Re-using the just-proven (S.82) as well as Corollary S.27, it follows that

$$\left\|P_{\mathcal{A}}(x^{n-1}z)\right\|_F^2 \le \left\|G_n^{-1}\right\|_2 \sum_{\ell=1}^K \zeta_\ell^{2n-2}\langle a_\ell, z \rangle^2 \le \frac{1 - \zeta_i^{2n} + \rho_n}{1 - \rho_n}. \tag{S.85}$$

To complete the proof, we plug (S.85) and (S.83) into (S.76) and rearrange terms to get (S.75). □

As a corollary of S.29, we can now prove the existence of exactly one local maximizer within spherical caps of the global maximizers $\pm a_i's$ of the clean objective $F_{\mathcal{A}}$.

**Corollary S.30.** *Consider a system $\{a_i : i \in [K]\}$ that satisfies the assumptions in Theorem M.7. Define the height $r_+ \in (0,1)$ by*

$$1 - r_+ := \left(1 - \frac{1}{n}\frac{1-\rho_n}{2-\rho_n} + \frac{3n-2}{n}\frac{\rho_n}{2-\rho_n} + \frac{4}{n}\frac{1-\rho_n}{2-\rho_n}\Delta_{\mathcal{A}}\right)^{\frac{1}{2n}}$$

$$\le \left(1 - \frac{1}{3n} + 2\rho_n + \frac{2}{n}\Delta_{\mathcal{A}}\right)^{\frac{1}{2n}}.$$

*Then the $2K$ spherical caps $B_{r_+}(a_1), B_{r_+}(-a_1), B_{r_+}(a_2), \ldots, B_{r_+}(-a_K)$ are disjoint. Further for each $s \in \{-1, +1\}$ and $i \in [K]$, there exists exactly one first-order critical point $x_*$ of $F_{\hat{\mathcal{A}}}$ in $B_{r_+}(sa_i)$. Morerover $x_*$ is a strict local maximum of $F_{\hat{\mathcal{A}}}$ (hence second-order critical), and*

$$\|x_* - sa_i\|_2^2 \le \frac{2\Delta_{\mathcal{A}}}{n}.$$

*Proof.* First, we check that $r_+ \in (0,1)$. Indeed $r_+ > 0$ is clear from (S.66), and $r_+ < 1$ is equivalent to $(3n-1)\rho_n + 4(1-\rho_n)\Delta_{\mathcal{A}} < 1$ which holds by the assumptions in Theorem M.7. Also $1 - r_+ \le (1 - \frac{1}{3n} + 2\rho_n + \frac{2}{n}\Delta_{\mathcal{A}})^{\frac{1}{2n}}$ as stated, by the bound on $\rho_n$ (S.66).

Next, we check that the $2K$ spherical caps are disjoint. For a contradiction, suppose there exist distinct $(s,i), (s',i') \in \{-1,1\} \times [K]$ and $x \in \mathbb{S}^{D-1}$ such that $x \in B_{r_+}(sa_i) \cap B_{r_+}(s'a_{i'})$. Thus $\langle x, sa_i \rangle \ge 1 - r_+$ and $\langle x, s'a_{i'} \rangle \ge 1 - r_+$. By the triangle inequality with respect to geodesic distances on the sphere, we know $\arccos\langle sa_i, s'a_{i'} \rangle \le \arccos\langle sa_i, x \rangle + \arccos\langle x, s'a_{i'} \rangle \le 2\arccos(1 - r_+)$, hence by the double angle formula for cosine $\langle sa_i, s'a_{i'} \rangle \ge 2(1 - r_+)^2 - 1$. But note that this gives a lower bound on $\rho_2$: letting $y \in \mathbb{S}^{D-1}$ be the midpoint of the arc joining $sa_i$ and $s'a_{i'}$, we have

$$\rho_2 \ge \langle y, a_i \rangle^2 + \langle y, a_{i'} \rangle^2 - 1 = \langle y, sa_i \rangle^2 + \langle y, s'a_{i'} \rangle^2 - 1 = \langle sa_i, s'a_{i'} \rangle \ge 2(1-r_+)^2 - 1,$$

where the second equality is again by the double angle formula. Therefore

$$\rho_2 \ge 2(1-r_+)^2 - 1 \ge 2\left(1 - \frac{1}{2n}\right)^{\frac{1}{n}} - 1 > 2\left(1 - \frac{1}{2n}\right) - 1 = 1 - \frac{1}{n},$$

which contradicts the implied upper bound on $\rho_2$ (S.65). So the $2K$ spherical caps are disjoint.

For the remaining statements we apply Proposition S.24. Thus it suffices to check that $F_{\hat{\mathcal{A}}}$ is strictly $n$-concave on $B(sa_i, 1 - \frac{\Delta_{\mathcal{A}}}{n})$ and strictly concave in the interior of $B_{r_+}(sa_i)$. To verify these, we will use Proposition S.29 to upper-bound the eigenvalues of the Riemmanian Hessian of $F_{\hat{\mathcal{A}}}$.

So for the strict $n$-concavity condition, we want to show:

$$-2n + 2n^2\frac{2 - \rho_n}{1 - \rho_n}\left(1 - (1 - \frac{\Delta_{\mathcal{A}}}{n})^{2n}\right) + 2n(3n-2)\frac{\rho_n}{1 - \rho_n} + 8n\Delta_{\mathcal{A}} < -n. \tag{S.86}$$

Since $(1-x)^{2n} \geq 1 - 2nx$ for $x \in \mathbb{R}$, we have

$$\left(1 - \frac{\Delta_{\mathcal{A}}}{n}\right)^{2n} \geq 1 - 2\Delta_{\mathcal{A}},$$

whence (S.86) is implied by

$$-2n + 4n^2 \frac{2 - \rho_n}{1 - \rho_n} \Delta_{\mathcal{A}} + 2n(3n - 2) \frac{\rho_n}{1 - \rho_n} + 8n\Delta_{\mathcal{A}} < -n. \tag{S.87}$$

Rearranging terms, Eq. (S.87) is equivalent to

$$(4n(2 - \rho_n) + 8(1 - \rho_n)) \Delta_{\mathcal{A}} + (6n - 3)\rho_n < 1. \tag{S.88}$$

However indeed, (S.88) holds as a consequence of the assumptions in Theorem M.7.

As for the concavity condition on $B_{r_+}(sa_i)$, we want to show:

$$-2n + 2n^2 \frac{2 - \rho_n}{1 - \rho_n}(1 - (1 - r_+)^{2n}) + 2n(3n - 2)\frac{\rho_n}{1 - \rho_n} + 8n\Delta_{\mathcal{A}} \leq 0. \tag{S.89}$$

However upon substituting in the definition of $r_+$, we see that the left-hand side of (S.89) is precisely 0, i.e., $r_+$ was chosen to be maximal so that (S.89) holds. This completes the proof of the corollary. $\square$

### G.4 Completing the proof of Theorem M.7

Finally, we can conclude the proof of Theorem M.7 by combining the previous three steps.

*Proof of Theorem M.7.* Let $\ell = \frac{3n^2 + 2}{2\tau}\Delta_{\mathcal{A}} + \Delta_{\mathcal{A}}$. It suffices to verify the following two claims:

1. Each second-order critical point $x^*$ of **nSPM-P** satisfying $F_{\hat{\mathcal{A}}}(x^*) > \ell$ must lie in one of the spherical caps $B_{r_+}(a_1), B_{r_+}(-a_1), \ldots, B_{r_+}(-a_K)$ from Corollary S.30.

2. The level $\ell$ satisfies $\ell < 1 - \Delta_{\mathcal{A}}$.

It is easy to see that these statements imply Theorem M.7. By the first claim and Corollary S.30, there are *at most* $2K$ second-order critical points $x^*$ with $F_{\hat{\mathcal{A}}}(x^*) > \ell$. They are all strict local maximizers of $F_{\hat{\mathcal{A}}}$. For each such point, there exists unique $i \in [K], s \in \{-1, +1\}$ with $\|x^* - sa_i\|_2^2 \leq \frac{2\Delta_{\mathcal{A}}}{n}$. Meanwhile by the second claim, for each $s \in \{-1, +1\}, i \in [K]$ the maximizer $x^*$ of $F_{\hat{\mathcal{A}}}$ in $B_{r_+}(sa_i)$ from Corollary S.30 satisfies $F_{\hat{\mathcal{A}}}(x) \geq F_{\hat{\mathcal{A}}}(sa_i) \geq F_{\mathcal{A}}(sa_i) - \Delta_{\mathcal{A}} = 1 - \Delta_{\mathcal{A}} > \ell$. Thus there are *at least* $2K$ second-order critical points in the superlevel set, and the theorem follows.

Now we verify the first claim. Letting $\zeta = A^\top x^*$, we want $\|\zeta\|_\infty \geq 1 - r_+$. Indeed using the bound

$$\frac{\Delta_{\mathcal{A}}}{F_{\mathcal{A}}(x^*)} \leq \frac{\Delta_{\mathcal{A}}}{F_{\hat{\mathcal{A}}}(x^*) - \Delta_{\mathcal{A}}} < \frac{\Delta_{\mathcal{A}}}{\ell - \Delta_{\mathcal{A}}} = \frac{2\tau}{2 + 3n^2}$$

$$= \frac{2\tau}{3n^2}\frac{3n^2}{2 + 3n^2} \leq \frac{2\tau}{3n^2}\frac{3n^2 + 4\tau}{2 + 3n^2 + 4\tau} = \frac{2\tau}{3n^2}\left(1 - \frac{2}{2 + 3n^2 + 4\tau}\right) = \frac{2}{3n^2}(\tau - \Delta_0)$$

$$\leq \frac{2}{3n^2}(\tau - \Delta_{\mathcal{A}}),$$

where we used $\frac{a+c}{b+c} \geq \frac{a}{b}$ if $b \geq a \geq 0$ and $c \geq 0$, Proposition S.28 gives

$$\|\zeta\|_\infty^2 \geq 1 - 2\rho_2 - 2\rho_n - 3\frac{\Delta_{\mathcal{A}}}{F_{\mathcal{A}}(x)} \geq 1 - 2\rho_2 - 2\rho_n - \frac{2}{n^2}(\tau - \Delta_{\mathcal{A}})$$

$$= 1 - 2\rho_2 - 2\rho_n - \frac{2}{n^2}\left(\frac{1}{6} - \Delta_{\mathcal{A}} - n^2\rho_2 - (n^2 + n)\rho_n\right)$$

$$= 1 - \frac{1}{3n^2} + \frac{2}{n}\rho_n + \frac{2}{n^2}\Delta_{\mathcal{A}}$$

$$\geq \left(1 - \frac{1}{3n} + 2\rho_n + \frac{2}{n}\Delta_{\mathcal{A}}\right)^{\frac{1}{n}}$$

$$\geq (1 - r_+)^2.$$

Note that the penultimate line used $(1 - x)^{\frac{1}{n}} \leq 1 - \frac{x}{n}$ for $x \leq 1$, and the last line was a part of Corollary S.30.

Next we check the second claim. Substituting the definition of $\ell$, we require

$$\frac{3n^2 + 2}{2\tau}\Delta_{\mathcal{A}} + \Delta_{\mathcal{A}} < 1 - \Delta_{\mathcal{A}} \iff \Delta_{\mathcal{A}} < \frac{1}{\frac{3n^2+2}{2\tau} + 2} = \Delta_0.$$

But this is guaranteed by the assumption $\Delta_{\mathcal{A}} < \Delta_0$. The proof of Theorem M.7 is finished. $\quad\square$

## H   Proof of Theorem M.16

In this section we prove our random overcomplete theorem. Here we define $\zeta$, $\sigma$ and $R_{i,s}$ as in Definition S.1. We recall the shorthand $\varepsilon_K = K \log^n(K)/D^n$. Furthermore, as required by the statement of Theorem M.16, we assume that conditions hold **A1**–**A3** hold.

### H.1   Bounds on $R_{i,s}$

A central part of the proof is to bound the remainder $|R_{i,s}|$. We start with a general result that bounds inner products with the basis coefficients $\sigma$. Then we deduce a first bound on $R_{i,s}$.

**Lemma S.31.** *Assume that conditions **A2** and **A3** hold. For any vector $\xi \in \mathbb{R}^K$, we have*

$$\left|\sigma^\top \xi\right| \leq \sqrt{c_2 F_{\mathcal{A}}(x)}\|\xi\|_2. \tag{S.90}$$

*In particular, letting $R_s = \max_i |R_{i,s}|$ as in Definition S.1, we have*

$$R_s \leq \sqrt{c_2 c_1^s F_{\mathcal{A}}(x)}\, \varepsilon_K^{\frac{s}{2n}}\, \|\zeta\|_{2n}^{n-s}. \tag{S.91}$$

*Proof.* Since $G_n^{-1}$ is symmetric and positive definite, we can reason as follows for the first claim:

$$\left|\sigma^\top \xi\right| = \left|(\zeta^{\odot n})^\top G_n^{-1}\xi\right| \tag{Eq. (S.14)}$$

$$\leq \sqrt{(\zeta^{\odot n})^\top G_n^{-1}\zeta^{\odot n}}\sqrt{\xi^\top G_n^{-1}\xi} \qquad \text{(Cauchy-Schwartz w.r.t. } G_n^{-1})$$

$$= \sqrt{F_{\mathcal{A}}(x)}\sqrt{\xi^\top G_n^{-1}\xi} \tag{Eq. (S.12)}$$

$$\leq \sqrt{F_{\mathcal{A}}(x)}\sqrt{\|G_n^{-1}\|_2}\|\xi\|_2$$

$$\leq \sqrt{c_2 F_{\mathcal{A}}(x)}\|\xi\|_2 \qquad \text{(condition } \mathbf{A3}).$$

For (S.91), we note that $R_{i,s} = \sigma^\top \xi$, where $\xi \in \mathbb{R}^K$ is given by $\xi_j = \mathbf{1}(j \neq i)\zeta_j^{n-s}\langle a_i, a_j\rangle^s$ (each $j \in [K]$). Substituting this into (S.90) gives (S.91), because

$$\|\xi\|_2 = \sqrt{\sum_{j:j\neq i} \zeta_j^{2n-2s}\langle a_i, a_j\rangle^{2s}}$$

$$\leq \sqrt{\left(\sum_{j:j\neq i} \zeta_j^{2n}\right)^{\frac{2n-2s}{2n}}\left(\sum_{j:j\neq i} \langle a_i, a_j\rangle^{2n}\right)^{\frac{2s}{2n}}} \qquad \text{(Hölder's inequality)}$$

$$\leq \|\zeta\|_{2n}^{n-s}\left(K\left(\frac{c_1 \log(K)}{D}\right)^n\right)^{\frac{s}{2n}} \qquad \text{(condition } \mathbf{A2})$$

$$= c_1^{\frac{s}{2}}\varepsilon_K^{\frac{s}{2n}}\, \|\zeta\|_{2n}^{n-s},$$

where the last equation follows from the definition of $\varepsilon_K$. This completes the proof of the lemma. $\quad\square$

The restricted isometry property plays an important part in our analysis of the overcomplete regime. In the next statement, we present the inequalities that we will use which rely on the RIP assumption. Then as a corollary, we obtain a bound on the remainders $R_{i,s}$ that will be more useful than (S.91).

**Lemma S.32** (Consequences of RIP). *Assume that conditions* **A1**-**A3** *hold. Let* $x \in \mathbb{S}^{D-1}$ *and let* $\mathcal{I} = \mathcal{I}(x) \subseteq [K]$ *satisfy the conclusions of Lemma S.12. Then it holds*

$$\left| F_{\mathcal{A}}(x) - \sum_{i \in \mathcal{I}} \sigma_i \zeta_i^n \right| \leq \sqrt{c_2 \tilde{c}_\delta^n F_{\mathcal{A}}(x) \varepsilon_K}. \tag{S.92}$$

*Further, there exists a constant* $C_1$*, depending only on* $n$*,* $c_1$*,* $c_2$ *and* $\tilde{c}_\delta$*, and another constant* $C_2$*, depending only on* $n$ *and* $\tilde{c}_\delta^n$*, such that we have*

$$\|\zeta\|_{2n}^{2n} \leq F_{\mathcal{A}}(x) + C_1 \sqrt{\varepsilon_K F_{\mathcal{A}}(x)} + C_2 \varepsilon_K. \tag{S.93}$$

*Proof.* To show (S.92), start with the identity $F_{\mathcal{A}}(x) = \sum_{i=1}^{K} \sigma_i \zeta_i^n$, then apply the inner product bound (S.90) with $\xi \in \mathbb{R}^K$ given by $\xi_j = \mathbf{1}(j \notin \mathcal{I}) \zeta_j^n$, and then use Lemma S.12 based on RIP:

$$\left| F_{\mathcal{A}}(x) - \sum_{i \in \mathcal{I}} \sigma_i \zeta_i^n \right| = \left| \sum_{i \notin \mathcal{I}} \sigma_i \zeta_i^n \right|$$

$$\leq \sqrt{c_2 F_{\mathcal{A}}(x)} \sqrt{\sum_{i \notin \mathcal{I}} \zeta_i^{2n}} \qquad \text{(Eq. (S.90))}$$

$$\leq \sqrt{c_2 F_{\mathcal{A}}(x)} \sqrt{K \left( \tilde{c}_\delta^n \frac{\log(K)}{D} \right)^n} \qquad \text{(Lemma S.12)}$$

$$= \sqrt{c_2 \tilde{c}_\delta^n F_{\mathcal{A}}(x) \varepsilon_K}.$$

Next to show (S.93), split $\|\zeta\|_{2n}^{2n}$ into a sum over $\mathcal{I}$ and $[K] \setminus \mathcal{I}$:

$$\|\zeta\|_{2n}^{2n} = \sum_{i \in \mathcal{I}} \zeta_i^{2n} + \sum_{i \notin \mathcal{I}} \zeta_i^{2n}. \tag{S.94}$$

As before, the sum over $[K] \setminus \mathcal{I}$ is upper-bounded using the second conclusion in Lemma S.12:

$$\sum_{i \notin \mathcal{I}} \zeta_i^{2n} \leq \tilde{c}_\delta^n \varepsilon_K. \tag{S.95}$$

As for the sum over $\mathcal{I}$ in (S.94), we can relate this to $F_{\mathcal{A}}(x)$ by using

$$\sum_{i \in \mathcal{I}} \zeta_i^{2n} = \sum_{i \in \mathcal{I}} \sigma_i \zeta_i^n + \sum_{i \in \mathcal{I}} R_{i,n} \zeta_i^n. \tag{S.96}$$

Indeed we have already proven that the first sum on the right-hand side of (S.96) is approximately $F_{\mathcal{A}}(x)$; more precisely (S.92) holds. We can see that the other sum is small by using our initial bound on $R_s$ (S.91) together with the first conclusion of Lemma S.12:

$$\left| \sum_{i \in \mathcal{I}} R_{i,n} \zeta_i^n \right| \leq R_n \sum_{i \in \mathcal{I}} |\zeta_i|^n \leq R_n \sum_{i \in \mathcal{I}} \zeta_i^2 \leq R_n (1 + \delta)$$

$$\leq (1 + \delta) \sqrt{c_2 c_1^n F_{\mathcal{A}}(x) \varepsilon_K} \leq 2 \sqrt{c_2 c_1^n F_{\mathcal{A}}(x) \varepsilon_K}.$$

Putting the last three sentences together via the triangle inequality gives

$$\left| \sum_{i \in \mathcal{I}} \zeta_i^{2n} - F_{\mathcal{A}}(x) \right| \leq \sqrt{c_2 \tilde{c}_\delta^n F_{\mathcal{A}}(x) \varepsilon_K} + 2 \sqrt{c_2 c_1^n F_{\mathcal{A}}(x) \varepsilon_K}.$$

Combining (S.94), (S.95), (S.96), we conclude that we may set

$$C_1 = \tilde{c}_\delta^n \qquad \text{and} \qquad C_2 = \sqrt{c_2 \tilde{c}_\delta^n} + 2 \sqrt{c_2 c_1^n},$$

and (S.93) follows as desired. $\qquad \square$

**Corollary S.33.** *Assume that conditions* **A1-A3** *hold. Let* $x \in \mathbb{S}^{D-1}$ *be arbitrary and* $s \in \{0, \ldots, n\}$. *Then there exists a constant* $\tilde{C}_s$, *depending only on* $s$, $n$, $c_1$, $c_2$ *and* $\tilde{c}_\delta$, *such that*

$$R_s \leq \tilde{C}_s \varepsilon_K^{\frac{s}{2n}} \max\{\varepsilon_K, F_{\mathcal{A}}(x)\}^{1-\frac{s}{2n}}. \tag{S.97}$$

*Proof.* Plugging (S.93) into (S.91) yields

$$R_s \leq \sqrt{c_2 c_1^s F_{\mathcal{A}}(x)} \, \varepsilon_K^{\frac{s}{2n}} \left( F_{\mathcal{A}}(x) + C_1 \sqrt{\varepsilon_K F_{\mathcal{A}}(x)} + C_2 \varepsilon_K \right)^{\frac{n-s}{2n}} \leq \tilde{C}_s \varepsilon_K^{\frac{s}{2n}} \max\{\varepsilon_K, F_{\mathcal{A}}(x)\}^{1-\frac{s}{2n}},$$

where $\tilde{C}_s = \sqrt{c_2 c_1^s} \left(1 + C_1 + C_2\right)^{\frac{n-s}{2n}}$. $\qquad\qquad\qquad\qquad\qquad\qquad\qquad\qquad\qquad\qquad\square$

## H.2   Lower bound on maximum correlation coefficient

We now provide a lower bound on $\|\zeta\|_\infty$ depending on $\frac{\varepsilon_K}{F_{\mathcal{A}}(x)}$.

**Proposition S.34.** *Suppose that* $x$ *is a second-order critical point of* **nSPM-P** *and* $F_{\mathcal{A}}(x) \geq \varepsilon_K$. *Then there exists a constant* $\overline{C}_2$ *depending only on* $n$, $c_1$, $c_2$, $\tilde{c}_\delta$, $\delta$, *and another constant* $\overline{C}_n$ *depending only on* $n, c_1, c_2, \tilde{c}_\delta$, *such that we have*

$$\|\zeta\|_\infty^2 \geq 1 - \frac{2n-1}{2n-2} \frac{\delta}{1+\delta} - \overline{C}_2 \left( \frac{\varepsilon_K}{F_{\mathcal{A}}(x)} \right)^{\frac{1}{2}} - \overline{C}_n \left( \frac{\varepsilon_K}{F_{\mathcal{A}}(x)} \right)^{\frac{1}{n}} - 3 \frac{\Delta_{\mathcal{A}}}{F_{\mathcal{A}}(x)}, \tag{S.98}$$

$$\geq 1 - \frac{2n-1}{2n-2} \frac{\delta}{1+\delta} - (\overline{C}_2 + \overline{C}_n) \left( \frac{\varepsilon_K}{F_{\mathcal{A}}(x)} \right)^{\frac{1}{n}} - 3 \frac{\Delta_{\mathcal{A}}}{F_{\mathcal{A}}(x)}.$$

**Remark S.35.** Note that while the statement is for a constrained second-order critical point of $F_{\hat{\mathcal{A}}}$, (S.98) contains $F_{\mathcal{A}}(x)$. This is intentional: later we use this result together with a bound on $F_{\mathcal{A}}(x)$.

*Proof.* Our starting point is Eq. (S.44) in Proposition S.18, which says that for each $i \in [K]$ we have

$$\left(1 + 2(n-1)\zeta_i^2\right) F_{\mathcal{A}}(x) \geq (2n-1)\sigma_i \zeta_i^{n-2} + n\zeta_i^{n-2} R_{i,n} + (n-1) R_{i,2} - (4n-2)\Delta_{\mathcal{A}}.$$

Multiplying both sides by $\zeta_i^2$ and using that $1 + 2(n-1)\|\zeta\|_\infty^2 \geq 1 + 2(n-1)\zeta_i^2$, we have

$$\zeta_i^2 \left(1 + 2(n-1)\|\zeta\|_\infty^2\right) F_{\mathcal{A}}(x) \geq (2n-1)\sigma_i \zeta_i^n + \zeta_i^2 \left(n\zeta_i^{n-2} R_{i,n} + (n-1)R_{i,2} - (4n-2)\Delta_{\mathcal{A}}\right). \tag{S.99}$$

Now let $\mathcal{I} = \mathcal{I}(x) \subseteq [K]$ satisfy the conclusions of Lemma S.12, and sum the inequalities (S.99) as $i$ ranges over $\mathcal{I}$. Denoting $\|\zeta\|_{2,\mathcal{I}}^2 = \sum_{i \in \mathcal{I}} \zeta_i^2$, this gives

$$\|\zeta\|_{2,\mathcal{I}}^2 \left(1 + 2(n-1)\|\zeta\|_\infty^2\right) F_{\mathcal{A}}(x) \geq (2n-1) \sum_{i \in \mathcal{I}} \sigma_i \zeta_i^n + \sum_{i \in \mathcal{I}} \zeta_i^2 (n\zeta_i^{n-2} R_{i,n} + (n-1)R_{i,2} \\ - (4n-2)\Delta_{\mathcal{A}}). \tag{S.100}$$

We lower-bound each of the sums on the right-hand side in turn. Firstly by (S.92),

$$(2n-1) \sum_{i \in \mathcal{I}} \sigma_i \zeta_i^n \geq (2n-1) \left( F_{\mathcal{A}}(x) - \sqrt{c_2 \tilde{c}_\delta^n F_{\mathcal{A}}(x) \varepsilon_K} \right). \tag{S.101}$$

Secondly by combining the triangle inequality, the trivial bound $|\zeta_i^{n-2}| \leq 1$, Corollary S.33 with $s = 2, n$ and the assumption $F_{\mathcal{A}}(x) \geq \varepsilon_K$, we have

$$\sum_{i \in \mathcal{I}} \zeta_i^2 (n\zeta_i^{n-2} R_{i,n} + (n-1)R_{i,2} - (4n-2)\Delta_{\mathcal{A}})$$

$$\geq -\|\zeta\|_{2,\mathcal{I}}^2 \left(nR_n + (n-1)R_2 + (4n-2)\Delta_{\mathcal{A}}\right)$$

$$\geq -\|\zeta\|_{2,\mathcal{I}}^2 \left(n\tilde{C}_n \sqrt{\varepsilon_K F_{\mathcal{A}}(x)} + (n-1)\tilde{C}_2 \varepsilon_K^{\frac{1}{n}} F_{\mathcal{A}}(x)^{\frac{n-1}{n}} + (4n-2)\Delta_{\mathcal{A}}\right). \tag{S.102}$$

Inserting (S.101) and (S.102) into (S.100) gives

$$\|\zeta\|_{2,\mathcal{I}}^2(1+2(n-1)\|\zeta\|_\infty^2)F_{\mathcal{A}}(x) \geq (2n-1)\left(F_{\mathcal{A}}(x) - \sqrt{c_2\tilde{c}_\delta^n F_{\mathcal{A}}(x)\varepsilon_K}\right)$$

$$- \|\zeta\|_{2,\mathcal{I}}^2\left(n\tilde{C}_n\sqrt{\varepsilon_K F_{\mathcal{A}}(x)} + (n-1)\tilde{C}_2\varepsilon_K^{\frac{1}{n}}F_{\mathcal{A}}(x)^{\frac{n-1}{n}} + (4n-2)\Delta_{\mathcal{A}}\right).$$

Isolating $\|\zeta\|_\infty^2$, and using $1-\delta \leq \|\zeta\|_{2,\mathcal{I}}^2 \leq 1+\delta$ and $\frac{4n-2}{2n-2} \leq 3$, yields (S.98) as desired where

$$\overline{C}_2 = \frac{1}{1-\delta}\frac{\sqrt{c_2\tilde{c}_\delta^n}}{2n-2} + \frac{n}{2n-2}\tilde{C}_n \qquad \text{and} \qquad \overline{C}_n = \frac{1}{2}\tilde{C}_2.$$

This completes the proof of the proposition. $\qquad\square$

## H.3  Concavity

We now show that $F_{\mathcal{A}}$ is strictly geodesically concave in a spherical cap around each $\pm a_i$.

**Proposition S.36.** *There exist constants $C$, $\Delta_0$ and $D_0$, where $C$ and $\Delta_0$ depend only on $n$, $c_2$, and $D_0$ depends additionally on $c_1$, $\tilde{c}_\delta$, such that $C < 1$ and if $\Delta_{\mathcal{A}} < \Delta_0$, and $D \geq D_0$, then for all $i \in [K]$ and $s \in \{-1,1\}$, it holds*

1. *$F_{\hat{\mathcal{A}}}$ is strictly concave on the spherical cap $B_{r_+}(sa_i)$, where*

$$r_+ := 1 - \left(1 - C\left(0.99 - 4\Delta_{\mathcal{A}}\right)^2\right)^{\frac{1}{2n}}.$$

2. *There exists exactly one local maximum in each spherical cap $B_{r_+}(sa_i)$, and denoting it by $x_*$, we have*

$$\|x_* - sa_i\|^2 \leq \frac{2\Delta_{\mathcal{A}}}{n}. \tag{S.103}$$

*Proof.* We first note that Lemma S.32 implies that, for all $y \in \mathbb{S}^{D-1}$,

$$\sum_{i=1}^K \langle y, a_i\rangle^{2n} = \|\zeta(y)\|_{2n}^{2n} \leq F_{\mathcal{A}}(y) + C\sqrt{\varepsilon_K \max\{\varepsilon_K, F_{\mathcal{A}}(y)\}} \leq 1 + O(\sqrt{\varepsilon_K}),$$

thus since $\lim_{D\to\infty}\varepsilon_K = 0$, for all $\mu \in (0,1)$ there exist $D_0$ such that if $D \geq D_0$,

$$\sum_{i=1}^K \langle y, a_i\rangle^{2n} \leq 1 + \mu, \quad \text{uniformly for all } y \in \mathbb{S}^{D-1}. \tag{S.104}$$

Let $i \in [K]$, $s \in \{-1,1\}$ and $y \perp x$ arbitrary and recall (S.49),

$$\frac{1}{2n}y^\top \nabla_{\mathbb{S}^{D-1}}^2 F_{\hat{\mathcal{A}}}(x)y \leq n\left\|P_{\mathcal{A}}(x^{n-1}y)\right\|^2 + (n-1)\langle P_{\mathcal{A}}(x^n), x^{n-2}y^2\rangle - \zeta_i^{2n} + 4\Delta_{\mathcal{A}}. \tag{S.105}$$

The second term can be bounded using Lemma S.16.

$$\left|\langle P_{\mathcal{A}}(x^n), x^{n-2}y^2\rangle\right| \leq \left|\langle x^n, x^{n-2}y^2\rangle\right| + \left|\langle P_{\mathcal{A}}(x^n) - x^n, x^{n-2}y^2\rangle\right|$$
$$\leq 0 + \|P_{\mathcal{A}^\perp}(x^n)\|_F\,\|\operatorname{Sym}(x^{n-2}y^2)\|$$
$$= \frac{\sqrt{2}}{\sqrt{n(n-1)}}\sqrt{1 - \|P_{\mathcal{A}}(x^n)\|_F^2} \leq \frac{2}{n-1}\sqrt{1 - \langle a_i, x\rangle^{2n}}.$$

The first term in (S.105) is slightly more involved. We first use Lemma S.5 and **A3** to write

$$\left\|P_{\mathcal{A}}(x^{n-1}y)\right\|_F^2 = \beta(x^{n-1}y)^\top G_n^{-1}\beta(x^{n-1}y) \leq c_2\left\|\beta(x^{n-1}y)\right\|_2^2.$$

Then, by leveraging the fact that $y \perp x$, we further get

$$\left\|\beta(x^{n-1}, y)\right\|_2^2 = \sum_{j=1}^K \langle a_j, x\rangle^{(2n-2)}\langle a_j, y\rangle^2 \leq \langle a_i, x\rangle^{2n-2}\langle a_i, y\rangle^2 + \sum_{j \neq i}^K \langle a_j, x\rangle^{(2n-2)}\langle a_j, y\rangle^2$$

$$\leq \langle a_i, x\rangle^{2n-2}(\|a_i\|^2 - \langle a_i, x\rangle^2) + \sum_{j \neq i}^K \langle a_j, x\rangle^{(2n-2)}\langle a_j, y\rangle^2$$

$$= \langle a_i, x\rangle^{2n-2} - \langle a_i, x\rangle^{2n} + \sum_{j \neq i}^K \langle a_j, x\rangle^{(2n-2)}\langle a_j, y\rangle^2,$$

where $\langle a_i, y\rangle^2 + \langle a_i, x\rangle^2 \leq \|a_i\|^2$ is a consequence of Bessel's inequality. On one hand, since the function $f(x) = (1-x)^{\frac{n-1}{n}} - (1-x)$ is concave in $[0,1]$, and $f'(0) = \frac{1}{n}$, we have $f(x) \leq \frac{x}{n}$, which implies $\langle a_i, x\rangle^{2n-2}(1 - \langle a_i, x\rangle^2) \leq \frac{1-\langle a_i, x\rangle^{2n}}{n}$. On the other hand, using Hölder's inequality with $p = n$ and $q = \frac{n}{n-1}$ satisfying $1/n + (n-1)/n = 1$, and Equation (S.104), we get

$$\sum_{j \neq i}^K \langle a_j, x\rangle^{(2n-2)}\langle a_j, y\rangle^2 \leq \left(\sum_{j \neq i}^K \langle a_j, x\rangle^{2n}\right)^{\frac{n-1}{n}} \left(\sum_{j \neq i}\langle a_j, y\rangle^{2n}\right)^{\frac{1}{n}}$$

$$\leq (1+\mu)^{\frac{1}{n}}\left(1 + \mu - \langle a_i, x\rangle^{2n}\right)^{\frac{n-1}{n}}$$

$$\leq \left(1 + \frac{\mu}{n}\right)\left(1 + \mu - \langle a_i, x\rangle^{2n}\right)^{\frac{n-1}{n}}.$$

Combining the estimates and writing $\eta := 1 - \langle a_i, x\rangle^{2n}$ for short, we obtain

$$\frac{1}{2n}y^\top \nabla^2_{\mathbb{S}^{D-1}} F_{\hat{\mathcal{A}}}(x)y \leq \left\|G_n^{-1}\right\|_2 \left(\eta + (n+\mu)(\eta+\mu)^{\frac{n-1}{n}}\right) + (n-1)\sqrt{\eta} - 1 + \eta + 4\Delta_{\mathcal{A}}$$

$$\leq \left\|G_n^{-1}\right\|_2 \left(\eta + (n+\mu)(\eta^{\frac{n-1}{n}} + \mu^{\frac{n-1}{n}})\right) + (n-1)\sqrt{\eta} - 1 + \eta + 4\Delta_{\mathcal{A}}$$

$$\leq \left\|G_n^{-1}\right\|_2 \left((1+n+\mu)\sqrt{\eta} + (n+\mu)\mu^{\frac{n-1}{n}}\right) + n\sqrt{\eta} + 4\Delta_{\mathcal{A}} - 1$$

$$\leq (n + (n+\mu+1)c_2)\sqrt{\eta} + (n+\mu)c_2\mu^{\frac{n-1}{n}} + 4\Delta_{\mathcal{A}} - 1.$$

Now choosing $\mu$ (and consequently $D_0$) such that $(n+\mu)c_2\mu^{\frac{n-1}{n}} \leq 0.01$, noticing that $\mu c_2 \leq (n+\mu)c_2\mu^{\frac{n-1}{n}}$, and setting $C = \left(\frac{1}{n+0.01+(n+1)c_2}\right)^2$, we get

$$\frac{1}{2n}y^\top \nabla^2_{\mathbb{S}^{D-1}} F_{\hat{\mathcal{A}}}(x)y \leq \sqrt{\frac{\eta}{C}} + 4\Delta_{\mathcal{A}} - 0.99, \tag{S.106}$$

and thus, by rearranging terms, we get that $F_{\hat{\mathcal{A}}}$ is strictly geodesically concave in $B_{r_+}sa_i$.

We now define

$$\tilde{r}_+ := 1 - \left(1 - C(0.49 - 4\Delta_{\mathcal{A}})^2\right)^{\frac{1}{2n}},$$

and notice that for all $x \in B_{\tilde{r}_+}(sa_i)$ and $y \in \mathbb{S}^{D-1}$, (S.106) implies that $y^\top \nabla^2_{\mathbb{S}^{D-1}} F_{\hat{\mathcal{A}}}(x)y \leq -n$. We also have

$$1 - \sqrt[2n]{1 - C(0.49 - 4\Delta_{\mathcal{A}})^2} > 1 - \left(1 - \frac{C}{2n}(0.49 - 4\Delta_{\mathcal{A}})^2\right) = 0.49^2\frac{C}{2n}\left(1 - \frac{4}{0.49}\Delta_{\mathcal{A}}\right)^2$$

$$\geq 0.49^2\frac{C}{2n}\left(1 - \frac{8}{0.49}\Delta_{\mathcal{A}}\right) > \frac{\Delta_{\mathcal{A}}}{n},$$

where the last equality follows if we choose $\Delta_0 = 0.49^2\frac{C}{2n}/\left(\frac{1}{n} + 3.92\frac{C}{2n}\right)$, since $\Delta_{\mathcal{A}} < \Delta_0$. Therefore, applying Proposition S.24, we obtain that it exists exactly one local maximum $x_* \in B_{\tilde{r}_+}(sa_i)$, and $x_*$ satisfies (S.103).

Finally, since $F_{\hat{\mathcal{A}}}$ is strictly geodesically concave on $B_{r_+}(sa_i)$, Proposition S.24 implies that there exists at most one local maximum in $B_{r_+}(sa_i)$, thus since $B_{\tilde{r}_+}(sa_i) \subset B_{r_+}(sa_i)$, $x_*$ is also the only local maximum of $B_{r_+}(sa_i)$. $\qquad\square$

## H.4 Completing the proof

We now have all the pieces to conclude the argument for Theorem M.16.

*Proof of Theorem M.16.* Suppose that $x$ is a local maximum and $F_{\hat{\mathcal{A}}}(x) \geq C\varepsilon_K + 5\Delta_{\mathcal{A}}$, where $C$ will be defined below. Our proof strategy is to show that this implies that $x$ is in one of the $2K$ spherical caps defined in Proposition S.36. The rest of the proof then follows from Proposition S.36, since each spherical cap contains exactly one local maximum.

Denote $\tilde{C} = \overline{C}_2 + \overline{C}_n$ from the statement of Proposition S.34, and define $C$ so that $C \geq 1$. Therefore $F_{\hat{\mathcal{A}}}(x) \geq \varepsilon_K$ and Proposition S.34 implies

$$\|\zeta\|_{\infty}^2 \geq 1 - \frac{2n-1}{2n-2}\frac{\delta}{1+\delta} - \tilde{C}\left(\frac{\varepsilon_K}{F_{\mathcal{A}}(x)}\right)^{\frac{1}{n}} - 3\frac{\Delta_{\mathcal{A}}}{F_{\mathcal{A}}(x)}.$$

By the AM-GM inequality, we have

$$\tilde{C}\left(\frac{\varepsilon_K}{F_{\mathcal{A}}(x)}\right)^{\frac{1}{n}} \leq \frac{1}{n}\left(4^{n-1}\tilde{C}^n\frac{\varepsilon_K}{F_{\mathcal{A}}(x)} + \frac{1}{4}(n-1)\right),$$

thus

$$\tilde{C}\left(\frac{\varepsilon_K}{F_{\mathcal{A}}(x)}\right)^{\frac{1}{n}} + 3\frac{\Delta_{\mathcal{A}}}{F_{\mathcal{A}}(x)} \leq \frac{1}{F_{\mathcal{A}}(x)}\left(\frac{4^{n-1}}{n}\tilde{C}^n\varepsilon_K + 3\Delta_{\mathcal{A}}\right) + \frac{1}{4} - \frac{1}{4n}.$$

We now define $C := \frac{1}{3}4^n\tilde{C}^n$, thus having

$$F_{\mathcal{A}}(x) \geq F_{\hat{\mathcal{A}}}(x) - \Delta_{\mathcal{A}} \geq C\varepsilon_K + 4\Delta_{\mathcal{A}} \geq \frac{4}{3}\left(\frac{4^{n-1}}{n}\tilde{C}^n\varepsilon_K + 3\Delta_{\mathcal{A}}\right),$$

and

$$\tilde{C}\left(\frac{\varepsilon_K}{F_{\mathcal{A}}(x)}\right)^{\frac{1}{n}} + 3\frac{\Delta_{\mathcal{A}}}{F_{\mathcal{A}}(x)} \leq \frac{3}{4} + \frac{1}{4} - \frac{1}{4n} \leq 1 - \frac{1}{4n}.$$

If we now choose $\delta_0$ such that $\frac{2n-1}{2n-2}\frac{\delta_0}{1+\delta_0} \leq \frac{1}{8n}$, we have that if $\delta < \delta_0$, then

$$\|\zeta\|_{\infty}^2 \geq 1 - \frac{2n-1}{2n-2}\frac{\delta}{1+\delta} - \left(1 - \frac{1}{4n}\right) \geq \frac{1}{8n}. \tag{S.107}$$

Note that we always have $F_{\mathcal{A}}(x) \geq \|\zeta\|_{\infty}^{2n}$. To show this, define $i$ such that $\|\zeta\|_{\infty} = |\zeta_i|$. Then $F_{\mathcal{A}}(x) = \|P_{\mathcal{A}}(x^n)\|^2 \geq \|P_{a_i^n}(x^n)\|^2 = \zeta_i^{2n} = \|\zeta\|_{\infty}^{2n}$. Therefore, (S.107) implies that $F_{\mathcal{A}}(x) \geq \left(\frac{1}{8n}\right)^n$, and since $\lim_{D\to\infty}\varepsilon_K = 0$, for any $\nu \in (0,1)$, there exists $D_0$ such that if $D \geq D_0$, then

$$\tilde{C}\left(\frac{\varepsilon_K}{F_{\mathcal{A}}(x)}\right)^{\frac{1}{n}} \leq 8n\tilde{C}\varepsilon_K^{\frac{1}{n}} \leq \frac{1}{2}\nu.$$

Furthermore, choosing $\delta_0$ such that $\frac{2n-1}{2n-2}\frac{\delta_0}{1+\delta_0} \leq \min\left\{\frac{1}{8n}, \frac{1}{2}\nu\right\}$, we obtain, by applying Proposition S.34 again, that for all $\nu \in (0,1)$ there exists $D_0$ and $\delta_0$ such that, if $D \leq D_0$ and $\delta \leq \delta_0$,

$$\|\zeta\|_{\infty}^2 \geq 1 - \nu - 3\frac{\Delta_{\mathcal{A}}}{F_{\mathcal{A}}(x)}. \tag{S.108}$$

Simply to improve the implicit constant, we can choose $\nu = 0.01$, and notice that $F_{\mathcal{A}}(x) \geq 4\Delta_{\mathcal{A}}$, to get $\|\zeta\|_{\infty}^2 \geq 0.96/4$, and consequently, $F_{\mathcal{A}}(x) \geq (0.96/4)^n$. Plugging this in (S.108), we obtain

$$\|\zeta\|_{\infty}^2 \geq 1 - \nu - 3(4/0.96)^n\Delta_{\mathcal{A}}.$$

We now further choose $\Delta_0$ and a smaller $\nu$ such that, if $\Delta_{\mathcal{A}} < \Delta_0$, then

$$\sqrt[n]{1 - C(0.99 - 4\Delta_{\mathcal{A}})^2} \leq 1 - \frac{C}{n}(0.99 - 4\Delta_{\mathcal{A}})^2$$

$$\leq 1 - \frac{C}{n}(0.98 - 7.92\Delta_{\mathcal{A}})$$

$$\leq 1 - \nu - 3(4/0.96)^n\Delta_{\mathcal{A}} \leq \|\zeta\|_{\infty}^2. \tag{S.109}$$

This then implies that $x$ must lie in one of the caps defined in Proposition S.36, and the rest of the result follows from Proposition S.36. For (S.109) to hold, it suffices to choose $\nu = \min(0.49\frac{C}{n}, 0.01)$ and $\Delta_0 = 0.49\frac{C}{n}/\left(3(4/0.96)^n + 7.92\frac{C}{n}\right)$. $\qquad\square$

# I Deflation bounds and proof of Theorem M.18

Suppose we want to decompose the tensor $\hat{T}$, which is an estimate of the low rank tensor $T = \sum_{i=1}^{K} \lambda_i a_i^{\otimes m}$. We will consider the following slight modification of the SPM algorithm [8].

---
**Algorithm 1** SPM with modified deflation step

---
**Input:** $\hat{T} \in \mathcal{T}_D^m$
**Hyper-parameters:** $\alpha, \tau \in \mathbb{R}^+$
**Returns:** $(\hat{a}_k, \hat{\lambda}_k)_{k \in [K]}$
  $\hat{M} \leftarrow \text{Reshape}(\hat{T}, [D^n, D^{m-n}])$.
  Set $K$ as the number of singular values of $\hat{M}$ exceeding $\alpha$.
  Set $\hat{M}_K$ as the rank-$K$ truncation of the SVD of $\hat{M}$.
  Let $\hat{\mathcal{A}} = \text{colspan}(\hat{U})$ and $\hat{\mathcal{A}}_1 = \hat{\mathcal{A}}$
  **for** $k = 1, \ldots, K$ **do**
    Obtain $\tilde{a}_k$ by applying POWER METHOD [8] on functional $F_{\hat{\mathcal{A}}_k}$ until convergence.
    Repeat last step with new initializations until $F_{\hat{\mathcal{A}}_k}(\tilde{a}_k) \geq \tau$.
    **if** $k = 1$ **then** $\hat{a}_1 \leftarrow \tilde{a}_1$
    **else** Obtain $\hat{a}_k$ by applying POWER METHOD on $F_{\hat{\mathcal{A}}}$ with $\tilde{a}_k$ as starting point until convergence.

$$\hat{\lambda}_k \leftarrow \frac{1}{\text{vec}(\hat{a}_k^{\otimes n})^\top (\hat{M}_K^\top)^\dagger \text{vec}(\hat{a}_k^{\otimes m-n})} \tag{S.110}$$

    **if** $k < K$ **then**
      $\hat{\mathcal{A}}_{k+1} \leftarrow \hat{\mathcal{A}}_k \cap ((\hat{M}_K^\top)^\dagger \text{vec}(\hat{a}_k^{\otimes m-n}))^\perp$

---

We analyze Algorithm 1 assuming the conditions of Theorems M.7 or M.16. The conclusion will be Theorem M.18. We note the sign and permutation ambiguity there are inherent to the CP decomposition.

**Theorem S.37** (= Theorem M.18). *Let* $T = \sum_{i=1}^{K} \lambda_i a_i^{\otimes m} \in \text{Sym}(\mathcal{T}_D^m)$, $M := \text{Reshape}(T, [D^n, D^{m-n}])$ *and* $\sigma_1(M) \geq \ldots \geq \sigma_K(M) > 0$ *be the singular values of* $M$. *Define* $\alpha, \tau, \hat{M}$ *as in Algorithm 1, assume* $\Delta_M := \|M - \hat{M}\|_2 < \frac{1}{2}\sigma_K(M)$ *and let* $\hat{\Delta}_{\mathcal{A}} = \frac{\Delta_M}{\sigma_K(M) - \Delta_M}$. *Suppose that* $T$ *verifies the assumptions of either Theorem M.7 or Theorem M.16, define* $\Delta_0$ *as in the corresponding theorem statement and let* $\ell(\Delta_{\mathcal{A}})$ *be the corresponding level set threshold, which in both statements depends on* $\Delta_{\mathcal{A}}$. *Then there exists constants* $C_1, C_2$, *not depending on* $\hat{T}$ *or* $\Delta_M$ *(see (S.117) - (S.118) for definitions of these two contants), such that if* $\Delta_M < \alpha < \sigma_K(M) - \Delta_M$, $\tilde{\Delta}_{\mathcal{A}} := C_1 \hat{\Delta}_{\mathcal{A}} + C_2 \sqrt{\hat{\Delta}_{\mathcal{A}}} < \Delta_0$ *and* $\tau > \ell(\tilde{\Delta}_{\mathcal{A}})$, *there exist a permutation* $\pi \in S^K$ *and unit scalars* $s_1, \ldots, s_K \in \{-1, 1\}$ *such that Algorithm 1 returns* $(\hat{a}_k, \hat{\lambda}_k)_{k \in [K]}$ *with error bounded by*

$$\|s_k a_{\pi(k)} - \hat{a}_k\| \leq \sqrt{\frac{2\hat{\Delta}_{\mathcal{A}}}{n}} \quad and \quad \left| \frac{s_k^m}{\lambda_{\pi(k)}} - \frac{1}{\hat{\lambda}_k} \right| \leq \frac{2\sqrt{m/n}}{\sigma_K(M)} \sqrt{\hat{\Delta}_{\mathcal{A}}} + \frac{4}{\sigma_K(M)} \hat{\Delta}_{\mathcal{A}} \tag{S.111}$$

*In particular, in the noiseless case* ($\Delta_M = 0$), *Algorithm 1 returns the exact CP decomposition of* $T$, *up to permutation and sign ambiguity.*

A result on the stability of matrix pseudo-inversion is needed; we include a proof for completeness.

**Lemma S.38.** *Let* $W, \hat{W} \in \mathbb{R}^{p \times q}$ *and suppose that* $W$ *has exactly* $r$ *nonzero singular values* $\sigma_1(W) \geq \ldots \geq \sigma_r(W) > 0$ *and* $\Delta_W := \|W - \hat{W}\|_2 < \sigma_r(W)$. *Define* $\hat{W}_r$ *as the SVD of* $\hat{W}$ *truncated at the* $r$-*th singular value. Then*

$$\|W^\dagger - \hat{W}_r^\dagger\|_2 \leq \frac{\Delta_W}{\sigma_r(W) - \Delta_W} \left( \frac{2}{\sigma_r(W)} + \frac{1}{\sigma_r(W) - \Delta_W} \right).$$

*Proof.* We denote the singular value decomposition of matrices $W, \hat{W} \in \mathbb{R}^{p \times q}$ as

$$W = \begin{pmatrix} U_1 & U_2 \end{pmatrix} \begin{pmatrix} \Sigma_1 & 0 \\ 0 & \Sigma_2 \end{pmatrix} \begin{pmatrix} V_1^\top \\ V_2^\top \end{pmatrix} \quad and \quad \hat{W} = \begin{pmatrix} \hat{U}_1 & \hat{U}_2 \end{pmatrix} \begin{pmatrix} \hat{\Sigma}_1 & 0 \\ 0 & \hat{\Sigma}_2 \end{pmatrix} \begin{pmatrix} \hat{V}_1^\top \\ \hat{V}_2^\top \end{pmatrix},$$

where $\Sigma_1$ and $\hat\Sigma_1$ are diagonal matrices with the largest $r$ singular values of $W$ and $\hat W$ on the diagonal, respectively. We have $\|W^\dagger - \hat W_r^\dagger\|_2 = \left\|U_1\Sigma_1^{-1}V_1^\top - \hat U_1\hat\Sigma_1^{-1}\hat V_1^\top\right\|_2$. Then

$$U_1\Sigma_1^{-1}V_1^\top - \hat U_1\hat\Sigma_1^{-1}\hat V_1^\top = (U_1U_1^\top - \hat U_1\hat U_1^\top)U_1\Sigma_1^{-1}V_1^\top + \hat U_1\hat U_1^\top(U_1\Sigma_1^{-1}V_1^\top - \hat U_1\hat\Sigma_1^{-1}\hat V_1^\top)$$
$$= (U_1U_1^\top - \hat U_1\hat U_1^\top)U_1\Sigma_1^{-1}V_1^\top + \hat U_1\hat\Sigma_1^{-1}\hat V_1^\top(\hat W^\top U_1\Sigma_1^{-1}V_1^\top - \hat V_1\hat V_1^\top),$$

where we used $U_1\hat U_1^\top\hat U_1\hat\Sigma_1^{-1}\hat V_1^\top = \hat U_1\hat\Sigma_1^{-1}\hat V_1^\top = \hat U_1\hat\Sigma_1^{-1}\hat V_1^\top\hat V_1\hat V_1^\top$ and

$$\hat U_1\hat\Sigma_1^{-1}\hat V_1^\top W^\top = \hat U_1\hat\Sigma_1^{-1}\hat V_1^\top(\hat V_1\hat\Sigma_1\hat U_1^\top + \hat V_2\hat\Sigma_2\hat U_2^\top) = \hat U_1\hat U_1^\top.$$

Furthermore,

$$\hat W^\top U_1\Sigma_1^{-1}V_1^\top - \hat V_1\hat V_1^\top = \hat W^\top U_1\Sigma_1^{-1}V_1^\top - V_1V_1^\top + (V_1V_1^\top - \hat V_1\hat V_1^\top)$$
$$= (\hat W^\top - V_1\Sigma_1 U_1^\top)U_1\Sigma_1^{-1}V_1^\top + (V_1V_1^\top - \hat V_1\hat V_1^\top)$$
$$= (\hat W^\top - W^\top)U_1\Sigma_1^{-1}V_1^\top + (V_1V_1^\top - \hat V_1\hat V_1^\top).$$

We then have

$$\|W^\dagger - \hat W_r^\dagger\|_2 = \left\|U_1\Sigma_1^{-1}V_1^\top - \hat U_1\hat\Sigma_1^{-1}\hat V_1^\top\right\|_2$$
$$= \left\|(U_1U_1^\top - \hat U_1\hat U_1^\top)U_1\Sigma_1^{-1}V_1^\top + \hat U_1\hat\Sigma_1^{-1}\hat V_1^\top(\hat W^\top - W^\top)U_1\Sigma_1^{-1}V_1^\top\right.$$
$$\left.+ \hat U_1\hat\Sigma_1^{-1}\hat V_1^\top(V_1V_1^\top - \hat V_1\hat V_1^\top)\right\|_2$$
$$\le \left\|U_1U_1^\top - \hat U_1\hat U_1^\top\right\|_2\left\|U_1\Sigma_1^{-1}V_1^\top\right\|_2 + \left\|\hat U_1\hat\Sigma_1^{-1}\hat V_1^\top\right\|_2\left\|\hat W^\top - W^\top\right\|_2\left\|U_1\Sigma_1^{-1}V_1^\top\right\|_2$$
$$+ \left\|\hat U_1\hat\Sigma_1^{-1}\hat V_1^\top\right\|_2\left\|V_1V_1^\top - \hat V_1\hat V_1^\top\right\|_2$$
$$\le \frac{\Delta_W}{\sigma_r(W) - \Delta_W}\left(\frac{2}{\sigma_r(W)} + \frac{1}{\sigma_r(W) - \Delta_W}\right).$$

The last line uses Wedin's bound and $\|\hat U_1\hat\Sigma_1^{-1}\hat V_1^\top\|_2 = \frac{1}{\sigma_r(\hat W)} \le \frac{1}{\sigma_r(W) - \Delta_W}$, which comes from Weyl's inequality. $\qquad\square$

Next we control the propagation of error on the weights and intermediate subspaces in Algorithm 1.

**Proposition S.39.** *Define $T, \hat T$ as in Algorithm 1, with $M := \mathrm{Reshape}(T, [D^n, D^{m-n}])$ and its singular values $\sigma_1(M) \ge \ldots \ge \sigma_K(M) > 0$. Let $\hat M := \mathrm{Reshape}(\hat T, [D^n, D^{m-n}])$, and suppose that $\Delta_M := \|M - \hat M\|_2 < \sigma_K(M)$. Then the error for $\hat\lambda_k$ obtained by Algorithm 1 is bounded as follows:*

$$\left|\frac{1}{\lambda_k} - \frac{1}{\hat\lambda_k}\right| \le \frac{1}{\sigma_K(M)}\sqrt{2m}\|a_k - \hat a_k\| + \frac{\Delta_M}{\sigma_K(M) - \Delta_M}\left(\frac{2}{\sigma_K(M)} + \frac{1}{\sigma_K(M) - \Delta_M}\right). \quad \text{(S.112)}$$

*Additionally, let $\mathcal{A}_k = \mathrm{span}\{a_i^{\otimes n}, i \in \{k, \ldots, K\}\}$, $A_{[k-1]}$ the submatrix of $A$ with columns in $[k-1]$, $G_{d,[k-1]} = (A_{[k-1]}^{\bullet s})^\top(A_{[k-1]}^{\bullet k}) = (A_{[k-1]}^\top A_{[k-1]})^{\odot k}$ and $\varphi_{d,k-1} = \sqrt{\|G_{d,[k-1]}\|_2}$. Then the error of the deflated subspace is bounded by*

$$\left\|P_{\mathcal{A}_k} - P_{\hat{\mathcal{A}}_k}\right\|_2 \le \frac{\Delta_M}{\sigma_K(M) - \Delta_M}\left(1 + \max_{i\in[K]}|\lambda_i|\sqrt{\|G_n\|_2\|G_{m-n}\|_2}\left(\frac{2}{\sigma_K(M)} + \frac{1}{\sigma_K(M) - \Delta_M}\right)\right)$$

$$+ \max_{i\in[K]}|\lambda_i|\sqrt{\|G_n\|_2}\frac{1}{\sigma_K(M) - \Delta_M}\sqrt{(m-n)\sum_{i=1}^{k-1}\|a_s - \hat a_s\|^2}.$$

$$\text{(S.113)}$$

*Proof.* We start by showing that (S.110) holds in the clean case, that is,

$$\lambda_k = \frac{1}{\mathrm{vec}(a_k^{\otimes n})^\top(M^\top)^\dagger\,\mathrm{vec}(a_k^{\otimes m-n})}.$$

We have

$$M = \sum_{i=1}^{K} \lambda_i \operatorname{vec}(a_i^{\otimes n}) \operatorname{vec}(a_i^{\otimes m-n})^\top = A^{\bullet n} \Lambda (A^{\bullet m-n})^\top,$$

where $\Lambda = \operatorname{diag}(\lambda_1, \ldots, \lambda_K)$. Therefore, $(M^\top)^\dagger = ((A^{\bullet n})^\top)^\dagger \Lambda^{-1} (A^{\bullet m-n})^\dagger$ and

$$(A^{\bullet n})^\top (M^\top)^\dagger A^{\bullet m-n} = \Lambda^{-1}. \tag{S.114}$$

Denoting by $e_k$ the $k$-th canonical basis vector, we have $\operatorname{vec}(a_k^{\otimes n}) = (A^{\bullet n}) e_k$, thus

$$\operatorname{vec}(a_k^{\otimes n})^\top (M^\top)^\dagger \operatorname{vec}(a_k^{\otimes m-n}) = e_k^\top \left( (A^{\bullet n})^\top (M^\top)^\dagger A^{\bullet m-n} \right) e_k = e_k^\top \Lambda^{-1} e_k = \frac{1}{\lambda_k}$$

Then

$$\left| \frac{1}{\hat{\lambda}_k} - \frac{1}{\lambda_k} \right| = \left| \operatorname{vec}(\hat{a}_k^{\otimes n})^\top (\hat{M}_K^\top)^\dagger \operatorname{vec}(\hat{a}_k^{\otimes m-n}) - \operatorname{vec}(a_k^{\otimes n})^\top (M^\top)^\dagger \operatorname{vec}(a_k^{\otimes m-n}) \right|$$

$$\leq \left| \operatorname{vec}(\hat{a}_k^{\otimes n})^\top ((\hat{M}_K^\top)^\dagger - (M^\top)^\dagger) \operatorname{vec}(\hat{a}_k^{\otimes m-n}) \right|$$

$$+ \left| \operatorname{vec}(\hat{a}_k^{\otimes n})^\top (M^\top)^\dagger (\operatorname{vec}(\hat{a}_k^{\otimes m-n}) - \operatorname{vec}(a_k^{\otimes m-n})) \right|$$

$$+ \left| (\operatorname{vec}(\hat{a}_k^{\otimes n}) - \operatorname{vec}(a_k^{\otimes n}))^\top (M^\top)^\dagger \operatorname{vec}(a_k^{\otimes m-n}) \right|$$

$$\leq \left\| \hat{M}_K^\dagger - M^\dagger \right\|_2 + \|M^\dagger\|_2 \left( \|a_i^{\otimes n} - \hat{a}_i^{\otimes n}\|_F + \|a_i^{\otimes m-n} - \hat{a}_i^{\otimes m-n}\|_F \right)$$

The first term is bounded using Lemma S.38, while the bound for the second term follows from $\|M^\dagger\|_2 = \frac{1}{\sigma_k(M)}$ and

$$\|x^{\otimes d} - y^{\otimes d}\|^2 = 2 - 2(x^\top y)^d = 2 - 2\left( 1 - \frac{1}{2} \|x - y\|^2 \right)^d$$

$$\leq 2 - 2 + d\|x - y\|^2 = d\|x - y\|^2, \tag{S.115}$$

where we used $(1 - x)^d \geq 1 - dx$ for all $x \leq 1$ and $d \geq 1$. Using AM – GM, we obtain

$$\|a_i^{\otimes n} - \hat{a}_i^{\otimes n}\|_F + \|a_i^{\otimes m-n} - \hat{a}_i^{\otimes m-n}\|_F \leq \left( \sqrt{m-n} + \sqrt{n} \right) \|a_i - \hat{a}_i\| \leq \sqrt{2m} \|a_i - \hat{a}_i\|,$$

and the proof of (S.110) is complete. Regarding, (S.113), we have that

$$\hat{\mathcal{A}}_k = \hat{\mathcal{A}} \cap \bigcap_{i=1}^{k-1} ((\hat{M}_K^\top)^\dagger \operatorname{vec}(\hat{a}_i^{\otimes m-n}))^\perp = \hat{\mathcal{A}} \cap \operatorname{colspan}((\hat{M}_K^\top)^\dagger \hat{A}_{[k-1]}^{\bullet m-n})^\perp$$

and

$$\hat{\mathcal{A}} = \operatorname{colspan}(\hat{M}_K) = \operatorname{colspan}((\hat{M}_K^\top)^\dagger) \supseteq \operatorname{colspan}((\hat{M}_K^\top)^\dagger \hat{A}_{[k-1]}^{\bullet m-n}). \tag{S.116}$$

We first show that if $\hat{T} = T$, $\hat{\mathcal{A}}_k = \mathcal{A}_k$, that is,

$$\mathcal{A} \cap \operatorname{colspan}((M^\top)^\dagger A_{[k-1]}^{\bullet m-n})^\perp = \operatorname{span}\{a_i^{\otimes n}, i \in \{k, \ldots, K\}\}.$$

It is enough to show that $\dim(\hat{\mathcal{A}}_k) = K - k + 1 = \dim(\mathcal{A}_k)$ and that $\mathcal{A}_k \subset \hat{\mathcal{A}}_k$. We have that $\dim(\hat{\mathcal{A}}) = K$, $\dim(\operatorname{colspan}((\hat{M}_K^\top)^\dagger A_{[k-1]}^{\bullet m-n})) = k - 1$, thus (S.116) implies that $\dim(\hat{\mathcal{A}}_k) = K - k + 1$. To show that $\hat{\mathcal{A}}_k \subset \mathcal{A} \cap \operatorname{colspan}((M^\top)^\dagger A_{[k-1]}^{\bullet n})^\perp$, note that,

$$\mathcal{A}_k = \operatorname{span}\{a_i^{\otimes n}, i \in \{k, \ldots, K\}\} = \operatorname{colspan}(A_{[k:K]}^{\bullet n}) \subset \operatorname{colspan}(A^{\bullet m-n}) = \mathcal{A},$$

where $[k : K] = \{k, \ldots, K\}$. On other hand, $\operatorname{colspan}(A_{[k:K]}^{\bullet n}) \subset ((M^\top)^\dagger A_{[k-1]}^{\bullet m-n})^\perp$ is equivalent to, for all $i \in [k : K], j \in [k-1]$, $\operatorname{vec}(a_i^{\otimes n})^\top (M^\top)^\dagger \operatorname{vec}(a_k^{\otimes m-n}) = 0$. In fact, (S.114) implies that, $\operatorname{vec}(a_i^{\otimes n})^\top (M^\top)^\dagger \operatorname{vec}(a_k^{\otimes m-n}) = (\Lambda^{-1})_{ij} = 0$ for all $i \in [k : K], j \in [k-1]$, as we wanted to show.

Let $U = (U_1, U_2), \hat{U} = (\hat{U}_1, \hat{U}_2)$ be orthogonal matrices such that the columns of $U_1$ and $\hat{U}_1$ are orthonormal basis for $\mathcal{A}$ and $\hat{\mathcal{A}}$, respectively. Moreover, since $\operatorname{colspan}((\hat{M}_K^\top)^\dagger \hat{A}_{[k-1]}^{\bullet m-n}) \subset \hat{\mathcal{A}}$,

there exist orthogonal matrices $O = (O_1, O_2)$ and $\hat{O} = (\hat{O}_1, \hat{O}_2)$ such that $\mathrm{colspan}(U_1 O_1) = \mathrm{colspan}((M^\top)^\dagger A^{\bullet m-n}_{[k-1]})$ and $\mathrm{colspan}(\hat{U}_1 \hat{O}_1) = \mathrm{colspan}((\hat{M}_K^\top)^\dagger \hat{A}^{\bullet m-n}_{[k-1]})$, respectively. We then have $\mathcal{A}_k = \mathrm{colspan}(U_1 O_2)$ $\hat{\mathcal{A}}_k = \mathrm{colspan}(\hat{U}_1 \hat{O}_2)$, thus:

$$
\begin{aligned}
\left\| P_{\mathcal{A}_k} - P_{\hat{\mathcal{A}}_k} \right\|_2 &= \left\| U_1 O_2 O_2^\top U_1^\top - \hat{U}_1 \hat{O}_2 \hat{O}_2^\top \hat{U}_1^\top \right\|_2, \\
&= \left\| U_1 U_1^\top - \hat{U}_1 \hat{U}_1^\top - U_1 O_1 O_1^\top U_1^\top - \hat{U}_1 \hat{O}_1 \hat{O}_1^\top \hat{U}_1^\top \right\|_2, \\
&\leq \left\| U_1 U_1^\top - \hat{U}_1 \hat{U}_1^\top \right\|_2 + \left\| U_1 O_1 O_1^\top U_1^\top - \hat{U}_1 \hat{O}_1 \hat{O}_1^\top \hat{U}_1^\top \right\|_2.
\end{aligned}
$$

The first term is bounded by Lemma S.3, while the second term is bounded using [5, Theorem 2.5]:

$$
\begin{aligned}
\left\| U_1 O_1 O_1^\top U_1^\top - \hat{U}_1 \hat{O}_1 \hat{O}_1^\top \hat{U}_1^\top \right\|_2 &\leq \left\| \left( (M^\top)^\dagger A^{\bullet m-n}_{[k-1]} - (\hat{M}_K^\top)^\dagger \hat{A}^{\bullet m-n}_{[k-1]} \right) \left( (M^\top)^\dagger A^{\bullet m-n}_{[k-1]} \right)^\dagger \right\|_2, \\
&\leq \left\| (M^\top)^\dagger A^{\bullet m-n}_{[k-1]} - (\hat{M}_K^\top)^\dagger \hat{A}^{\bullet m-n}_{[k-1]} \right\|_2 \left\| \left( (M^\top)^\dagger A^{\bullet m-n}_{[k-1]} \right)^\dagger \right\|_2.
\end{aligned}
$$

Let $\eta_1 \geq \cdots \geq \eta_K$ be the singular values of $(M^\top)^\dagger A^{\bullet m-n}$, and $\nu_1 \geq \cdots \geq \nu_{k-1}$ the singular values of $(M^\top)^\dagger A^{\bullet m-n}_{[k-1]}$. We have $\left\| \left( (M^\top)^\dagger A^{\bullet m-n} \right)^\dagger \right\|_2 = \frac{1}{\eta_K}$ and $\left\| \left( (M^\top)^\dagger A^{\bullet m-n}_{[k-1]} \right)^\dagger \right\|_2 = \frac{1}{\nu_{k-1}}$. Moreover, by [10, Theorem 1], we have $\nu_{k-1} \geq \eta_K$, thus $\left\| \left( (M^\top)^\dagger A^{\bullet m-n}_{[k-1]} \right)^\dagger \right\|_2 \leq \left\| \left( (M^\top)^\dagger A^{\bullet m-n} \right)^\dagger \right\|_2$. Furthermore (S.114) implies that $\left( (M^\top)^\dagger A^{\bullet m-n} \right)^\dagger = \Lambda (A^{\bullet n})^\top$, therefore,

$$
\left\| \left( (M^\top)^\dagger A^{\bullet m-n}_{[k-1]} \right)^\dagger \right\|_2 \leq \left\| \Lambda (A^{\bullet n})^\top \right\|_2 \leq \max_{i \in [K]} |\lambda_i| \sqrt{\|G_n\|_2}.
$$

On other hand, we have

$$
\begin{aligned}
\left\| (M^\top)^\dagger A^{\bullet m-n}_{[k-1]} - (\hat{M}_K^\top)^\dagger \hat{A}^{\bullet m-n}_{[k-1]} \right\|_2 & \\
&\leq \left\| \left( (M^\top)^\dagger - (\hat{M}_K^\top)^\dagger \right) A^{\bullet m-n}_{[k-1]} \right\|_2 + \left\| (\hat{M}_K^\top)^\dagger \left( A^{\bullet m-n}_{[k-1]} - \hat{A}^{\bullet m-n}_{[k-1]} \right) \right\|_2 \\
&\leq \left\| (M^\top)^\dagger - (\hat{M}_K^\top)^\dagger \right\|_2 \left\| A^{\bullet m-n}_{[k-1]} \right\|_2 + \left\| (\hat{M}_K^\top)^\dagger \right\|_2 \left\| A^{\bullet m-n}_{[k-1]} - \hat{A}^{\bullet m-n}_{[k-1]} \right\|_2.
\end{aligned}
$$

The term $\left\| (M^\top)^\dagger - (\hat{M}_K^\top)^\dagger \right\|_2$ is bounded by Lemma S.38,

$$
\left\| A^{\bullet m-n}_{[k-1]} \right\|_2 \leq \left\| A^{\bullet m-n} \right\|_2 = \sqrt{\|G_{m-n}\|_2},
$$

and $\left\| (\hat{M}_K^\top)^\dagger \right\|_2 = \frac{1}{\sigma_K(\hat{M}_K)} \leq \frac{1}{\sigma_K(M) - \Delta_M}$. Finally,

$$
\begin{aligned}
\left\| A^{\bullet m-n}_{[k-1]} - \hat{A}^{\bullet m-n}_{[k-1]} \right\|_2 &\leq \left\| A^{\bullet m-n}_{[k-1]} - \hat{A}^{\bullet m-n}_{[k-1]} \right\|_F = \sqrt{\sum_{i=1}^{k-1} \left\| a_i^{\otimes m-n} - \hat{a}_i^{\otimes m-n} \right\|_F^2} \\
&\leq \sqrt{(m-n) \sum_{i=1}^{k-1} \|a_i - \hat{a}_i\|_2^2},
\end{aligned}
$$

where we used (S.115) in the last line. $\qquad\square$

We can now prove our guarantee for end-to-end symmetric tensor decomposition using SPM.

*Proof of Theorem S.37.* First, to show that Algorithm 1 picks the correct tensor rank $K$, we use the assumptions of Theorem S.37 and Weyl's inequality to obtain

$$
\sigma_K(\hat{M}) \geq \sigma_K(M) - \Delta_M > \alpha > \Delta_M \geq \sigma_{K+1}(\hat{M}).
$$

Then, we show the bound on $\|\hat{a}_k - a_k\|$ using induction on $k$. When $k = 1$, we have $\hat{\mathcal{A}}_1 = \hat{\mathcal{A}}$. Then, Lemma M.1 implies that $\|P_{\hat{\mathcal{A}}} - P_{\mathcal{A}}\| \leq \hat{\Delta}_{\mathcal{A}}$. Since $\hat{a}_1$ was returned by the POWER METHOD, it is a second order critical point of $F_{\hat{\mathcal{A}}}$, and Algorithm 1 asserts that $F_{\hat{\mathcal{A}}}(\hat{a}_i) > \tau \geq \ell(\tilde{\Delta}_{\mathcal{A}}) \geq \ell(\hat{\Delta}_{\mathcal{A}})$. Therefore $\hat{a}_1$ is a second order critical point in the level set stated in Theorem M.7 / M.16, which implies that there exists $\pi(1) := i \in [K]$ and $s_1 \in \{-1, 1\}$ such that $\|\hat{a}_1 - s_1 a_{\pi(1)}\| \leq \sqrt{\frac{2\hat{\Delta}_{\mathcal{A}}}{n}}$. If $k > 1$, let $\pi([k-1]) = \{\pi(i), i \in [k-1]\}$ and $\mathcal{A}_k = \text{span}\{a_i^{\otimes n}, i \in [K] \backslash \pi([k-1])\}$. Since the subspace deflation step in Algorithm 1 does not depend on the sign of $\hat{a}_i$, $i \in [k-1]$, we can flip the sign in the bound provided in Proposition S.39

$$\left\| P_{\mathcal{A}_k} - P_{\hat{\mathcal{A}}_k} \right\|_2 \leq \frac{\Delta_M}{\sigma_K(M) - \Delta_M} \left( 1 + \max_{i \in [K]} |\lambda_i| \sqrt{\|G_n\|_2 \|G_{m-n}\|_2} \left( \frac{2}{\sigma_K(M)} + \frac{1}{\sigma_K(M) - \Delta_M} \right) \right)$$

$$+ \max_{i \in [K]} |\lambda_i| \sqrt{\|G_n\|_2} \frac{1}{\sigma_K(M) - \Delta_M} \sqrt{(m-n) \sum_{i=1}^{k-1} \|a_{\pi(i)} - s_i \hat{a}_i\|^2}$$

$$\leq \hat{\Delta}_{\mathcal{A}} \left( 1 + \max_{i \in [K]} |\lambda_i| \sqrt{\|G_n\|_2 \|G_{m-n}\|_2} \frac{4}{\sigma_K(M)} \right)$$

$$+ \max_{i \in [K]} |\lambda_i| \sqrt{\|G_n\|_2} \frac{2}{\sigma_K(M)} \sqrt{\frac{2(m-n)(k-1)}{n}} \sqrt{\hat{\Delta}_{\mathcal{A}}}$$

$$\leq \hat{\Delta}_{\mathcal{A}} \left( 1 + \max_{i \in [K]} |\lambda_i| \sqrt{\|G_n\|_2 \|G_{m-n}\|_2} \frac{4}{\sigma_K(M)} \right)$$

$$+ \max_{i \in [K]} |\lambda_i| \sqrt{\|G_n\|_2} \frac{2}{\sigma_K(M)} \sqrt{\frac{2(m-n)K}{n}} \sqrt{\hat{\Delta}_{\mathcal{A}}}$$

$$= \tilde{\Delta}_{\mathcal{A}},$$

where we set

$$C_1 = 1 + \max_{i \in [K]} |\lambda_i| \sqrt{\|G_n\|_2 \|G_{m-n}\|_2} \frac{4}{\sigma_K(M)}, \tag{S.117}$$

and

$$C_2 = \max_{i \in [K]} |\lambda_i| \sqrt{\|G_n\|_2} \frac{2}{\sigma_K(M)} \sqrt{\frac{2(m-n)K}{n}}. \tag{S.118}$$

Then, since $F_{\hat{\mathcal{A}}_k}(\tilde{a}_k) > \tau \geq \ell(\tilde{\Delta}_{\mathcal{A}})$, and $\tilde{a}_k$ is a second order critical point of $\hat{\mathcal{A}}_k$, Theorem M.7 / M.16 implies that there exists $\pi(k) \in [K] \backslash \pi([k-1]$ and $s_k \in \{-1, 1\}$, such that $\|\hat{a}_k - s_k a_{\pi(k)}\| \leq \sqrt{\frac{2\tilde{\Delta}_{\mathcal{A}}}{n}}$. In particular, it is shown in the proof of both theorems that $\hat{a}_k$ lies in a spherical cap centered around $s_k a_{\pi(k)}$ where $F_{\hat{\mathcal{A}}_k}$ is concave. However, in both theorem statements, the radius of the spherical cap where concavity holds is a decreasing function of $\Delta_{\mathcal{A}}$, therefore $\hat{a}_k$ is also in a spherical cap centered around $s_k a_{\pi(k)}$ where $F_{\hat{\mathcal{A}}}$ is concave. Applying the POWER METHOD with the functional $F_{\hat{\mathcal{A}}}$, using $\tilde{a}_i$ as a starting point will then converge for the local maxima in this concave region, and the bound for $\|\hat{a}_k - s_k a_{\pi(k)}\|$ follows.

Finally, the error bound for $\hat{\lambda}_k$ follows from the bound on $\|\hat{a}_k - s_k a_{\pi(k)}\|$ and Proposition S.39. □

We conclude with a bound on the error in terms of the scalar coefficients $\lambda_i$, which follows from a bound on $\sigma_K(M)$.

**Lemma S.40.** *With $\sigma_K(M)$ defined and under the same conditions of Theorem S.37, it holds*

$$\sigma_K(M) \geq \frac{\min_i |\lambda_i|}{\sqrt{\|G_n^{-1}\|_2 \|G_{m-n}^{-1}\|_2}} \tag{S.119}$$

*Proof.* Let $\mu_j(A)$ denote the $j$-th algorithm (in descending order) of $A$, that is $\mu_1(A) \geq \mu_2(A) \geq \cdots$. We first show that for all matrices $A_{n \times m}$ and $B_{n \times n}$ such that $m \geq n$ and $B$ is symmetric and positive definite, we have

$$\mu_n(A^\top B A) \geq \mu_n(B) \mu_n(A^\top A)$$

In fact, we have $A^\top BA = \mu_n(B)A^\top A + A^\top(B - \mu_n(B)I)A$, therefore by Weyl's inequality [13]:

$$\mu_n(A^\top BA) \geq \mu_n(B)\mu_n(A^\top A) + \mu_n(A^\top(B - \mu_n(B)I)A) \geq \mu_n(B)\mu_n(A^\top A).$$

Applying this twice, we have

$$\sigma_K(M) = \sigma_K\left(A^{\bullet n}\Lambda(A^{\bullet m-n})^\top\right) = \sqrt{\mu_K\left(A^{\bullet n}\Lambda(A^{\bullet m-n})^\top A^{\bullet m-n}\Lambda(A^{\bullet n})^\top\right)},$$

$$\geq \sqrt{\mu_K(G_{m-n})}\sqrt{\mu_K\left(A^{\bullet n}\Lambda^2(A^{\bullet n})^\top\right)},$$

$$\geq \min_i |\lambda_i|\sqrt{\mu_K(G_{m-n})}\sqrt{\mu_K\left(A^{\bullet n}(A^{\bullet n})^\top\right)} = \min_i |\lambda_i|\sqrt{\sigma_K(G_n)\sigma_K(G_{m-n})}.$$

Now the lemma follows from $\sigma_K(G_n) = 1/\|G_n^{-1}\|_2$. $\qquad\qquad\square$

**Remark S.41** (Small $\lambda_i$). Lemma S.40 indicates that the smaller $\min_i |\lambda_i|$ is, the larger is the error of the decomposition obtained by SPM. For instance, the lemma can be used to obtain the bound

$$\hat{\Delta}_{\mathcal{A}} \leq \frac{\Delta_M\sqrt{\|G_n^{-1}\|_2\|G_{m-n}^{-1}\|_2}}{\min_i |\lambda_i| - \Delta_M\sqrt{\|G_n^{-1}\|_2\|G_{m-n}^{-1}\|_2}},$$

which suggests a smaller $\min_i |\lambda_i|$ increases the subspace error $\Delta_{\mathcal{A}}$. This is also evident in the error arising from deflation. Substituting Lemma S.40 into the definition of $C_1$ and $C_2$ in Theorem S.37,

$$C_1 \leq 1 + 4\frac{\max_i |\lambda_i|}{\min_i |\lambda_i|}\sqrt{\kappa(G_n)\kappa(G_{m-n})},$$

and

$$C_2 \leq 2\frac{\max_i |\lambda_i|}{\min_i |\lambda_i|}\sqrt{\kappa(G_n)\|G_{m-n}^{-1}\|_2}\sqrt{\frac{2(m-n)K}{n}}.$$

Here we denoted the condition number of a matrix $A$ by $\kappa(A) := \|A\|_2 \|A^{-1}\|_2$.

Finally, the issue also shows up in the bound of the tensor coefficient error of Theorem S.37:

$$\left|\frac{s_k^m}{\lambda_{\pi(k)}} - \frac{1}{\hat{\lambda}_k}\right| \leq \frac{\sqrt{\|G_n^{-1}\|_2\|G_{m-n}^{-1}\|_2}}{\min_i |\lambda_i|}\left(2\sqrt{m/n}\hat{\Delta}_{\mathcal{A}} + 4\hat{\Delta}_{\mathcal{A}}\right).$$