# OpenReview forum: "Landscape analysis of an improved power method for tensor decomposition"
_NeurIPS.cc/2021/Conference — NeurIPS 2021 Poster_

### Official Review · Reviewer_pP73 · 2021-07-08

**Rating:** 7
**Confidence:** 4

**Summary:**

This paper studies the non-convex landscape of the SPM-P optimization objective (in page 2) for tensor decomposition. Under both deterministic and random models of the components, the work shows that all second-order critical points coincide with the ground truth vectors (up to a sign difference) with different levels of overcompleteness. The result can also be generalized to the noisy cases under certain assumptions.


**Limitations And Societal Impact:**

Described in the question part in the main review.


**Main Review:**

The work studies the nonconvex landscape of the symmetric tensor decomposition problem under the overcomplete setting. Overall this is a well-written and well-presented paper, the geometrical landscape analysis on the landscape provides new insight to tensor decomposition problems.

* Pros:
This paper experimentally demonstrates that there are no spurious local maximizers of the SPM-P objective (in Figure 1) and corroborates their experiments with extensive theoretical results (Theorem 6 for the deterministic case and Theorem 15 for the noisy case).

* Questions: The result for the random model critically depends on a conjecture (Conjecture 14). Also, the landscape analysis is shown in super level set which a random initialization cannot be achieved. Can the author compare the size of the super level set with that in [18] and discuss it? On the other hand, are all other saddle points to be strict saddle points?

In Theorem 6, it seems that the assumption requires \rho_2 and \rho_n to be at the scale of O(1/n^2), does this happen in general? Or what other assumptions do we require for a_1,a_2,\dots,a_K so that \rho_2 and \rho_n satisfy the assumption in (eq. 10)? Perhaps the authors could either provide some theoretical justifications of empirical evidence?

* Relation to Literature. There are several recent related results that are missing in reference, and it would better to be discussed:

``Maziar Sanjabi, Sina Baharlouei, Meisam Razaviyayn, Jason D. Lee When Does Non-Orthogonal Tensor Decomposition Have No Spurious Local Minima?''
``Qing Qu, Yuexiang Zhai, Xiao Li, Yuqian Zhang, Zhihui Zhu Geometric Analysis of Nonconvex Optimization Landscapes for Overcomplete Learning''
In particular, the latter work considered the fourth-order overcomplete tensor decomposition problem under tight frame and incoherent assumptions, with application to overcomplete dictionary learning. The work showed a global landscape when overcompleteness is a constant, but with a different PM loss function.


**Time Spent Reviewing:**

10 hours

---

> ### Author Response · Authors · 2021-08-10
> **Thanks for the detailed review! Scenarios covered by the deterministic conditions, and the connection to existing literature about PM-P for n=2**
>
> We thank the reviewer for the thorough review of our submission.
> > Also, the landscape analysis is shown in super level set which a random initialization cannot be achieved. Can the author compare the size of the super level set with that in [18] and discuss it?}
> * That is correct, a starting point sampled from $\textrm{Unif}(\mathbb{S}^{D-1})$ does not lie within the specified superlevel set with high probability. Due to the presence of the inverse Grammian matrix $G_n^{-1}$ in the SPM objective, see Lemma 3, it is not straightforward to analytically compute the expected value of $u \sim \textrm{Unif}(\mathbb{S}^{D-1})$, as compared to the PM objective. However, based on Figure S.2 in the supplementary material, we conjecture that $\mathbb{E}[F_{\mathcal{A}}(x_0)] = \mathcal{O}\left(\frac{K}{D^n}\right)$ for $a_1,\ldots,a_K \sim \textrm{Uni}(\mathbb{S}^{D-1})$ and $u_0 \sim \textrm{Unif}(\mathbb{S}^{D-1})$, see also Remark S.28. Based on this conjecture, our results fall short by a factor of order $\mathcal{O}(\textrm{polylog}(K))$ of achieving a global guarantee.  We will move Figure S.2 to the main body in the final version.
> The super level set achieved in [18] is slightly better, because the authors there are only a factor of $1 + o(1)$ short of achieving a global guarantee. However [18] studies the PM objective and achieves this improvement through a technically-involved argument using the so-called Kac-Rice formula, which is used to count the expected number of roots of a random polynomial over the unit sphere. For our case of the SPM objective, the presence of the inverse Grammian $G_n^{-1}$, see Lemma 3, poses an additional challenge for employing the Kac-Rice formula. Thus far, we were not able to extend the PM analysis. Furthermore, the analysis in [18] is limited to the case $n=2$, and extending the Kac-Rice analysis to larger $n$, even for the PM objective, appears to be challenging.
> To highlight this issue, we will move Remark S.28 from the supplementary material into the main body.
>
> > On the other hand, are all other saddle points to be strict saddle points?
> * Indeed this is true, thank you for pointing it out.  For both of our theorems 6 and 15, our proofs (as is) show that there are only $2K$ *second-order critical points* in the specified superlevel sets. Thus all saddle points are strict there. We will update the statement in the final version.
>
> > In Theorem 6, it seems that the assumption requires $\rho_2$ and $\rho_n$ to be at the scale of $\mathcal{O}(1/n^2)$, does this happen in general? Or what other assumptions do we require for $a_1,a_2,\dots,a_K$ so that $\rho_2$ and $\rho_n$ satisfy the assumption in (eq. 10)? Perhaps the authors could either provide some theoretical justifications of empirical evidence?}
> - First, we want to point out that we can, without loss of generality, always assume that $n=2$ if the subspace $\mathcal{A}$ arises from flattening a rank $\mathcal{O}(D^2)$ tensor $T$. This is because we can flatten the tensor into a short matrix of size $D^{(m-2)\times 2}$ instead of a square or nearly-square matrix $D^{n \times n}$ matrix, and then we can consider the subspace $\textrm{Span}(a_i^{\otimes 2} : i =1,\ldots,K)$ spanned by right singular vectors. Therefore the constraint on $\rho_2$ is not as severe as it may seem since frequently we can take $n=2$. We analyzed the SPM objective for arbitrary $n \geq 2$, because larger $n$ is required for higher-rank tensors $T$, and because in some other applications besides tensor decomposition, the SPM objective may be relevant and one may not be able to choose the subspace there.   For example, see \emph{Nakatsukasa, Yuji, Tasuku Soma, and André Uschmajew. "Finding a low-rank basis in a matrix subspace." Mathematical Programming 162.1 (2017): 325-361} where the subspace $\mathcal{A}$ is given.   We will add a corresponding remark in the final version.
> Focusing on the tensor decomposition case, we can often consider $n = 2$. The most prototypical examples are the following (partially been described in Remark 7):
>   * Perturbations of tensors with orthogonal components.
>   * Unit norm tight frame (UNTF) (especially equiangular tight frames) of size $K$ in $D$ dimensions have $\rho_2 = (K-D)/D$, see *Mixon, D. G. "Unit norm tight frames in finite-dimensional spaces." Finite Frame Theory: A Complete Introduction to Overcompleteness 93 (2016): 53.* or *Casazza, Peter G., Dan Redmond, and Janet C. Tremain. "Real equiangular frames." 2008 42nd annual conference on information sciences and systems. IEEE, 2008.*. We can tolerate a degree of overcompleteness $K = CD$ for some $C > 1$, so that $\rho_2$ remains smaller than the asserted bound for $n =2$. Moreover, perturbations of such systems of vectors are also allowed.
>   * Low rank mutually incoherent systems with $\mu :=  \max_{i\neq j}|\langle a_i,a_j\rangle| \ll 1/\sqrt{D}$. As described in Remark 7, in this case we have $\rho_2 \leq (K-1)\mu  \approx (K-1)/\sqrt{D}$.
>   * Low-rank random tensors with $K \approx CD$ as mentioned in Remark 7. We can achieve arbitrarily small $\delta_2 > 0$ with high probability.
>   Given this range of scenarios, we feel that Theorem 6 is a good complement to Theorem 15.
> Last we mention that using $\rho_n \leq \rho_2$, our deterministic condition is implied by an upper bound condition on $\rho_2$.  That is easily checkable given a system $\{ a_i : i \in [K]\}$, since $\rho_2$ is one less than the largest eigenvalue of $(G_2)_{i,j} = \langle a_i, a_j \rangle^2$.
>
> > Relation to Literature. There are several recent related results that are missing in reference, and it would better to be discussed:
> Maziar Sanjabi, Sina Baharlouei, Meisam Razaviyayn, Jason D. Lee When Does Non-Orthogonal Tensor Decomposition Have No Spurious Local Minima?'' Qing Qu, Yuexiang Zhai, Xiao Li, Yuqian Zhang, Zhihui Zhu Geometric Analysis of Nonconvex Optimization Landscapes for Overcomplete Learning'' In particular, the latter work considered the fourth-order overcomplete tensor decomposition problem under tight frame and incoherent assumptions, with application to overcomplete dictionary learning. The work showed a global landscape when overcompleteness is a constant, but with a different PM loss function.
> * We thank the reviewer for pointing out the additional references and we will incorporate them into a revised version. We would also like to take the opportunity to put our paper into proper context with respect to the mentioned references, as both references justify our Theorem 6 and show that it is comparable to other considered regimes in the literature.
>   * Regarding *When Does Non-Orthogonal Tensor Decomposition Have No Spurious Local Minima?*: The authors consider the PM objective for $m = 4$ and perform a deterministic analysis of local maximizers under a mutual coherence condition (also an analysis about deflation, which we omit in this brief discussion as it is less relevant for Theorems 6 and 15 in our manuscript). Assuming a mutual coherence of $1/\sqrt{D}$ for simplicity (e.g. isotropic random vectors),  their condition  $K/\sqrt{D} \leq 0.005$ is a lot more restrictive than either one of Theorem 6 or Theorem 15 in our manuscript. In fact, this scenario is a possible subcase of Theorem 6; see the earlier comment about mutually incoherent systems. We further want to emphasize that the PM objective suffers from the problem that global optimizers do not coincide with the true tensor components (even in a noiseless case), which we view as a major problem with the PM objective. While SPM is more challenging to analyze, the lack of an additional bias with respect to the true components is, from our point of view, a big advantage.
>   * Regarding *Geometric Analysis of Nonconvex Optimization Landscapes for Overcomplete Learning*: First, we note that the analysis also considers the PM objective, so the same comments between the difference of PM and SPM from before apply. In this work the authors consider UNTFs with constant oversampling $K = CD$ for some $C\approx 3$ and an additional bound for the mutual incoherence. They globally characterize the landscape of the PM objective, showing that all there are no spurious local maximizers. We note that the studied regime is comparable to a subcase in Theorem 6, where in the case of UNTFs we also allow a degree of overcompleteness (though with a constant $C$ closer to $1$, as mentioned in an earlier comment). Hence, we believe that our Theorem 6 is similar in terms of the characterized regime, and goes beyond with respect to analyzing effects of perturbed problems (arising from decomposing perturbed tensors). The gap between the overcompleteness constants may be, because we only rely on $\rho_2$, but we do not make explicit use of a `small mutual incoherence condition'.
> During the review, we also came across the paper *Auddy, Arnab, and Ming Yuan. "Perturbation bounds for orthogonally decomposable tensors and their applications in high dimensional data analysis." arXiv preprint arXiv:2007.09024 (2020)*, which performs an analysis of robust decompositions of approximately Odeco tensors (tensors with orthogonal components). Given that we also allow a degree of noise, we believe the reference should also be added to our manuscript.  We will add a bit to the prior theory section to better contextualize our results.  Also, please see our responses to Reviewer #1.

---

### Official Review · Reviewer_diJc · 2021-07-16

**Rating:** 6
**Confidence:** 2

**Summary:**

This paper analyzes a different optimization problem for the task of finding one scaler/vector pair.

**Limitations And Societal Impact:**

Limitations are not discussed.

**Main Review:**

I am not familiar with this kind of work. I have one question concerns the deflation procedure.

So far the paper considers finding one scaler/vector pair and guarantees are for the second-order stationary points. After finding one scaler/vector pair, due to the error in the pair found, simple deflation procedures, finding one pair each time and deflating the tensor by subtracting the rank 1 tensors found so far, will accumulate errors. Are there any guarantees for this deflation procedure? In remark 18, the author suggests a procedure in doing the deflation. Can the author say something about the technical difficulty in proving the correctness of the procedure proposed in Remark 18?


**Time Spent Reviewing:**

3

---

> ### Author Response · Authors · 2021-08-10
> **Our focus is on the non-convex optimization landscape, but we have error analysis for deflation**
>
> > So far the paper considers finding one scaler/vector pair and guarantees are for the second-order stationary points. After finding one scaler/vector pair, due to the error in the pair found, simple deflation procedures, finding one pair each time and deflating the tensor by subtracting the rank 1 tensors found so far, will accumulate errors. Are there any guarantees for this deflation procedure? In remark 18, the author suggests a procedure in doing the deflation. Can the author say something about the technical difficulty in proving the correctness of the procedure proposed in Remark 18?
> * Indeed our main results, Theorems 6 and 15, focus on the recovery of one scalar/vector pair and the associated non-convex landscape; deflation is not the primary focus of this work. This is due to length constraints and because the deflation step was described in [22]. However we will add some description of deflation to the final version. Motivated by your question, we have obtained a bound on the accumulation of errors incurred by deflation. We would now like to clarify two points:
>   * In the noise-free case where $\Delta_{\mathcal{A}} = 0$, we note that every non-spurious local maximizer coincides with one of the true tensor components (spurious local maximizers have objective value close to $0$ and can thus be discarded by checking the objective). Furthermore, projected gradient ascent converges with a power rate to a second-order critical point, see [22], so that we can recover true tensor components exactly (up to rounding errors). If we use exact vectors in the deflation, the method is *provably* correct, i.e., we can exactly reduce the original tensor. This guarantees correctness of the procedure.
>   * The noisy case with $\Delta_{\mathcal{A}} > 0$ is a little more subtle. Even if we can ensure (by checking the objective after finding a second-order critical point) that we only use non-spurious local maximizers in the deflation process, in each deflation step we may accumulate an additional error. In the revised version, we will include a lemma that provably bounds the error accumulated through the deflation, and upper bounds the permissible input error $\Delta_{\mathcal{A}}$ such that the SPM algorithm, with the procedure proposed in Remark 18 is guaranteed to succeed with bounded error, under the regimes of Theorems 6 and 15.  The following result may not be not sharp, but illustrates how our analysis implies end-to-end results for tensor decomposition:
>
>     **Lemma** (Deflation bounds, informal)
>     *Let $T = \sum_{i=1}^{K}\lambda_i a_i^{\otimes 2n} \in \mathcal{T}_D^{m}$ and assume that $M:=\textrm{reshape}(T, [D^{n}, D^{m-n}])$ has exactly $K$ nonzero singular values $\sigma_1(M) \geq \ldots \geq \sigma_K(M) > 0$. Let $\hat{T} \in \mathcal{T}_D^{m}$, $\hat{M} := \textrm{reshape}(\hat{T}, [D^{n}, D^{m-n}])$, and assume $\Delta_M := \| M - \hat{M} \|_2 < \frac{1}{4}\sigma_K(M)$. Then there exists constants $C_i, i=1,2,3,4$ only depending on $T$ (and not  $\hat T$ nor $\Delta_0$) such that if $C_1 \Delta_M + C_2\sqrt{\Delta_M K} < \Delta_0$, then the procedure of optimizing first in the deflated subspace with a suitable initialization, and then refining in the first, not deflated noisy subspace, recovers all tensor components with error bounded in Theorems 6 and 15, and the corresponding tensor coefficients are determined with the following error*
> $$\frac1{\lambda_i} - \frac1{\hat \lambda_i} \le C_3 \|a_i - \hat a_i\| + C_4 \Delta_{M}.$$

---

### Official Review · Reviewer_ABgz · 2021-07-17

**Rating:** 9
**Confidence:** 5

**Summary:**

In this paper, the authors consider the optimization formulation for symmetric tensor decomposition introduced in the Subspace Power Method (SPM). They analyze the non-convex optimization landscape associated with the SPM objective. The analysis focuses on the noisy tensors, and implies SPM with suitable initialization is a provable, efficient, robust algorithm for low-rank symmetric tensor decomposition. Numerical results coincide with the theoretical analysis.

The contributions is to analyze the landscape of SPM-P by characterizing all second-order critical points.

**Limitations And Societal Impact:**

Yes. The authors give some "near-global" guarantee under a random tensor model and the deterministic frame conditions.

**Main Review:**

Nowadays, high-dimensional data sets are considered as higher-order arrays. That is, tensor. Finding the CP decomposition of given tensor is very important. The authors consider the symmetric tensor decomposition introduced in Subspace Power Method.

**Time Spent Reviewing:**

10

---

> ### Author Response · Authors · 2021-08-10
> **Thank you for appreciating our work!**
>
> We are excited by your positive feedback and for highlighting the significance of our work. In particular, we did work hard to include noise in our analysis because tensors in applications are invariably noisy. Thank you for recognizing our contributions.

---

### Official Review · Reviewer_4wNn · 2021-07-21

**Rating:** 6
**Confidence:** 4

**Summary:**

This paper considers the subspace power method (SPM) for symmetric tensors.
Specifically, this paper analyzes the non-convex optimization landscape associated with the SPM objective.
The main results are two theorems, theorem 6 and theorem 15. Under some assumptions, it is shown in theorem 6  that for low rank case, nSPM-P has 2K local maximizers which are good approximations of all components. For the overcomplete tensors, similar conclusion is given. Numerical experiments is presented to validate the theoretical findings.

**Main Review:**


Major concern:
(1) Theorem 6, the requirement (10) is too restrictive.
When n is relatively large,  (10) implies that rho_2 is tiny, as a result, all components  a_i's are almost orthogonal.
To be more specific,  let us relax the requirement (10) a little bit, say rho_2 <= 1 / (8n^2).
Set  K=2 (only two components a_1 and a_2), assume the angle between a_1 and a_2 is pi/3,
then it is easy to see that rho_2 = 2 * (cos pi/6)^2  - 1 = 0.5.
However, 0.5 > 1 / (8n^2) for all n>= 2.
When all components  a_i's are almost orthogonal, it is not surprising to draw the conclusion of Theorem 6.
I would like to raise my score if the authors are able to convince me that the requirement (10) is not restrictive.

(2) Theorem 15 is established under the assumption A3, which is only shown for n=2. The authors formulate the result as conjecture 14.
It is OK to make a such conjecture, but the readers may not buy it, especially c_2 is independent of K and D. It is better to provide the evidence (even numerical evidence) in the main text of this paper.

More comments

(1) The notation for || ||, || ||_F is unclear. In the notation, || || is declared to be the F-norm, but || ||_F still appears in the subsequent discussion. This leads to some confusing in Lemma 1, see details in (2).

(2) Lemma 1, eqn (3).  If F norm is used, \Delta_T in the denominator should be replaced by ||M - \hat M||_2.
If 2-norm is used,  \Delta_T in both numerator and denominator should be replaced by ||M - \hat M||_2.
The inequality between Lines 38 and 39 in the supplementary, the term pi/2 can be removed.
Therefore, the bound in Lemma 1 can be improved.

(3) Lines 125 and 126, it is not appropriate to cite [9] here. This deflation procedure is closely related to the spectral theory of symmetric tensors, see the following paper for example.
L. Qi, “The best rank-one approximation ratio of a tensor space”,SIAM Journalon Matrix Analysis and Applications, 32 (2011) 430-432.
Furthermore, it is unclear that such a procedure is able to find all components.

(4) Line 241, remark 18, is it possible to show that \tilde{a}_{\ell +1} falls into the a neibourhood of the true component, nSPM-P is able to recover?

(5) To make the numerical experiments more convincing, try larger n, rather than simply set m=4 (n=2).

I am satisfied with the authors' replies, and raised the score.


**Time Spent Reviewing:**

5

---

> ### Author Response · Authors · 2021-08-10
> **Value of Theorem 6 in addition to Theorem 15, guaranteed deflation result, numerics for higher-order tensors, and Conjecture 14**
>
> We thank the reviewer for taking the time to read our paper, and for the thorough feedback.
> > Theorem 6, the requirement (10) is too restrictive... I would like to raise my score ...
> - We thank the reviewer for this honest criticism of Theorem 6 and we agree that it is not  characterizing all possible deterministic systems for which global and local maxima of SPM coincide. However, we would like to defend the result as follows, and explain why we added it as a complement to Theorem 15.
> First, we would like to mention that one can construct systems of vectors with $\rho_2=0.5$, where spurious local maximizers exist. For instance, consider $a_1 = [1,0]$, $a_2 = [-1/2, \sqrt{3}/2]$, $a_3 = [-1/2, -\sqrt{3}/2]$, giving $\rho_2 = 0.5$. In this case $\mathcal{A}$, spans the full subspace of symmetric rank-one tensors in $\mathbb{R}^2$, hence, the energy landscape of the SPM objective is equal to the constant function $1$ over the unit sphere, and all points are local maxima.
> Secondly, and this is a point that we propose to include as a remark in the final version, we can, without loss of generality, always assume that $n=2$ if the subspace $\mathcal{A}$ arises from flattening a rank $\mathcal{O}(D^2)$ tensor $T$. This is because we can flatten the tensor into a short matrix of size $D^{(m-2)\times 2}$ instead of a square or nearly-square matrix $D^{n \times n}$ matrix, and then we can consider the subspace $\textrm{Span} (a_i^{\otimes 2} : i =1,\ldots,K)$ spanned by right singular vectors. Therefore the constraint on $\rho_2$ is not as severe as it may seem since frequently we can take $n=2$.
> The third point we would like to make is that the conditions of Theorem 6 do allow a range of scenarios:
>   * Perturbations of tensors with orthogonal components.
>   * Unit norm tight frames (UNTF) (especially equiangular tight frames) of size $K$ in $D$ dimensions have $\rho_2 = (K-D)/D$.
>   * Low rank mutually incoherent systems with $\mu :=  \max_{i\neq j}|\langle a_i,a_j\rangle| \ll 1/\sqrt{D}$. As described in Remark 7, in this case we have $\rho_2 \leq (K-1)\mu  \approx (K-1)/\sqrt{D}$.
>   * Low-rank random tensors with $K \approx CD$ as mentioned in Remark 7. We can achieve arbitrarily small $\delta_2 > 0$ with high probability.
> Even for the first bullet, there has been recent and substantial work to understand the effect of perturbations; see the paper *Auddy, Arnab, and Ming Yuan. "Perturbation bounds for orthogonally decomposable tensors and their applications in high dimensional data analysis." arXiv preprint arXiv:2007.09024 (2020).*
> Furthermore, we want to emphasize that deterministic conditions for non-convex optimization are in general difficult to come by. For instance, here are the results and main assumptions for the well-studied PM objective in the papers mentioned by Reviewer #4:
>   * *Qu, Qing, Yuexiang Zhai, Xiao Li, Yuqian Zhang, Zhihui Zhu. "Geometric analysis of nonconvex optimization landscapes for overcomplete learning." International Conference on Learning Representations. 2019.*: Global characterization of local maximizers if the vectors $a_1,\ldots,a_K$ are UNTFs and additionally have pairwise small mutual incoherence (more restrictive than item 2 above).
>   * *Sanjabi, M., Baharlouei, S., Razaviyayn, M., \& Lee, J. D. (2019). When does non-orthogonal tensor decomposition have no spurious local minima?. arXiv preprint arXiv:1911.09815.* Global characterization of local maximizers if $K \leq 0.005\sqrt{D}$ based on mutual incoherence (similar to scenario 3 above, the 0.005 in the guarantee is not as tight a constant as our Theorem 6 gives).
>
>   Finally, many prior works for PM stick to fourth-order input tensors (corresponding to $n=2$ in our setup), and they assume no noise on the given tensor.
> However Theorem 6 considers a more complicated - and as we argued a more relevant - objective function than PM.  It allows for noise in the tensor which is important for real applications, and allows for general $n\geq 2$. Given these contributions to the literature, we think Theorem 6 does add value along side Theorem 15.  In a revised version, we propose to add to the prior theory section to explain the context better.
>
> > Theorem 15 is established under the assumption A3, which is only shown for $n=2$... better to provide the evidence (even numerical evidence) in the main text...
> * Thank you for the suggestion, we will move the numerical evidence previously given in the supplementary material into the main body of the paper. While these numerics look convincing to us, actually proving the conjecture for $n>2$ appears to be at the cutting edge of current random matrix theory methods.  The conjecture has strong connections to the well-posedness of $K$ tensors $a_1^{\otimes n},\ldots,a_K^{\otimes n}$, where $a_i \sim \textrm{Unif}(\mathbb{S}^{D-1})$, and such well-posedness has been partially addressed in some recent work (e.g., *Vershynin, Roman. "Concentration inequalities for random tensors." Bernoulli 26.4 (2020): 3139-3162* and *Bryson, Jennifer, Roman Vershynin, and Hongkai Zhao. "Marchenko-Pastur law with relaxed independence conditions." arXiv preprint arXiv:1912.12724 (2019)*). Unfortunately however, the analyses conducted there crucially depend on the independence of the components and/or the coordinates of each component (i.e., they cover tensors such as $a_i \otimes b_i \otimes \ldots$, with $a_i, b_i$ being independent vectors with independent coordinates). The case of $a_i^{\otimes n}$ induces additional statistical dependencies in the random tensor model. Based on our attempts, extending these prior analyses to this case presents a challenging problem.
>
> > The notation for $\Vert \cdot \Vert$, $\Vert \cdot \Vert_F$ is unclear...
> * Yes, we apologize for this. Some instances of $\Vert \cdot \Vert_F$ slipped through our hands.  To have the most unambiguous notation, we will include the subscript $F$ on all Frobenius norms throughout the main body and supplemental materials.
>
> > Lemma 1, eqn (3) ... Therefore, the bound in Lemma 1 can be improved.
> * Thanks for pointing this out. Thanks to your comment, we revisited Lemma 1 and sharpened it to directly bound the operator norm of the projection error, rather than its Frobenius norm. If $\Vert M-\hat M\Vert_2 < \mu_K(M)$, we have
> $$\Vert P_{\textrm{Im}(M)}-P_{\textrm{Im}_K(\hat M)}\Vert_2 \leq \frac{\Vert M - \hat M\Vert_2}{\mu_K(M) - \Vert M-\hat M\Vert_2}.$$
>
> > Lines 125 and 126, it is not appropriate to cite [9] here...  unclear that such a procedure is able to find all components.
> * Thank you for this reference, we will include this. For our paper on the non-convexity, the most relevant point is that after deflation is performed we will again face another instance of the problem nSPM-P. Therefore Theorems 6 and 15 may be applied again, and we again have guarantees for finding the subsequent components of the tensor.  However for completeness we propose to add more details on the deflation procedure in the main body in the final version. The exact procedure we are using is the one described in Appendix B of the reference [22].  It is shown in [22] that given a noiseless tensor $T$ and an exact CP component $a_i$, the method is guaranteed to find the correct weight $\lambda_i$ if the rank-$1$ tensors $\{a_i^{\otimes n}, i\in [K]\}$ are linearly independent (which holds Zariski-generically).  In particular letting $M = \textrm{Reshape}(T, [D^{n}, D^{m-n}])$, the deflation method sets $\lambda_i$ to be the unique scalar such that  $M - \lambda_i a_i^{\bullet n} (a_i^{\bullet (m-n)})^{\top}$ has (matrix) rank one less than that of $M$, which amounts to setting $\lambda_i = ((a_i^{\bullet (m-n)})^{\top}M^{\dagger}a_i^{\bullet n})^{-1}$.  In the noiseless case, $M$ is then replaced by $M - \lambda_i a_i^{\bullet n} (a_i^{\bullet (m-n)})$.  In the noisy case, for better efficiency and stability, the deflation procedure differs slightly in the update of $M$ (but matches the last sentence in the noiseless case) according to [22]. While deflation is not the main focus on this paper, we have obtained a naive bound for error propagation under deflation which we'll add to the final version; please see "Lemma 1" in our response to Reviewer #3.
> Deflation issues aside, we would like to remark that nSPM-P itself is actually able to find all components.  This is because the reference [22] proved each CP component $\pm a_i$ is locally attractive for projected gradient ascent applied to the SPM functional, under Zariski genericity assumptions. In particular, there is not the phenomenon of a ``dominant eigenvector" that we converge to for almost all initializations (as in the classical power method for matrices).
>
> > Line 241, remark 18, is it possible to show that $\tilde{a}_{\ell +1}$ falls into the a neighbourhood of the true component, nSPM-P is able to recover?
> * Yes. After $\ell$ steps of deflation we face another instance of the problem nSPM-P, so our theorems apply to the deflated tensor which is a noisy copy of $\sum_{i=\ell+1}^K \lambda_i a_i^{\otimes m}$. However to make this more explicit in terms of the originally inputted tensor $\hat{T}$, we have obtained a bound for effect of deflation on the subspace error $\Delta_{\mathcal{A}}$.  Please see "Lemma 1" in the response to Reviewer #3. We will include this in the final version.  The consequence is that our analysis implies bounds end-to-end bounds for symmetric tensor decomposition.
>
> > To make the numerical experiments more convincing, try larger n, rather than simply set m=4 (n=2).
> * Thank you for this suggestion, we have tested $n=3,4$.  The results show more variance but are qualitatively very similar to the case $n=2$.  If you would like to see the new plots, please check out this anonymous link (with anonymous browsing as well): [https://imgur.com/a/6QOQxpr](https://imgur.com/a/6QOQxpr)

---

> > ### Comment · Reviewer_4wNn · 2021-09-10
> > **raise score**
> >
> > The authors' replies are satisfactory, I would like to raise my score from 5 to 6.

---

### Decision · Program_Chairs · 2021-09-27

**Decision:**

Accept (Poster)

**Comment:**

The paper studies the symmetric tensor decomposition problem, in which we are given a (noisy) superposition of outer products, and the goal is to recover these generating components. The paper analyzes the “subspace power method”, which seeks local maximizers of the norm of the projection of a component onto the span of a certain reshaping of the input tensor. For orthogonal tensors, this coincides with the standard power method; for general tensors with correlated components, the subspace power method has advantages: global maximizers are known to coincide with the generating components. In contrast, previous work does not prove that the problem lacks local maximizers, and as a consequence, does not establish algorithmic recovery guarantees.

 The contribution of this work is a landscape analysis of the SPM, which proves that under certain deterministic incoherence conditions, there are no suboptimal maximizers in a certain super level set of the objective function. For low rank incoherent tensors (r = O(D)), this establishes that local maximizers are global. For random tensors with rank approaching the conjectured barrier O(D^{m/2}) for efficient methods, this result rules out suboptimal maximizers with large objective function (nonvanishing objective function in certain scalings of the problem parameters). The latter result can be compared to analyses of Ge and Ma for the traditional power method.

Initial reviews were generally positive, acknowledging the theoretical contributions of the work, while also raising questions about (i) the strength of the paper’s incoherence conditions, and (ii) the semi-global nature of the paper’s results for random overcomplete tensors. The reviewers found the authors’ response on these points generally satisfying; the final consensus is to accept the paper. The AC concurs — the paper contributes a novel analysis of a method that seems to have practical advantages in handling correlated tensors. While there are many remaining issues (global results for highly over complete tensors?), the paper makes a contribution to the literature in this area.